# Hybrid calculation of hadronic vacuum polarization in muon $g − 2$ to 0.48%

A. Boccaletti[1,2], Sz. Borsanyi[1], A. Cotellucci[2], M. Davier[3], Z. Fodor[1,2,4,5,6,7 ✉], F. Frech[1], A. Gérardin[8], D. Giusti[2,9], A. Yu. Kotov[2], L. Lellouch[8], Th. Lippert[2], A. Lupo[8], B. Malaescu[10], S. Mutzel[8,11], A. Portelli[12,13], A. Risch[1], M. Sjö[8], F. Stokes[2,14], K. K. Szabo[1,2], B. C. Toth[1,2], G. Wang[8] & Z. Zhang[3]

For 50 years, the standard model of particle physics has been very successful in describing subatomic phenomena. In the past quarter of a century, this was challenged by a mismatch between its predictions and precision measurements of the anomalous magnetic moment of the muon, $a_\mu$. This disagreement was eventually reconciled, first through a determination in an ab initio lattice calculation[1] of the most uncertain theoretical contribution, the leading-order hadronic vacuum polarization (LO-HVP), $a_\mu^{\text{LO-HVP}}$, and subsequently by experimental results[2] and updates of the reference standard-model predictions using lattice results for $a_\mu^{\text{LO-HVP}}$ (ref. 3). Here we present a new calculation for this crucial quantity, obtaining $a_\mu^{\text{LO-HVP}} = 715.1(2.5)(2.3)[3.4] \times 10^{-10}$. This reduces the uncertainty by a factor of 1.6 compared with our earlier computation[1]. We use a hybrid approach that includes a small, long-distance contribution from experiments in a low-energy regime in which they all agree. Our approach combines the strengths of experimental and lattice data in different energy ranges, achieving better precision than with either alone. Our lattice quantum chromodynamics (QCD) simulations are performed on finer lattices than in ref. 1, allowing for an even more accurate continuum extrapolation. Combined with the calculations of the other standard-model contributions summarized in ref. 3, our result leads to a prediction that differs from the recent measurement of $a_\mu$ (ref. 4) by only 0.5 standard deviations. This provides a notable validation of the standard model to 11 digits.

The muon is a short-lived elementary particle with spin 1/2 and a mass 207 times larger than that of the electron. Both particles create a magnetic field around them, characterized by a magnetic dipole moment. This moment is proportional to the spin and charge of the particle and inversely proportional to twice its mass. Dirac's relativistic quantum mechanics predicts that the constant of proportionality, $g_\mu$, known as the Landé factor, is precisely 2. Relativistic quantum field theory introduces further small corrections induced not only by all particles and interactions of the standard model but also potentially by yet undiscovered ones. Because muons are more massive than electrons, quantum corrections associated with heavy particles are generically much larger for the former than for the latter[5]. This increased sensitivity to the effects of possible unknown particles is the reason for the present focus on the muon. The corrections to $g_\mu$ are commonly called the anomalous magnetic moment and are quantified as $a_\mu = (g_\mu − 2)/2$.

When calculating $a_\mu$, the uncertainty comes almost exclusively from the strong interaction, described in the standard model by QCD. In particular, the dominant source of uncertainty comes from hadronic vacuum polarization (HVP) at leading order in the fine-structure constant (LO-HVP). More generally, HVP induces a modification in the propagation of a virtual photon in the vacuum, caused by the strong interaction.

Here we present a calculation of this LO-HVP contribution to $a_\mu$ ($a_\mu^{\text{LO-HVP}}$) with unprecedented accuracy. To that end, we apply numerical lattice quantum field theory techniques that allow QCD predictions to be made in the highly nonlinear regime that is relevant here. Mathematically, QCD is a generalized version of quantum electrodynamics (QED). However, QCD predicts physical phenomena that are very different from those described by QED. The Euclidean Lagrangian for a quark of mass $m$ and charge $q$ (in units of the positron charge, $e$), subject to strong and electromagnetic interactions, can be written as $\mathcal{L} = 1/(4e^2)F_{\mu\nu}F_{\mu\nu} + 1/(2g^2)\text{Tr}G_{\mu\nu}G_{\mu\nu} + \bar{\psi}[\gamma_\mu(\partial_\mu + iqA_\mu + iG_\mu) + m]\psi$, in which $F_{\mu\nu} = \partial_\mu A_\nu − \partial_\nu A_\mu$, $G_{\mu\nu} = \partial_\mu G_\nu − \partial_\nu G_\mu + i[G_\mu, G_\nu]$ and $g$ is the QCD coupling constant. The fermionic quark fields $\psi$ have an extra 'colour' index in QCD, which runs from 1 to 3. Different 'flavours' of quarks are represented by independent fermionic fields, with different masses and

[1]Department of Physics, University of Wuppertal, Wuppertal, Germany. [2]Jülich Supercomputing Centre, Forschungszentrum Jülich, Jülich, Germany. [3]IJCLab, Université Paris-Saclay et CNRS/IN2P3, Orsay, France. [4]Department of Physics, Pennsylvania State University, University Park, PA, USA. [5]Institute for Computational and Data Sciences, Pennsylvania State University, University Park, PA, USA. [6]Institute for Theoretical Physics, Eötvös University, Budapest, Hungary. [7]University of California, San Diego, La Jolla, CA, USA. [8]Aix Marseille Université, Université de Toulon, CNRS, CPT, IPhU, Marseille, France. [9]Fakultät für Physik, Universität Regensburg, Regensburg, Germany. [10]LPNHE, Sorbonne Université, Université Paris Cité, CNRS/IN2P3, Paris, France. [11]Laboratoire de Physique de l'Ecole Normale Supérieure, Mines Paris - PSL, CNRS, Inria, PSL Research University, Paris, France. [12]School of Physics and Astronomy, University of Edinburgh, Edinburgh, UK. [13]RIKEN Center for Computational Science, Kobe, Japan. [14]Special Research Centre for the Subatomic Structure of Matter, Department of Physics, University of Adelaide, Adelaide, South Australia, Australia. ✉e-mail: fodor@bodri.elte.hu

charges. In QED, the gauge potential $A_\mu$ is a real-valued field, whereas in QCD, $G_\mu$ is a $3 \times 3$ traceless Hermitian matrix field acting in 'colour' space. In the present work, we include both QCD and QED as well as four nondegenerate quark flavours (up, down, strange and charm) in a fully dynamical, staggered-fermion formulation. We also consider the tiny contribution of the bottom quark. Its error is subdominant and we repeat the treatment of our earlier analysis[1].

To calculate the LO-HVP contribution to $a_\mu$, we start with the zero-three-momentum, two-point function of the quark electromagnetic current in Euclidean time $t$ (ref. 6). In this so-called time-momentum representation, it is given by

$$G(t) = -\frac{1}{3e^2} \sum_{\mu=1,2,3} \int d^3x \langle J_\mu(\vec{x}, t) J_\mu(0) \rangle, \qquad (1)$$

in which $J_\mu$ is the quark electromagnetic current with $J_\mu/e = \frac{2}{3}\bar{u}\gamma_\mu u - \frac{1}{3}\bar{d}\gamma_\mu d - \frac{1}{3}\bar{s}\gamma_\mu s + \frac{2}{3}\bar{c}\gamma_\mu c$. u, d, s and c are the up, down, strange and charm quark fields, respectively. The angle brackets stand for the QCD + QED expectation value to order $e^2$. It is convenient to decompose $G(t)$ into light (u and d), strange, charm and disconnected components, which have very different statistical and systematic uncertainties. Performing a weighted integral of the one-photon-irreducible part, $G_{1\gamma I}(t)$, of $G(t)$ from $t = 0$ to infinity yields the LO-HVP contribution to $a_\mu$ (ref. 6). The weight is a known kinematic function, $K(tm_\mu)$ (refs. 6–9). Thus:

$$a_\mu^{\text{LO-HVP}} = \alpha^2 \int_0^\infty dt K(tm_\mu) G_{1\gamma I}(t), \qquad (2)$$

in which $\alpha$ is the fine-structure constant at vanishing recoil and $m_\mu$ is the mass of the muon.

Reducing the uncertainty in the calculation of $a_\mu^{\text{LO-HVP}}$ to below half a percent is a notable challenge. In particular, several contributions to this uncertainty must be controlled. They are: (1) statistical uncertainties; (2) those associated with the finite spatial size $L$ and time $T$ of the lattice; (3) with the extrapolation to the continuum limit; (4) with fixing the five parameters of four-flavour QCD; (5) with isospin symmetry breaking. The progress made in our successive lattice calculations of $a_\mu^{\text{LO-HVP}}$ is illustrated in Fig. 1, in which those contributions to the uncertainty are shown. In the present work, we focus on reducing the two largest ones in our 2020 calculation, which are (3) and (2). We discuss all of these contributions ((1)–(5)) in detail now.

(1) Statistical uncertainties in the light-quark-connected and disconnected contributions to the correlation function of equation (1), associated with the stochastic evaluation of the QCD and QED path integrals, increase exponentially at large Euclidean times $t$. As well as the many improvements made in ref. 1, to reduce those uncertainties further, we use mock analyses to determine which ensembles required more statistics. In particular, we increase the statistics on the lattices that have the smallest lattice spacings and are critical for controlling the necessary continuum extrapolations. Moreover, to control the statistical uncertainties at large $t$, we replace the lattice calculation of the contribution to $a_\mu^{\text{LO-HVP}}$ from $G(t)$ above $t \geq 2.8$ fm by a state-of-the-art, data-driven determination, by means of the HVPTools set-up[10–13]. (Such a combination was originally proposed in ref. 14. There, however, lattice results were replaced by $e^+e^- \to$ hadrons data above a much earlier Euclidean time, $t \geq 1$ fm.) Here and in the rest of the paper, the expression 'data-driven' refers to predictions based on measurements of the hadron spectrum in $e^+e^-$ annihilation and $\tau$-decay experiments. Before combining the two results, we verify that the lattice and the data-driven determinations of part of this long-distance 'tail' contribution agree within errors. We compute this tail contribution using the most precise measurements of the two-pion spectrum by BaBar[15,16], KLOE[17–20] and CMD-3 (ref. 2), as well as the one obtained from hadronic $\tau$ decays[21,22]. These experiments almost fully cover the relevant energy range. For estimating the uncertainty of the tail

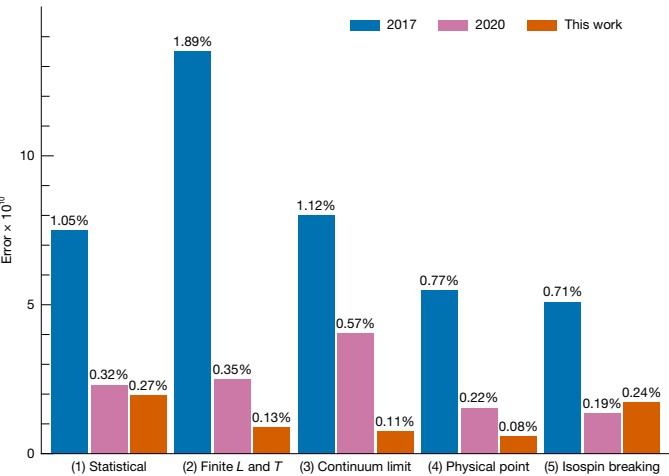

**Fig. 1 | Main uncertainties and their reduction in our successive lattice calculations of $a_\mu^{\text{LO-HVP}}$.** Their sources are labelled (1)–(5) in the text and are given a short descriptive title below the bars in the plot. Their approximate size relative to the total LO-HVP contribution obtained in the present work is also shown. The blue bars on the left of each group correspond to our 2017 result[24], the pink bars to our 2020 findings[1] and the orange bars to the work presented here. The isospin-breaking uncertainty (5) in this work is slightly larger than in 2020 owing to changes in the way we set the physical scale. We moved from using the $\Omega^-$ baryon to the pion decay rate, which reduced other uncertainties but increased the isospin-breaking uncertainty. Note: the statistical error (1) refers to that of the isospin-symmetric contribution in finite volume. The finite-size (2) and isospin-breaking (5) errors also contain statistical components of 0.08% and 0.16%, respectively.

observable, we also use other experiments with partial coverage. The two-pion spectra of these experiments are supplemented by the contributions from all of the other hadronic final states, as described in ref. 23. The tail contribution is dominated by centre-of-mass energies below the ρ-meson peak, a region in which all of the measurements agree very well. The tail only accounts for less than 5% of our final, lattice-dominated result for $a_\mu^{\text{LO-HVP}}$. The Supplementary Information describes our determination of this contribution and further justifies its use in our calculation.

(2) Finite $L$ and $T$ corrections gave the largest contribution to the error in 2017 (ref. 24). Even in our 2020 calculation[1], it was still a substantial source of uncertainty. Here our determination of the tail contribution using a data-driven approach reduces those corrections by a factor of about two and the associated uncertainties by roughly three. We compute those corrections using the dedicated simulations of ref. 1, supplemented by next-to-next-to-leading order chiral perturbation theory for distances beyond 11 fm (refs. 1,25). Those results are checked against nonperturbative analytical approaches to finite-volume corrections[26–30] that we complement with experimental $\pi^+\pi^-$ cross-section data below 1.3 GeV. Details are given in the Supplementary Information.

(3) The continuum extrapolation of the isovector contribution to $a_\mu^{\text{LO-HVP}}$ was the largest source of uncertainty in our 2020 computation[1] and we have dedicated substantial resources to further control it. The uncertainties were mainly because of long-distance, taste-breaking effects that are present in staggered-fermion computations. Here we add a new, finer lattice spacing. The corresponding simulations have a numerical cost close to that required for the full 2020 computation. In ref. 1 the smallest lattice spacing was 0.064 fm. The new lattice spacing is 0.048 fm. Because the leading discretization effects are proportional to the square of the lattice spacing, results at this new lattice spacing have cut-off effects reduced by a factor of nearly two. We further account for the fact that different $t$ regions in $G(t)$ have different cut-off effects by dividing the integral of equation (2) into four $t$

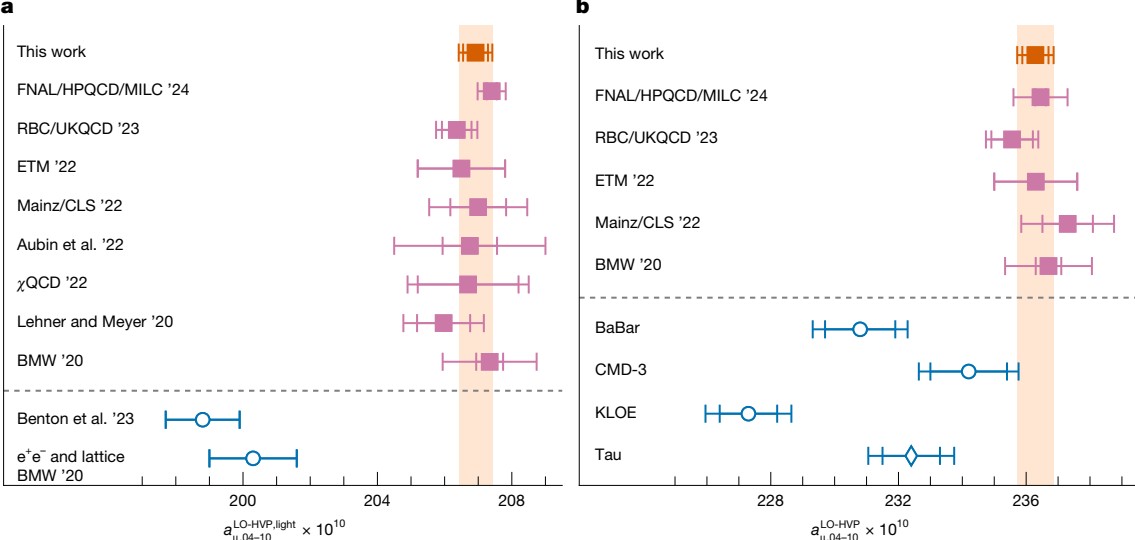

**Fig. 2 | Comparison of our intermediate-window results with others in the literature. a**, Light contribution to the ID window, $a_{\mu,04-10}^{\text{LO-HVP,light}}$. Our result is the orange square and the pink squares correspond to other lattice computations: Fermilab Lattice/HPQCD/MILC '24 (ref. 45), RBC/UKQCD '23 (ref. 38), ETM '22 (ref. 36), Mainz/CLS '22 (ref. 35), Aubin et al. '22 (ref. 34), $\chi$QCD '22 (ref. 33), Lehner and Meyer '20 (ref. 32) and our previous result BMW '20 (ref. 1). The blue

circles denote data-driven determinations of Benton et al. '23 (ref. 39) and BMW '20 (ref. 1). These two results are based on the KNT19 data compilation[40,41]. **b**, Full ID window, $a_{\mu,04-10}^{\text{LO-HVP}}$. Here, in the data-driven case, we show results[23] that use the measurements of the two-pion spectrum obtained in individual electron–positron annihilation experiments and in τ decays, as explained in ref. 23. The error bars correspond to the standard error of the mean.

intervals delimited by sigmoid functions. Such intervals or 'windows' were first proposed in ref. 14. The first window corresponds to the Euclidean-time interval 0.0 to 0.4 fm, known as the short-distance window[14,31] and denoted $a_{\mu,00-04}^{\text{LO-HVP}}$ here. We use three more intervals between 0.4 and 2.8 fm (separated at 2.0 and 2.4 fm) because this choice yields a reduced uncertainty on the final result for $a_{\mu}^{\text{LO-HVP}}$. We carry out the continuum extrapolation in those windows separately. We then add the individual extrapolated results to obtain the contribution to $a_{\mu}^{\text{LO-HVP}}$ from the Euclidean-time interval from 0 to 2.8 fm, taking correlations into account. The uncertainty on the light-connected contribution is decreased by the new ensembles by 37% and by using the data-driven approach to compute the tail by an extra 22%. The whole procedure is detailed in the Supplementary Information.

(4) We improve the determination of the physical point, which is now based on a very precise computation of the muonic decay rate of the charged pion. As a cross-check, we also perform the determination using the mass of the $\Omega^-$ baryon as input and find good agreement between the two approaches. The uncertainty associated with the physical point determination was already small in ref. 1 and is even smaller here. For details, see the Supplementary Information.

(5) The uncertainties on the isospin-symmetry-breaking contributions obtained in ref. 1 were already sufficiently small to reach the precision sought here. Our error on this contribution is now slightly increased: the isospin-breaking error on the pion decay rate is larger than it was on the $\Omega^-$ baryon. Also we perform a variety of cross-checks that confirm our earlier results on the isospin-breaking contributions. Our present uncertainty details are given in the Supplementary Information.

By far the largest contributions to the various windows considered in this work come from connected light-quark diagrams. We focus on these here and discuss the other contributions in the Supplementary Information.

For the connected contribution of the light u and d quarks to the intermediate-distance (ID) window, we find $a_{\mu,04-10}^{\text{LO-HVP,light}} = 206.92(37)(34)[50] \times 10^{-10}$, in which the first and second numbers in parentheses refer to the statistical and systematic uncertainties, respectively, and the number in square brackets is their quadrature sum, the

total uncertainty. As shown in Fig. 2a, our result agrees with eight other lattice calculations of this quantity[1,32–38], including our previous determination, within less than one standard deviation.

On the other hand, our new result for $a_{\mu,04-10}^{\text{LO-HVP,light}}$ differs from the data-driven one presented in ref. 1 by 4.3$\sigma$. This number was obtained by using the total result $a_{\mu,04-10}^{\text{LO-HVP}}$ from the data-driven approach and subtracting all but the light-connected contributions measured in our 2020 lattice simulations. There is another published result using the data-driven approach by Benton et al.[39]. These two results for the light-connected ID window are the only data-driven ones published. They are both based on the KNT data compilation[40,41] that does not include the more recent CMD-3 measurement nor the ones from τ decays. Their difference with our new result, as shown in Fig. 2a, reinforces the disagreement between the lattice and data-driven determinations found in ref. 1, which was a first strong indication that the lattice[1] and reference predictions for $a_{\mu}^{\text{LO-HVP}}$ (ref. 31) could not both be correct.

Note that the exact value of $a_{\mu,04-10}^{\text{LO-HVP,light}}$ depends on the scheme used to define the isospin-symmetric limit of QCD. Our scheme, originally defined in ref. 1, is specified in the Supplementary Information. In ref. 38, it is shown that the difference between the value of $a_{\mu,04-10}^{\text{LO-HVP,light}}$ obtained in the RBC/UKQCD scheme and in our scheme is approximately $0.10(24) \times 10^{-10}$, smaller than even our present uncertainties. The differences with other schemes used by the other collaborations are probably on the same level. However, we emphasize that this scheme dependence in no way affects our final result for $a_{\mu}^{\text{LO-HVP}}$ nor for the full value of $a_{\mu,04-10}^{\text{LO-HVP}}$ that includes all flavour, isospin-breaking contributions. Both are unambiguous physical quantities.

In Fig. 2b, we show a comparison of our result for the full ID window contribution, $a_{\mu,04-10}^{\text{LO-HVP}} = 236.29(41)(39)[57]$, with the five other lattice determinations of that quantity. Here the results do not depend on any scheme choice and agreement is still excellent. Also plotted are the individual data-driven results[23] obtained using the same datasets as for computing the central value of the tail. Those results show notable tensions that forbid an overall comparison between the lattice and data-driven approaches. However, important progress is being made on understanding the sources of those differences and we expect that the situation on the data-driven side will be clarified soon. The differences may be partly because of the treatment of radiative corrections,

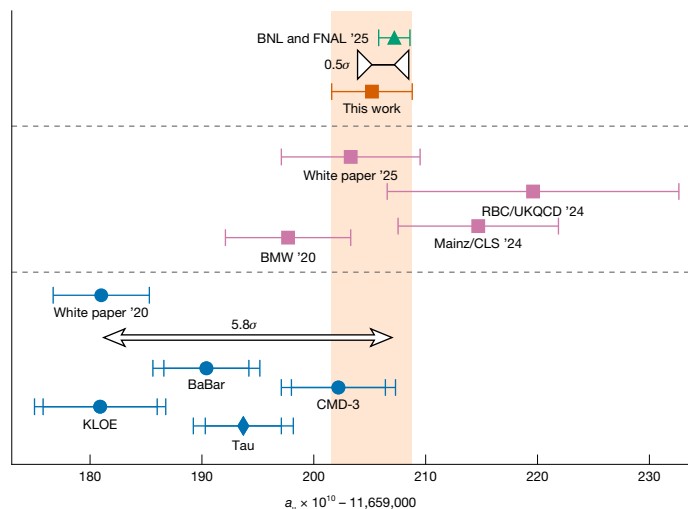

**Fig. 3 | Comparison of standard-model predictions for the muon anomalous magnetic moment with its measured value.** Top, world-average measurement of $a_\mu$ (ref. 4) and the standard-model prediction of this work. The latter is denoted by the orange band and is obtained by adding the value of $a_\mu^{\text{LO-HVP}}$ computed here to the results for all of the other contributions summarized in ref. 3. Middle, predictions using recent lattice computations for $a_\mu^{\text{LO-HVP}}$, RBC/UKQCD (refs. 14,38,51), Mainz/CLS[52] and our previous computation[1]. The Muon $g-2$ Theory Initiative combination from 2025 (ref. 3), which is obtained using lattice results for $a_\mu^{\text{LO-HVP}}$, is labelled 'White paper '25'. Bottom, predictions using the data-driven approach for $a_\mu^{\text{LO-HVP}}$ including the most precise measurements of the two-pion spectrum in electron–positron annihilation and τ-decay experiments[23]. These correspond to BaBar[15,16], KLOE[17–20] and CMD-3 (ref. 2) for $e^+e^-$ annihilation and Tau for τ decays[21,22]. The earlier Theory Initiative combination from 2020 (ref. 31), which is obtained using the data-driven results, is labelled 'White paper '20'. Note, all standard-model predictions include non-HVP contributions from 'White paper '25', except for 'White paper '20'. The error bars are the standard error of the mean.

as explained in refs. 23,42. Although the difference of our lattice result with that obtained using KLOE's measurement[17–20] is 6.2σ, it reduces to 3.5σ for the BaBar measurement[15,16] and even to 1.3σ for the one by CMD-3 (ref. 2). Compared with the determination obtained through τ decays[21,22], the difference is 2.7σ. With an alternative evaluation of the τ data[43], the difference is even smaller. These numbers illustrate the known discrepancies between measurements at energies around the ρ-meson peak. Note that these contributions are highly suppressed in the tail observable. Nevertheless, we take into account these discrepancies by performing the analysis of the tail with and without the most extreme experiments. The associated uncertainty is an order of magnitude below our final error on $a_\mu^{\text{LO-HVP}}$. Details can be found in the Supplementary Information.

Our result for the light-connected contribution to the short-distance window, $a_{\mu,00-04}^{\text{LO-HVP,light}} = 47.85(5)(13)[14] \times 10^{-10}$, is in excellent agreement with five other lattice computations of this quantity[36,38,44–46]. We also consider the window observable proposed in ref. 34, from 1.5 to 1.9 fm, and we obtain $a_{\mu,15-19}^{\text{LO-HVP,light}} = 97.57(1.76)(1.17)[2.11] \times 10^{-10}$. Again we find a good agreement with the other two computations of this quantity[34,37]. A more detailed comparison of our results for the above windows is provided in the Supplementary information.

Now, summing the connected-light and disconnected contributions obtained in our four chosen Euclidean-time intervals and combining them with all of the other required contributions, including the data-driven tail, we obtain $a_\mu^{\text{LO-HVP}} = 715.1(2.5)(2.3)[3.4] \times 10^{-10}$, as detailed in the Supplementary Information. This result agrees with our earlier 2017 and 2020 determinations but reduces uncertainties by a factor of 5.5 compared with the former and of 1.6 to the latter. The difference between our result and the 2020 result is $7.6 \times 10^{-10}$, with an uncertainty

of $5.2 \times 10^{-10}$, indicating that the new result is 1.5σ higher. To obtain that result, we assume zero correlation among some of the systematics. When assuming full correlation, the uncertainty becomes $4.5 \times 10^{-10}$ and, in this case, the new result is 1.7σ higher.

Adding our determination of $a_\mu^{\text{LO-HVP}}$ to the other standard-model contributions compiled in ref. 3 yields $a_\mu = 11,659,205.2(3.6) \times 10^{-10}$. In Fig. 3, we compare this result with the world average of the direct measurements of the magnetic moment of the muon[4]. Our prediction differs from that measurement by −0.5σ. Also given are the Muon $g-2$ Theory Initiative combinations from the years 2020 (ref. 31) and 2025 (ref. 3), in which the $a_\mu^{\text{LO-HVP}}$ contribution was obtained only from the data-driven and only from the lattice approach, respectively. As well as these combinations, we also provide individual results in both approaches. As the figure shows, some of the data-driven results are in serious tension both with our and the lattice-only estimates. Our $a_\mu^{\text{LO-HVP}}$ is in good agreement with the latest Theory Initiative combination and our uncertainty is a factor of 1.8 smaller.

In the near future, we expect more data for the $e^+e^- \to \pi^+\pi^-$ cross-section[47]. Beyond consolidating our present understanding[23,42] of the tensions in the measurements of that cross-section, these new data should improve the data-driven determination of $a_\mu^{\text{LO-HVP}}$. Also, the possibility of directly measuring HVP in the space-like region is being investigated by the MUonE collaboration[48]. Finally, combinations of lattice and data-driven results, beyond the simple one presented here, ought to be pursued, following, for example, the methods put forward in ref. 49. Investigations along all of those lines are underway.

The precise measurement and standard-model prediction for the muon anomalous magnetic moment reflect substantial scientific progress. Experimentally, Fermilab's 'Muon $g-2$' collaboration has measured $a_\mu$ to 0.127 ppm (ref. 4). Furthermore, there is the 'Muon $g-2$/EDM' experiment under development at KEK's J-PARC[50] to measure this quantity using a completely new and independent experimental approach. On the theoretical side, physicists from around the world have performed complex calculations (see, for example, ref. 3), some based on further precise measurements, incorporating all aspects of the standard model and many quantum field theory refinements. It is notable that the electromagnetic, electroweak and strong interactions, which require very different computational tools, can be combined into a single calculation with such precision. The result for $a_\mu^{\text{LO-HVP}}$ presented here, combined with other contributions to $a_\mu$ summarized in ref. 3, provides a standard-model prediction with a precision of 0.31 ppm. At such a level of precision, the agreement found between experiment and theory, to within less than one standard deviation, is a great success for the standard model and, from a broader perspective, for renormalized quantum field theory.

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

## Data availability

The datasets for the continuum extrapolation tables are publicly available from https://doi.org/10.5281/zenodo.17880027 (ref. 53). Those for the other figures and tables are available from the corresponding author on request.

## Code availability

A CPU code, which was used for configuration production and measurements, can be obtained from the corresponding author on request, subject to possible export control constraints. The Wilson flow evolution code, which was used to determine $w_0$, can be downloaded from arxiv.org/abs/1203.4469.

53.   BMW Collaboration. Supplementary data-files for continuum extrapolation tables in arxiv:2407.10913. *Zenodo* https://doi.org/10.5281/zenodo.17880027 (2026).

**Acknowledgements** We warmly thank A. El-Khadra, M. Hansen, E. Jenkins, L. Jin, Ch. Lehner, A. Manohar, H. Meyer, D. Nogradi, A. Patella, A. Ramos, R. Sommer, P. Stoffer and H. Wittig for enlightening discussions and K. L. Kelley and I. Frankel for insight on the scale-setting observables. We are grateful to A. Keshavarzi, D. Nomura and T. Teubner for sharing their KNT19 compilations of $e^+e^- \rightarrow$ hadrons cross-sections and for correspondence. We gratefully acknowledge the Gauss Centre for Supercomputing (GCS) e.V. (www.gauss-centre.eu), GENCI (www.genci.fr, grant 502275), EuroHPC Joint Undertaking (grants EXT-2023E02-063 and EXT-2024E02-109) and the Australian National Computational Merit Allocation Scheme for providing computer time on the GCS supercomputers SuperMUC-NG at Leibniz Supercomputing Centre in München, Hawk and Hunter at the High Performance Computing Center in Stuttgart and JUWELS, JURECA and JUPITER at Forschungszentrum Jülich, as well as on the GENCI supercomputers Joliot-Curie/Irène Rome at TGCC, Jean Zay V100 at IDRIS, Adastra at CINES, on the EuroHPC JU flagship supercomputers Leonardo at CINECA and LUMI at CSC, Gadi at NCI and Setonix at PSC. This work received funding from the French National Research Agency under contract ANR-22-CE31-0011, from the Excellence Initiative of Aix-Marseille University – A*Midex, a French 'Investissements d'Avenir' programme under grants AMX-18-ACE-005 and AMX-22-RE-AB-052 and from grants NW21-024-A and BMBF-05P21PXFCA. A.B., Z.F., D.G. and A.Y.K. are supported by ERC-MUON-101054515. Z.F. is also partially supported by NW21-024-B and DOE-0000278885. A.P. is partly supported by UK STFC Grant ST/X000494/1 and by long-term Invitational Fellowship L23530 from the Japan Society for the Promotion of Science. F.S. is supported by a Ramsay Fellowship from the University of Adelaide.

**Author contributions** Code development: S.B., K.K.S., B.C.T., G.W. Runs and data management: K.K.S., G.W. Autocorrelations, cross-checks: F.F., S.M., A.R. Meson masses: F.F., F.S., K.K.S., G.W. Omega masses: F.F., F.S., G.W. Pion decay rate: A.C., F.F., D.G., L.L., A.P., F.S., K.K.S., B.C.T. Analysis strategy: Z.F., L.L., S.M., F.S., K.K.S., B.C.T. Short-distance window: A.Y.K., L.L., S.M. Intermediate-distance and long-distance observables: A.Y.K., F.S., K.K.S., B.C.T. Strange/charm contributions: A.Y.K., B.C.T. Finite-size effects from lattice: K.K.S., B.C.T. Finite-size effects from data: A.B., D.G., L.L., A.L., M.S. Isospin-breaking contributions: A.G., L.L., A.P., F.S., K.K.S., B.C.T. Data-driven approach: M.D., Z.F., L.L., B.M., Z.Z. Acquisition of computer resources: S.B., Z.F., D.G., L.L., T.L., F.S., K.K.S., B.C.T. Main paper text: Z.F., L.L., F.S. Coordination: Z.F., L.L., F.S., K.K.S.

**Funding** Open access funding provided by Bergische Universität Wuppertal.

**Competing interests** The authors declare no competing interests.

**Additional information**
**Correspondence and requests for materials** should be addressed to Z. Fodor.
