## [Peer Review File · Nature]

Hybrid calculation of hadronic vacuum polarization in muon $g-2$ to 0.48%

Corresponding Author: Professor Zoltan Fodor

Version 0:

Reviewer comments:

Referee #1

(Remarks to the Author)

Key results. This manuscript discusses a calculation of the leading-order hadronic vacuum polarization (LOHVP) contribution that forms part of the theoretical calculation, in the Standard Model (SM), of the anomalous magnetic moment of the muon, a_μ . The motivation for this calculation is that the Muon $g-2$ experiment will announce their final result for a_μ next year and the current theoretical situation is somewhat unclear because of issues to do with the calculation of the LOHVP. This matters because different theory results for the LOHVP lead either to the conclusion that there is a signal for new physics visible as a difference between experiment and the SM result or there is not. Which is the case is obviously of critical importance to the field of particle physics, and more widely.

The calculation here improves on a previous one from 2020 by the BMW collaboration (published in Nature in 2021), reducing the uncertainty by a factor of 1.7. The current calculation then quotes an uncertainty for the LOHVP of 0.5% which is the benchmark uncertainty being aimed at by theory calculations. The result gives an a_μ value in agreement with the current experimental result at the level of 1 sigma. This is in contrast to an earlier result for the LOHVP from the 2020 theory white paper which is based on experimental data for the cross-section for $e+e-$ to hadrons and which gives a value for a_μ which disagrees with the current experimental result. The theory white paper result has been called into question by more recent experimental data for $e+e-$ to $\pi+\pi-$ from the CMD3 collaboration. The result obtained here for the LOHVP agrees with that obtained from experimental cross-section data if the CMD3 results are used for the $e+e-$ to $\pi+\pi-$ contribution, as shown in Figure 3.

I think this does represent a significant advance in understanding and it will influence the field (indeed it is being widely discussed).

Validity. This calculation uses lattice QCD for ~95% of the LOHVP contribution and a data-driven result (i.e. using $e+e-$ to hadrons data) for the remaining contribution in a region (at large Euclidean time above 2.8 fm) where lattice QCD uncertainties grow. It is the use of this technique, plus the determination of results at a finer lattice spacing than BMW used in 2020 that enables the reduction in uncertainty. The size of the uncertainty is important to the conclusions they are able to draw from their results and the impact they have.

The uncertainties of various pieces of the LOHVP are discussed in detail in the Supplementary Information. Plots are given, for example, for the 'time-window' between 0.4 fm and 1.0 fm (in Table 6) that show the lattice results at different values of the lattice spacing and the extrapolation to the zero lattice spacing continuum limit. This allows the reader to assess visually the validity of the result obtained in the continuum limit. An error budget is also given for the final value for this time-window.

Section 5.4 discusses their main result - the lattice QCD calculation of the light quark piece of the LOHVP in the 0.0-2.8 fm time-window. This section is notable for its lack of plots, except for the isospin zero contribution (which contains the disconnected piece). The value obtained for the largest contribution to the whole calculation i.e. the light quark connected contribution is not evidenced by any plots or error budgets. They need to be included to allow the reader to judge the

statements made.

The authors split up the 0.0-2.8 fm region into four windows. The results for the first window (0.0-0.4 fm) are displayed in Table 4, having been discussed in section 5.2. The other windows are not shown. The authors stress in section 5.4 that splitting the region up into several windows is beneficial because it allows different forms of continuum extrapolation to be used in each window, reducing the overall uncertainty. It is important for the reader to see how this works in practice through plots of the results of the different windows and the fit functions used. It is also important to have error budgets for each window to see how the total uncertainty is pieced together.

The lack of this information is a flaw in the manuscript that means it should not be published in its present form.

A further issue concerns the calculation of the contribution from the region above 2.8 fm. This is done using experimental data for the cross-section to $e+e-$ to hadrons. The authors note that at such large Euclidean times the contribution is dominated by the data at very low values of centre-of-mass energy. They claim that, at such low values, there is no significant disagreement in the experimental data despite the tensions seen in $e+e-$ to $\pi+\pi-$ between CMD3 and earlier experiments. Using results from BaBar, KLOE, CMD3 and tau data they quote a result for this small but not insignificant contribution with a 1% uncertainty. The statement about the lack of tension in the experimental data seems surprising given some of the other plots shown here and also other results in the literature. For example, arXiv:2311.09523 (Figure 3) shows a 4% difference between a data-driven result using the KNT19 compilation of experimental data and that substituting CMD3 numbers for $e+e-$ to $\pi+\pi-$ for a large-time window between 1.5 fm and 1.9 fm. Arxiv:2410.23832 Table 2 also shows a roughly 4% difference for these two cases in the 2.8fm to infinity window. I think a more complete analysis of the experimental situation is needed here to be sure that the 1% uncertainty on this piece is justified.

Originality and significance. The originality here lies in the new lattice QCD results (beyond those used in BMW's 2020 paper) and the novel technique for combining lattice QCD values in a finite time-window with data-driven results at large Euclidean times (this was suggested before, as the authors make clear in footnote 1, but not implemented). The level of accuracy achieved is a key point.

The results are of interest, both within the field and more broadly.

Data and methodology. In general the validity of the approach, quality of data and quality of presentation are fine. I discuss above the two major shortcomings in my view that need to be addressed in the supplementary information to make the reporting sufficiently detailed to convince the reader of the conclusions.

Appropriate use of statistics and treatment of uncertainties. In general this is fine. As discussed above more information is needed in the supplementary materials for the reader to assess the validity of the uncertainties for the overall result.

Conclusions. The robustness, validity and reliability of the conclusions need to be improved by addressing the points under 'validity' above.

Suggested improvements. Improvements that are needed are given in the section 'validity' above. Further suggestions that would clarify points and some typographical errors found are listed below.

- 1) The term 'Strong-isospin breaking' is used in the main text (e.g. line 85) with no description of what it is.
- 2) Line 69 - there is in fact no discussion of the top quark contribution, so perhaps drop that.
- 3) Line 180-181 1.5 fm TO 1.9 fm ?
- 4) Line 189 perhaps write $6.5(5.5) \times 10^{-10}$ since the 10^{-10} was reinstated in the few lines above.
- 5) caption to Figure 3 - computed in present -> computed in THE present
- 6) caption to Table 3 horizontal -> vertical
- 7) some discussion of the improvement in the uncertainty on w_0 from the 2020 value would be useful. This comes largely from improvements in the systematic uncertainties. How big a role does the new fine ensemble play here? (This would be helpful in setting the bigger picture of improvements from the new finer ensemble).
- 8) the comment on discussion of improvement over the 2020 result also applies to the intermediate distance window.
- 9) Table 8 - adding in the result for the 'old' data-driven value for the 1.5 to 1.9 fm window from 2311.09523 to the comparison plot might be a good idea to provide the reader with the 'bigger picture'(since this was done for the intermediate distance window).
- 10) section 5.3 - references for the taste-improvement procedure should be given. Referring to the 2020 paper for references is irritating for the reader.
- 11) Beginning of section 5.4 - the time separations of the different windows look a bit arbitrary. It would be good to have a bit more discussion of how these were arrived at. A more natural break would seem to be 1.0fm since that is used for the intermediate window. Instead we have 0.6fm and 1.2fm. Is the 1.2fm break used because of statements in the section before about the time value at which taste-improvement is introduced?
- 12) Figure 11 needs units (fm) for t^*
- 13) In the discussion of the comparisons for the 2.8-3.5 fm window (around Figure 14) it would be useful to know how much of the 2.8 to infinity contribution this represents.

References. These are fine. One suggestion for an improvement is given above as number 9.

Clarity and context. The main text is generally clear and accessible. One improvement is suggested under item 1 above.

(Remarks on code availability)

Referee #2

(Remarks to the Author)

A more readable LaTeX → PDF version is in an attachment.

This paper presents a precise result for $a_\mu^{\text{LO-HVP}}$, the leading order contribution to the muon's anomalous magnetic moment from hadronic vacuum polarisation (HVP). Despite what the title, abstract and various passages say, it is not a "calculation" or "computation" as these terms are usually understood in lattice quantum chromodynamics (LQCD). Instead, it is a *hybrid determination* of $a_\mu^{\text{LO-HVP}}$, consisting of 95% LQCD and 5% information from two other sources. One source is three collider experiments that measure the cross section for $e^+e^- \rightarrow \pi^+\pi^-$ (and some other hadronic final states), normalised to the cross section for $e^+e^- \rightarrow \mu^+\mu^-$; the other source is τ -lepton decay to hadronic final states (and a τ neutrino), analogously normalised and corrected for isospin effects.

A version of the paper was posted on the hep-lat section of the eprint arXiv in July [arXiv:2407.10913]. The analysis was presented in a seminar by one of the senior authors at CERN, in several talks at Lattice 2024 in Liverpool, and at a plenary workshop of the Muon g-2 Theory Initiative (TI). I participated in a subset of these events and have interacted with participants of all three. Several of my remarks capture widely held views. Of course, popular opinions need not be right, but perhaps it is useful for editors and authors to know whether a criticism arises from an individual referee versus many readers and audience members.

Following common practice since ref. [15/34], an integral over time (in practice, a sum over discrete LQCD data) is split into regions or "windows" with smooth edges. (Reference numbers refer to the main paper and the supplementary information, respectively; below [n/] ([/n]) is used for papers cited only in the main text (supplement).) The TI recommends that LQCD collaborations publish results for the windows $t/\text{fm} \in [0,0.4)$, $(0.4,1.0)$, and $(1.0,\infty)$. These three windows sum to the total $a_\mu^{\text{LO-HVP}}$. (In each window, and in the total, $a_\mu^{\text{LO-HVP}}$ can also be split by the quarks in the HVP: light, strange, charm, etc., which makes sense for "connected" diagrams, and in addition there is a "disconnected" contribution that is small and usually summed over all pairs of flavours in the two disconnected parts of the diagram.) Roughly speaking, they are around 50, 200, and 450 (in units of 10^{-10}), adding up to about 700.

It is important to follow the TI's guidelines because the current, overall status of $a_\mu^{\text{LO-HVP}}$ is very puzzling. In 2020, the TI put out a white paper [2/71] that recommended that particle physicists use a Standard-Model prediction based on experimental data for $e^+e^- \rightarrow \text{hadrons}$, which, when added to non-hadronic contributions, disagrees with BNL and FNAL measurements of a_μ by a large amount. The "disagreement" nourishes a community of beyond-the-Standard-Model speculators who are wasting their time if the discrepancy is not robust. Even at the time of the white paper, the not-yet-published ref. [3/7] cautioned against such speculation, as it was compatible with the combination of BNL and FNAL measurements. Since 2020, the $e^+e^- \rightarrow \pi^+\pi^-$ measurements of CMD-3 [4/61] have undermined the $e^+e^- \rightarrow \text{hadrons}$ determination of $a_\mu^{\text{LO-HVP}}$, because they disagree with BaBar, KLOE, BESIII, SND and even CMD-2. The experimenters involved in these experiments have ahead of them an enormous task to figure out where things have gone wrong. Meanwhile, to validate the value of $a_\mu^{\text{LO-HVP}}$ in ref. [3/7], LQCD experts will have to work together to make a persuasive case. It is thus in the interest of *Nature* (as the publisher of ref. [3/7]) and the authors common to ref. [3/7] and #2024-10-23178 to foster a robust average of LQCD results.

This paper presents detailed results for the connected (by flavour light, strange, charm) and disconnected diagrams for $t/\text{fm} \in [0,0.4)$ and $(0.4,1.0)$ but not for $t/\text{fm} \in (1.0,\infty)$. Instead, it splits $(0.4,\infty)$ into $(0.4,0.6)$, $(0.6,1.2)$, $(1.2,2.8)$ and $(2.8,\infty)$. The first three of these are analysed with a common χ^2 minimisation but with different choices for the functional forms of the extrapolation to the continuum limit. The reason given (which I and others accept) is that the statistical power of LQCD depends on the t variable, so shorter t needs more terms in the continuum extrapolation. The last time window, $(2.8,\infty)$, is determined not from LQCD but from the experimental data mentioned above. That is why one should call the work under review a hybrid determination and not a calculation.

There is no serious objection to the split into $(0.4,0.6)$, $(0.6,1.2)$, $(1.2,2.8)$ and $(2.8,\infty)$. Although more details could be provided justifying the edge times 0.6 and 1.2 fm, presumably they work well for the data set. That said, the lack of a result for $(1.0,\infty)$ detracts from the value of the current version of the manuscript. I've not encountered anyone who excuses this omission, because as noted above it is the window with more than half the total.

It is worth noting that there are several examples in the LQCD literature in which an LQCD result is presented and then, for those willing to make additional assumptions, a more precise hybrid from incorporating (compatible) experimental measurements is presented as a subsidiary result. (If the LQCD authors left out the hybrid combination, someone else would do it and get a higher ratio of credit/effort.) Everyone (apart from the authors?) wants an eventual published version of #2024-10-23178 to provide LQCD-only results for $(1.0,\infty)$, $(2.8,\infty)$ and the full $a_\mu^{\text{LO-HVP}}$. The LQCD data to obtain such results are clearly in hand, and the upper and lower bounds used in ref. [3/7] could be used to obtain them. A paper that wanted nothing but to move the science of the muon's anomalous magnetic moment forward would include this information. One can only speculate why this information is omitted (and I do so below).

Below this report lists several ways to improve the presentation of the LQCD part of the hybrid determination. First, however, let us consider the e^+e^- -scattering and τ -decay data used for $t/\text{fm} \in (2.8, \infty)$. For many years, these two approaches yielded not quite compatible results for $a_\mu^{\text{LO-HVP}}$. Eventually, the τ -decay method was discredited for various reasons, and #2024-10-23178 makes no effort to reassess/rehabilitate it. As noted above, the e^+e^- method seemed consistent for many years until the early 2023 measurement of $e^+e^- \rightarrow \pi^+\pi^-$ by CMD-3 [4/61]. Discussions within the TI suggest that understanding the discrepancies among experiments is still a work in progress, but a tentative conclusion is that in the application of radiative corrections either uncertainties were underestimated or mistakes (!) were made. For most of the community, using e^+e^- or τ -decay data is too risky (at least for the time being).

It is argued that for the window $(2.8, \infty)$ all e^+e^- experiments agree, as do the τ . So, we have a region in which a set of experiments (e^+e^-) in crisis agree with a set of measurements (τ) for which the evaluation of $a_\mu^{\text{LO-HVP}}$ has been deprecated. The authors are free to crawl out on this limb, but *Nature* has to decide whether it wants to take such a risk. I wouldn't do it, especially when both full LQCD and hybrid results could easily be presented.

I suspect (and as do several others) that the reason that a full LQCD result is omitted is that it would entail an incremental improvement over ref. [3/7]. It would not be impressive enough for *Nature*, at least so the authors fear. Who do you (editors and authors) want to impress? The experts, or some set of people whose interest in muon g-2 is shallow? What happens if the next TI white paper recommends putting hybrid results aside for now? Many of us see only a downside (for the authors and their journal, as well as the community) to omitting the full LQCD result.

For a revised version, whether resubmitted to *Nature* or submitted to, say, the *European Physics Journal*, the following aspects of #2024-10-23178 should be changed (working sequentially through the main text and supplement):

- If, despite the criticism given above, the hybrid result remains the central (or, heaven forbid, only) result, change the title to "Hybrid determination of the \dots ".
- Correct mild errors in the Abstract:
 - the "contradiction" of a_μ is not "recent" but has been with us for around a quarter century.
 - CMD-3 did not determine $a_\mu^{\text{LO-HVP}}$ in ref. [4/61] but merely indicated how things might change substituting their data for those of earlier experiments.
- 1st paragraph main text: QFT leads to additional small corrections to a_μ from particles and interactions whether they are included in the Standard Model or not. That's why the topic is so interesting!
- "QCD is a generalised version of QED": this phrasing appears in other papers by some of the present authors. The statement is both true and misleading, because the generalisation is a mathematical statement, while the difference in physical phenomena between QCD and QED is drastic (and why LQCD is often mandatory but standalone LQED never is). Thus, it is doubtful that the statement aids a general *Nature* reader. (In this vein, seven out of 14 authors of ref. [3/7] are among the 21 authors of #2024-10-23178. I think this clause is not the only (near) verbatim repeat, which some would say means the 14 new 2024 authors are plagiarising the seven departed 2020 authors. There are also many phrasings of the sort "our earlier work": is that appropriate with the overlap at hand? Although both works will often be attributed to the "BMW" collaboration, neither paper's masthead attributes the work to any collaboration at all. I don't have an opinion on this matter but raise the issue in case the Editors have thought about it and would see a need to apply policy for such circumstances.)
- Fig. 1 axis label: "Finite $L \ \& \ T$ " should be "Finite $L \ \& \ T$ " to conform with the notation in the text.
- *ad* (a): "we replaced" should be phrased in the present tense, because for the reader it is happening right now in this manuscript, not some time in the past. Several other places have inconsistent usage of verb tense; please review, as I cannot track all of them.
- Many places, including in *ad* (a): "data-driven" without referring to e^+e^- (or τ) will confuse readers not steeped in HVP jargon, because the LQCD analysis is also driven by (computer-generated) data.
- *ad* (b): NNLO is not defined, though spelling it out might not help many readers.
- *ad* (c): "The first of our windows": it's RBC's window unless one is of the view that it now belongs to everybody.
- Paragraph starting "For the connected": "ID" is not defined. The notation for errors should be defined here (and possibly repeated in the supplement). Perhaps the first number in parentheses is the statistical uncertainty, the second number in () is the systematic uncertainty, and the number in brackets [] is the quadrature sum.
- Fig. 2 and surrounding text: for the plots it is necessary to have short labels, but the text and/or caption should make clear that "RBC '23" is an abbreviation for RBC-UKQCD [32/36], "Lehner '20" for Lehner and Meyer [33/40], "FHM" for Fermilab Lattice, HPQCD, and MILC, "Benton" for Benton *et al.*, etc.
- Fig. 3: This figure is fine---don't change a thing. That said, I alert the Editors that the lower half shows what a mess the "data-driven" scenario has become, which undermines using it (as discussed above) without stronger arguments than those given in the main text or in Section 9 of the supplement.
- The initials in the acknowledgments and author contributions should match the author list, so KKS and BCT for K.K. Szabo and B.C. Toth.
- Fig. 1 (supplement): the meaning of the error bar on the physical point is not defined. The numbers in () for the β values are inscrutable.
- FLAG is cited for m_c/m_s but not the underlying work, which is a violation of FLAG policy. Please cite all papers entering the average. (Some authors are FLAG members and should know better.)
- It is not clear whether the numbers for m_s/m_l are exact or are rounded from am_s divided by an exact am_l .
- Bottom of p. 6: "It is usually assumed that these [hairpin] artefacts decrease \dots " but what a reader needs to know is not (just) the usual assumption but the assumption made in the work at hand.

• Section 2, Omega baryon computations:

— sloppiness in plural/singular: "In the continuum limit the masses of these states become degenerate." Also, there may be only one smearing parameter σ .

— more seriously, too little information on the continuum limit and strange retuning is provided. A revision should state how any readjustments of the strange quark mass propagate to uncertainty in M_Ω . In ref. [3/7], the M_Ω continuum extrapolation was slightly controversial---in response to community comments, it changed from arXiv-v1 to arXiv-v2. A revision must include a plot of the current status, not least to see how the new finer ensemble improves the extrapolation. Many people share this reservation about the current presentation.

— perhaps the requested information is actually already in Table 3 of Section 3? If so, the plots in Table 3 would convey this information by presenting the extrapolation as $w_0 M_\Omega$ instead of w_0 in attometres.

• The beginning of Section 3 muddles a bit the distinction between the simulation meson masses and their physical values.

• The definition of the lattice spacing is not made clear. If I've understood, on a given ensemble $a=[a/w_0]_{GF}[w_0]_{\text{phys}}$ with $[w_0]_{\text{phys}}$ from eq. (15). Further, I think in eq. (15) $[w_0]_{\text{phys}} = [M_\Omega^{-1}]_{\text{PDG}} \lim_{a \rightarrow 0} ([aM_\Omega]_{\text{cor.fit}}[a/w_0]_{GF})$ but it is never stated this way. (GF stands for gradient flow (Wilson had nothing to do with it.), and "cor.fit" for correlator fits from Section 2.)

• Caption for PDF plot in Table 3: like in similar tables, the median is presumably given by the blue *vertical* line. The caption states "horizontal".

• Captions for PDF plots in Tables 3, 4 state the 1-sigma/2-sigma band is formed at the 16th/2th and 84th/98th percentiles while the plot shows 68.4%/95.4%. It takes a while to understand how they are compatible, possibly because using % and the word "percentile" is at best redundant and probably actually incorrect. It would be better for the caption to use words closer to the content of the plot, perhaps

"the 1-sigma and 2-sigma bands contain 68.3% and 95.4% of the distribution, centred at the median."

• The work of ref. [30] raises the possibility that the anomalous dimension implies a power of $|\alpha_s^n|$ with $n < 0$ and worries that this behaviour might not be visible at current lattice spacings. The analysis should incorporate fits with $n < 0$, for example $n = -3$ as studied in some LQCD papers [37/39].

• Ref. [31] and follow-on works by Neil and Sitison advocate a Bayesian Akaike information criterion (BAIC) with a more severe penalty for fits that include fewer data. The submission could be improved by providing a quick reminder why, in the statistical framework adopted here, the AIC of eq. (19) (attributed to ref. [37]) is more appropriate than the BAIC. If the reasons are not strong, it would be good to state whether any significant change arises between the two criteria. Also, consider citing Akaike's work.

• Tables 3, 4, 5, 6, 7 and 8: more information (in the text) describing the corpus of fits is necessary. How are the p values distributed? Which fits---for example which functional forms for the continuum extrapolation---are most highly weighted? When the PDF has structure, as almost all do, what features predominate fits in the various humps? (This report is certainly not the first time these questions have been raised.) How large are the dimensionful coefficients of the discretisation effects?

• Can the improvement in continuum extrapolations from the new $a=0.048$ fm ensemble be quantified? Was it in retrospect a good investment of computer time? Would it have been better to take some of the time and devote it to higher statistics on coarser lattices? In particular to address the $1/\text{fm} \in (1.0, \infty)$ window in LQCD without resorting to troublesome experimental data?

• Table 6 does not include a line for taste-breaking effects, and the text is very terse. Given the strategy used elsewhere in this paper, shouldn't fits with and without taste-breaking corrections be included, with the AIC weighting them accordingly?

• Tables 7 and 8 refer the reader to Table 6 for the definition of symbols, etc., but Table 6 sends the reader to Table 4. An unnecessary annoyance.

• The "SRHO model [used] to get the central value" of the Aubin *et al.* window, $1/\text{fm} \in (0.15, 0.19)$. Given the general strategy of this paper, shouldn't fits with and without taste-breaking corrections be included, with the AIC weighting them accordingly?

• Section 6: the notations 4hex , $N^3\text{LO}$, $N^4\text{LO}$ and $N4\text{LO}$ are not defined. The last two presumably mean the same thing. 4hex could be elegantly introduced after the words "utilized a new staggered action", I suppose. Sometimes the superscripts are $l=1$, $l=0$ instead of using the customary maths-italic l .

• A strength of this work is the estimate of finite-size effects with dedicated simulations. Even so, it would be worth calling out in the text that NNLO for $L_{\text{ref}} \rightarrow L_{\text{big}}$ is consistent with the 4hex calculation. As it stands, the reader has to mine Tables 13 and 14 for this piece of information.

• Section 7 is written for deeply knowledgeable experts. It is very terse. Perhaps that is acceptable, but the papers that developed low-mode averaging should be cited.

• Section 8 is an alternative estimate of finite-volume (aka finite-size) effects. It would be better to integrate the discussion with Section 6---at least don't have an intervening section. A table summarising the $L_{\text{ref}} \rightarrow \infty$ corrections for NNLO, $4\text{hex}+\text{NNLO}$, data-driven, and final (i.e., cyan band of fig. 12) would be helpful: information now in Section 6 could perhaps be added to Table 17. By the way, Table 17 is not explained in the text, although it is used in Table 11.

• In Section 8, it is hard to follow the connection between the words and the width of the cyan bands in fig. 12. Perhaps paragraph explaining Table 17 and, hence, the cyan bands was dropped in editing.

• Using e^+e^- data for finite-volume corrections is subject to the same scepticism as for $1/\text{fm} \in (2.8, \infty)$. As in Section 9, not all available experiments are used.

• Both bibliographies need some hand-editing w.r.t. bibtext code from `\textsc{inSPIRE}`. For example, more recent papers' authors' names include diacritics (e.g., Cè) while from older papers they are left out (e.g., Luscher instead of Lüscher). Some math/symbol material is spelled out in ugly notation (compare $\pi^* \pi^*(\gamma)$ in refs. [59, /60]).

Apologies for interspersing serious comments with minor issues that should (especially with so many authors) have been caught during internal review.

As discussed above, Section 9 is key to a decision whether to publish #2024-10-23178 or not. The "data-driven" estimate of $1/\text{fm} \in (2.8, \infty)$ reduces the uncertainty in $a_\mu^{\text{LO-HVP}}$ hybrid determination compared with (my guess of the not provided) LQCD-only calculation. The text actually summarises well why one would *not* trust the e^+e^- (or τ) data. The three-bullet justification has the following weaknesses:

• *four experiments agree*: agreement is no guarantee of reliability;

• *compatibility with LQCD*: compatibility for the full tail window is not shown---fig. 14 shows only

$1/\text{fm} \in (2.8, 3.5)$; fig. 14 only validates this window to the 8% level;

• *impact on total uncertainty is negligible*: which means it disappears in a quadrature sum; yet if the uncertainty is merely at the 10% level (based on compatibility with LQCD), then that is not true; moreover, although it is stated "all" e^+e^- experiments agree, only three are actually included. I estimate that including them would reduce the final result by 1 or 2 units (10^{-10}), which is a bit smaller than the final uncertainty estimate, but not at all small compared with the uncertainty on $a_{\mu, 28 \dots}$ in eq. (50) of the supplement (27.59±0.26).

The reasons given for choosing 2.8 fm versus 2.6 fm or 3.5 fm seem arbitrary.

Finally, it is hard to understand the upside from taking this risk. Attached is a comparison of the 2020 result [3/7] with the present claim (inner/outer error bars are reported statistical and total uncertainties). The red (top) point is my estimate if, as inferred from fig. 14, the uncertainty on $a_{\mu, 28 \dots}$ is 10% of the value given in eq. (50). It is still an advance over 2020.

In summary, the case for publishing is the reduction in uncertainty shown in the figure. The case against is the risk entailed in trusting the treatment of radiative corrections of e^+e^- data and discarding the criticisms of the τ method that led to its deprecation. The risk could be mitigated by providing LQCD only results for $(1.0, \infty)$, $(2.8, \infty)$ and the full $a_\mu^{\text{LO-HVP}}$. Of course, if my guess of 10% uncertainty on $(2.8, \infty)$ is optimistic, then the LQCD-only result is not interesting, and it seems the large computing resource invested in the $a=0.048$ fm ensemble could have been deployed more wisely.

(Remarks on code availability)

QUADA is well known and widely used, as are the programs in <https://arxiv.org/abs/1203.4469>

Referee #3

(Remarks to the Author)

A. Summary of the key results

The authors present an improvement of their previous determination of the leading hadronic vacuum polarization (HVP) contribution to the muon's g-factor, which is the quantity that accounts for the bulk of the uncertainty in the Standard Model (SM) prediction for the muon anomalous magnetic moment. An accurate determination of the HVP contribution is crucial for deciding whether the SM shows a quantitative failure. The manuscript describes the update of an earlier calculation by members of the same group of authors [Borsanyi et al., Nature 593 (2021) 51], which is based on a combination of a lattice QCD calculation with the traditional data-driven method. The final result for the HVP contribution produces an estimate for the muon anomalous magnetic moment with a total error that is comparable to that quoted in the 2020 White Paper [Aoyama et al., Phys. Rept. 887 (2020) 1], which is solely based on the data-driven method but whose validity has been challenged after the publication of new hadronic cross section data by the CMD-3 experiment.

B. Originality and significance

For many years, the muon anomalous magnetic moment has provided one of the most robust hints for physics beyond the SM, chiefly because the traditional approach for evaluating the HVP contribution produced a sizeable discrepancy with the direct measurement. In recent years, starting with the earlier calculation by the authors, many lattice calculations have shown a significant tension with the data-driven method when applied to certain sets of hadronic cross section data. The current manuscript corroborates these findings, producing an estimate for the muon's g-factor that is compatible with the most recent experimental average. The authors argue that in their case the combination of lattice calculation with the data-driven method is safe, since the latter is applied in a kinematical region where there is no obvious tension among different experiments.

The hybrid approach, i.e. the combination of lattice QCD and data-driven methods has first been proposed and applied by the RBC/UKQCD collaboration [Blum et al., Phys. Rev. Lett. 121 (2018) 022003]. In that reference, however, the kinematical range was chosen such that the final result was dominated by the data-driven method. In the manuscript under review, lattice QCD accounts for about 95% of the final estimate.

Aside from employing the hybrid approach, the manuscript reports on the incremental improvement of the lattice calculation, by adding two ensembles at finer lattice spacing and increasing statistics on existing ensembles. While the availability of data at smaller lattice spacing improves the control over the continuum extrapolation, the use of the data-driven method is crucial for reaching the final precision of 0.46%, since it dramatically reduces the uncertainty associated with the long-distance tail of the integrand and also produces a more precise estimate of finite-volume corrections.

C. Data & methodology

Both lattice QCD calculations and the data-driven method are well-established methods. Compared to their previous publication, the main result is obtained by combining the lattice result for the HVP restricted to integrating over the Euclidean time "window" from 0-2.8 fm with the data-driven estimate for the remaining interval from 2.8 fm to infinity.

While the results are of high quality and also of high relevance for the search for new physics, I cannot help but noticing a certain lack of transparency regarding the presentation. This concerns the material per se, as well as the ability to perform comparisons with results obtained by other collaborations. Given the high interest in the muon anomaly in the entire particle physics community, I think this is a shortcoming that should be rectified in a revised version of the manuscript. I elaborate on this in section F of this report.

D. Appropriate use of statistics and treatment of uncertainties

The statistical treatment is based on standard procedures and methods, such as jackknife error estimates, binning and model averages. There are one or two instances in which one may question the soundness of the model average (see comment 8 in section F below).

E. Conclusions: robustness, validity, reliability

To judge the robustness and validity of the conclusions, additional information should be provided by the authors. This concerns, in particular, the addition of scaling plots for the isovector contribution in the range from 1.0-2.8 fm, an extended discussion of the difference with their previous calculation and additional comments on the validity of taking over the calculation of isospin breaking effects. This is particularly important in view of the high interest in the muon anomaly and its role regarding the validity of the SM. A detailed list of recommendations is given in the next section of this report.

F. Suggested improvements:

Main part of the paper:

1. The authors state on p6 that the added finer lattice spacing reduces the uncertainty in the light connected contribution by 37%. Does this reduction refer to the entire integration range or to the one-sided window from 0-2.8 fm? If it is the latter, this statement cannot be verified by the reader since there is no result for the one-sided window in the earlier publication.

2. In the penultimate paragraph on p8 the authors state that their new result is larger by 6.5(5.5) units of 10^{-10} . In order to improve transparency, this should be supplemented by a discussion which part(s) in their updated analysis is/are responsible for the upward shift (there is a similar request relating to p29, see comment 11 below).

Supplementary Material:

3. The calculation of the Omega mass described in section 2 is "based on the GEVP". The procedure that is employed by the authors is known as the "generalized pencil-of-function" (GPoF), and I suggest that, for clarity, the authors change their statement that their procedure is "related to the GEVP" and mention the GPoF as well.

4. When arguing for the stability of their results for the Omega baryon mass in Fig. 6, the authors provide no actual values. I suggest that the authors show the value of the ensemble average by adding a separate scale on the right-hand abscissa.

5. Top figure of Table 3: why do the data points appear slightly shifted along the y-axis between the plots of the left- and right-hand sides?

6. In section 4.1, the authors describe their motivation for the fit ansatz of the continuum extrapolation. To this end, they use either polynomials in a^2 or in Δt_{KS} but no combination of the two. They should elaborate on why they chose to avoid an ansatz mixing both terms.

7. In footnote 4 of the manuscript, the authors appeal to a new paper by RBC/UKQCD (which contains an estimate for the so-called long-distance window from 1.0 fm to infinity) as evidence supporting a larger value for the HVP contribution compared to their 2020/21 publication. It would be much more convincing if the authors included their own estimate for the long-distance window, to allow for a like-by-like comparison between their calculation and those of other groups. This would not only improve the transparency of the presentation but also give the authors the chance to provide more details as to which part of their calculation leads to the larger estimate (see comment 2 above).

8. The updated estimate for the disconnected window observable presented in Table 7 differs strongly from the 2020/21 result, yet this is not discussed in the manuscript. At the same time, the two peaks in the PDF are clearly inconsistent. Judging from the two scaling plots at the top, the two-peak structure arises from the two alternative fit forms, using either polynomials in a^2 or in Δt_{KS} . This may be an example where the restriction to polynomials in either variable but not allowing for mixed terms is too restrictive. Eventually the authors leave it to the AIC to select the most likely value, but this procedure may be biased if the input is too selective. I also wonder whether the addition of a finer lattice spacing is responsible for the smaller estimate for the disconnected contribution relative to the 2020/21 publication, at least when only polynomials in a^2 are considered. I recommend that the authors comment on this issue and expand the discussion.

9. The authors present the isoscalar contribution to the 04-28 window observable in Table 9. They should also include a similar table for the much more relevant isovector contribution. In fact, I am puzzled why the manuscript does not contain this information, which is by far the most crucial part of the lattice calculation. By including the corresponding scaling plots, PDF distribution and table, the authors will have the chance to address the lack of transparency which I have alluded to earlier. It would also be instructive to show individual scaling plots for the three windows 04-06, 06-12 and 12-28. I understand that a simultaneous fit is performed to all three, with individually chosen fit functions. Nevertheless, one can easily reconstruct and plot the fit curves for the individual windows. This would allow the reader to assess the influence of taste violations for the three "sub-windows" and how these are treated in different kinematical regimes.

10. Compared to the 2020/21 publication, the errors quoted for the strange and charm contributions listed in Table 10 are larger by a factor 2 and 4, respectively. The authors should provide an explanation what led to the larger uncertainties.

11. The discussion of the overall difference with the 2020/21 result on p29 is only focused on the estimation of the uncertainty attributed to the shift, but makes no attempt to trace the origin of the difference itself. I am also not convinced that one may treat the errors of the continuum extrapolation as uncorrelated - even if the observable is divided into several sub-windows - since the vast majority of gauge ensembles enters both the 2020/21 calculation and the 2024 update. At any rate, the difference with the earlier result is most likely due to the long-distance window, since the intermediate-distance window is consistent with the 2020/21 calculation, and a shift of about 6 units in the short-distance window would represent a 10% effect in that quantity, which is an unlikely scenario. This makes it even more important that the long-distance window is included and appropriately discussed in the manuscript.

12. As outlined in section 7, isospin-breaking contributions are taken over from the 2020/21 publication, except for the re-tuning of time cuts and fixing the value of the one-photon-reducible contribution. Given that the time cuts for the light-quark sea-QED contributions are shifted to larger values (2.2 and 2.7 fm, respectively, which corresponds to a factor of two), I find it very surprising that the statistical errors change only marginally. Furthermore, given that a hybrid approach is used, I would expect that the evaluation of isospin-breaking corrections should be restricted to the 0-2.8 fm window, to avoid double counting in the long-distance tail. If time cuts as large as 2.7 fm are employed, I wonder whether this has been implemented correctly. The authors should elaborate on the estimation of the uncertainties of the various isospin-breaking contributions and add a discussion on the evaluation of isospin-breaking corrections in the 0-2.8 fm window.

13. In the right-hand panel of Fig. 11, I presume that the discrepancies between the evaluations of finite-size effects based on the CMD-3, BaBar and KLOE data sets reflects the well-known tensions among the evaluation of the two-pion contribution via the dispersive method. If the authors agree, they could insert a comment to this effect.

14. Regarding Figs. 13 and 14, it would be much more transparent if the absolute values were plotted rather than just the difference from the average (which can be shown separately as a band in the plots).

G. References:

The discussion of the tension between data-driven and lattice estimates for the intermediate window observable on p39 lists only one lattice calculation (Ref. [39]) when there are many more such results with equal or even better precision. This should be corrected.

H. Clarity and context:

Abstract, introduction and conclusions are clearly written and appropriately discuss the context of the research.

(Remarks on code availability)

Referee #4

(Remarks to the Author)

I preped to my report short answers to the questions suggested by Nature for structuring the report. The details are found below.

A Summary of the key results:

.....
The submitted work is an update of a previous Nature publication on the comparison of Standard Model (SM) theory to high precision measurements of the muon magnetic moment. As emphasized in the paper this is of considerable interest for science as a whole. The reason is that there was disagreement with the then-considered best SM prediction before the previous publication. The disagreement and thus the evidence for a needed extension of the SM was eliminated by the previous publication.

The SM computation is complicated and partially also subtle and the precision of the experiment has been and will be further improved. Thus the question of agreement arises again - at a higher level of precision.

B Originality and significance:

.....
The update is not simply an extension of the simulations but also a change of strategy, namely the long time tail is taken from experimental data. While the strategy is not entirely new, it is the first time it is thoroughly and successfully applied.

C Data & methodology:

.....
Overall, the authors follow a valid approach (one could doubt the use of rooted staggered fermions, but mostly the community does not regard this a serious issue). The simulations are very well planned and carried out and exceed those of competing collaborations by a strategic planning and efficient investment of very large resources. There are some deficits in the analysis of the numerical results and in the presentation which can be fixed.

D Appropriate use of statistics and treatment of uncertainties:

.....
This is mostly contained in point C. Upon taking into account my comments on the analysis, the uncertainty may change a bit. It is not clear from the beginning whether the estimated uncertainty will become larger or smaller.

E Conclusions: robustness, validity, reliability

.....
The conclusions are solid but may change a bit due to C and D.

F Suggested improvements: experiments, data for possible revision

.....
Below I suggest improvements of the analysis and in particular a much improved publication of the results. The latter is necessary to support future use and checks of the simulation results by others.

G References: appropriate credit to previous work?

.....
References are correct concerning the latest particular publications on the various analysis methods. But there is not much effort to also cite the foundational works. In the main article this is probably due to restrictions imposed by Nature, but in the Supplementary Information it shows a somewhat insufficient consideration of the foundations / historical development. This deficit is linked in several places to a style of just stating what is done without explaining why. The latter question is then left to the reader to know or figure out. Explicit examples are given below.

E Clarity and context: lucidity of abstract/summary, appropriateness of abstract, introduction and conclusions

.....
The authors made a conscious choice of writing an update, not a self-contained

article. One has to read the previous article as well.

I agree with this choice, which is probably good for a large part of the readers.

Detailed Report

The submitted work is an update of a previous Nature publication on the comparison of Standard Model (SM) theory to high precision measurements of the muon magnetic moment. As emphasized in the paper this is of considerable interest for science as a whole. The reason is that there was disagreement with the then-considered best SM prediction before the previous publication. The disagreement and thus the evidence for a needed extension of the SM was eliminated by the previous publication.

The SM computation is complicated and partially also subtle and the precision of the experiment has been and will be further improved. Thus the question of agreement arises again - at a higher level of precision.

Because of the importance as well as the fact that the updated computation is by far the best one that exists, I support a publication in Nature despite a few shortcomings which I will list in the following. I expect most of them to be eliminated before publication.

I first describe my criticism in broad and general terms.

Afterwards I list details which explain the general criticism and go beyond.

1) General criticism.

1a) The abstract labels the reduction of uncertainty as "a factor of almost 2", while the main text makes this precise as a factor of 1.6. (This factor 1.6 has a significant uncertainty itself.)

1b) The results are not given in a transparent way.

The main part just lists results and numbers, which is a valid choice. However, one then expects that the Supplementary Information can be used, e.g. to understand how the uncertainties, in particular (c) and (d) of Figure 1, come about and how well one knows them. This is not really the case.

1c) Very much related to 1b) there is a lack of information which means that the results, except for the final one, will be very hard to be checked and used by others working in the field. In my opinion, this is not acceptable given how relevant checks are and even more how relevant it is to reduce uncertainties, using also results from other computations. Several such alternative computations exist and science should profit from their combination. One also has to note that the computations were very expensive, which means it is entirely unrealistic that they will be repeated from the beginning. There are various intermediate results which are not provided in the needed form. Also the gauge field configurations which they are based on are not publicly available. Most importantly the analysis of the numerical results is not explained and justified sufficiently. These facts mean that in the present form the publication does not really qualify as reproducible and accessible.

1d) A somewhat more technical but essential issue is how the continuum limit is taken and how its uncertainty is estimated. The lack of provided information makes it difficult to judge it completely, but the largest two lattice spacings seem outside of the region which is useful for an extrapolation. It looks like the expansion in terms of the lattice spacing has broken down for those lattice spacings. Removing them might not change final results much but simply needs to be done in order to be more confident. The continuum limit of the short distance window is hard to believe the way it is taken / checked.

1e) A further technical issue is the determination of the scale. The estimate of the systematic uncertainty is insufficient.

These issues may seem minor, but of course reducing an uncertainty by a factor 1.6 is a matter of detail.

2) Details (some of them provide explanations for 1).

Abstract:

=====

"factor of nearly 2" needs to be changed to "factor of 1.6" (or whatever emerges).

Main part:

=====

Figure 1 and its description:

It is unclear how much of the statistical error (a) is contained again in (b-e) or whether (b-e) contain independent statistical uncertainties.

E.g. (b) might contain a large statistical error. This is important since the statistical errors are much more firm than the systematic ones. A remark needs to be added.

I 129: "... light connected contribution is decreased by the new ensembles by 37% ..."

The statement is not supported by the Supplementary Information. The decrease could also be due to the change in analysis (separate analysis in different windows).

I 150: "expected to be similarly small" is a speculation. Furthermore it seems to me the authors want to say "difference with schemes used" and not "difference with results obtained".

I 160: "quantitatively exact" and the entire bullet point. I do not understand well what the authors want to say here. What is scheme dependent and what is not? This needs to be explained better.

Supplementary Information:

=====

Figure 1 shows that only half of the beta-values allow for an interpolation to the physical point. The extreme case is the largest lattice spacing with a single point only, which is rather far away. I see no evidence that the global fits -- in particular those that dominate the result -- solve this issue. Based on the figure I conclude that beta=3.700 and maybe also beta=3.7753 need to be removed from the analysis.

p5, 1st paragraph: The smallest lattice spacing is rather small and topology change is expected to be suppressed. While the shown autocorrelation function looks good, it will improve the paper and support its conclusions to also list the topological susceptibility at the smallest lattice spacings and compare it to expectation.

Of course, if that is published elsewhere, one can just point to the literature.

Figure 3: the figure compares data to a^4 while the text discusses $a^2 \alpha_s^3$. The latter should be plotted in the figure.

p6, last paragraph: referring to [14] does not help, since there is no justification given in [14].

Section 2: There is a small amount of references. Various techniques go back to the literature and this should be shown. An example is that eight sources (bottom of p 7) was proposed and tested already in Phys.Lett.B 162 (1985) 160.

The discussion on excited states on $p8$ is too naive. Why are the states resonances? The relation of finite volume states to resonances is complicated. Why do the authors think that the states are not close to 2-particle, 3-particle states? What is the (free) low-lying multi-particle spectrum in the present setting? Can one expect that the GEVP-type analysis determines such states? Such a discussion will impact the question whether the analysis which was carried out is sufficient, in particular to estimate its uncertainty. I disagree completely with the present way of estimating the error due to the fit-range. Changing β_{min} by 0.1 fm does not change the contribution of excited states much. Therefore such a change underestimates the uncertainty. More on the speculative side, in fig. 4 there seems to be a bit of a step-like structure below and above 0.8 fm. It looks as if below 0.8 fm the GEVP captures the 2nd excited state (or some state around there) and above it loses this information. One worries whether this means that the data points for the ground state are better between 0.5fm and 0.8fm than above. Of course this is against standard expectation. To be able to judge whether the step-structure is just a statistical fluctuation, one would like to see more (of the small a) ensemble results.

In the present form, the systematic error due to excited states is not seriously accounted for. The 0.15% shown in figure 1 could be quite off.

Section 3:

The last sentence before 3.1 triggers the question whether just the number from the publication is taken or whether the correlation with the new analysis is taken into account. This needs to be stated and possibly justified.

Figure 7 and its discussion: Ref.[24] does an electro-quenched computation. Should one mention that or compare to the electro-quenched result in the BMW scheme? In [25] I could not find a definition of the scheme at all. The scheme of [25] therefore needs to be defined here, such that the reader can judge what is shown. The last sentence of section 3 is incorrect. The Kaon decomposition can be correct at the level of 0.1MeV-0.2MeV in the splittings as seen here and still the scale can be different by e.g. 1% and then a_{mu} by 2%. The true check for scheme-sensitivity is the one given in the main text.

Section 4 and continuum extrapolations in other sections:

1) What is a_m in Table 3?

2) The global fit procedure and therefore the continuum extrapolations are rather intransparent.

A central deficit is that it is not possible to find out what kind of fits dominate the analysis and where systematic errors come from. There is no information on the AIC weights in the form of some table or graphical representation. This needs to be changed. A first step is to drop fits with very low weights from the figures.

For example, Table 3, top figures, contain a number of fits which miss the data points by far. These are plotted nevertheless and make it impossible to see where the green region below 172 in the lower figure as well as the long tail to higher values in the lower figure come from. One also has to note that the PDF discussion sounds like objective statistics, but it depends in an arbitrary way on which fit functions have been chosen. E.g. there is an underlying assumption that the true model is contained in the set of fit functions. This is definitely not the case here. The true function describing cutoff effects is much more complicated with many powers of a and α_s .

A few more specific remarks follow.

3) The data points in Table 3 provide very strong evidence that the largest lattice spacing is not described by a few powers any more. The a -expansion has broken down.

It could also be that the simple linear $B(a^2)$ and $C(a^2)$ are entirely insufficient to describe the deviations from the physical point at larger a . Either way, the largest one or two lattice spacings need to be dropped completely and then one would like to see what is the systematic error when more lattice spacings are dropped.

4) I see no justification for a fit-function with just $A(\Delta_{\text{KS}})$ for the discretisation errors. Everywhere w_0 is involved through the lattice spacing. This is a pure gauge and gradient flow quantity, where taste splittings are very subdominant.

The pure gauge and gradient flow terms in the Symanzik expansion will dominate. Of course a fit-function

with two or three powers of α may be reasonable. Again dropping larger lattice spacings helps to be less sensitive to the detailed form of the extrapolation. It is also not satisfactory that there is no discussion about what is known about the discretisation errors, in particular of gradient flow quantities and how this leads to the fit-functions used. In my opinion any discussion will lead to the conclusion that the fit-function is essentially unknown but (as stated earlier) it is extremely hard to imagine that the discretisation errors at small lattice spacings are dominated by taste breaking effects.

5) Table 4: the scale of the figure makes it impossible to judge what happens at small a , in particular at the level of the cited total error. In addition, for the small distance window the situation is special beyond this simple point. As noted in the text the quantity is not on-shell. Therefore there is no theory for the discretisation errors and statements such as "logarithmically enhanced" may be true at tree level but are not backed by anything beyond that. The functional forms (24) are ad-hoc. Without a demonstration that the short distance window (or if needed even shorter distance one) is compatible with a continuum perturbative estimate, the result (25) cannot be trusted.

The authors have a statement that they know the sign of the log-enhanced term at tree level. They should show how they arrive at this and give the numbers. The question also arises whether it is seen in the numerical data.

6) Table 6: the "Continuum parameter" seems too small, judged from the figure.

7) Table 7: same as Table 6.

8) I have expressed my strong reservation concerning the probability interpretation of the plotted PDF.

Still, as they appear in the paper, they show long tails and double-peak structures. The reader would then like to know where these structures come from and what this means for the systematic uncertainties.

Additional suggestions concerning 1c):

A way to make the results usable and testable for other groups is to publish the full covariance matrix of all results. Alternatively it may be convenient to publish the jackknife bins. Of course "publish" means to provide machine readable files.

By "all results" I mean the results for the physical point and all different contributions to a_μ , i.e. different windows, light, strange ...

I further consider it necessary to provide these numbers before extrapolation to the physical point and continuum limit as well. This level of details is standard in the field and thus a minimum for Nature publications.

(Remarks on code availability)

It is not possible to check what is used since "Oattice" mentioned in suppl. sect. 1.1 is unpublished. This is a major part of the analysis software.

Version 1:

Reviewer comments:

Referee #1

(Remarks to the Author)

Key results. This manuscript discusses a calculation of the leading-order hadronic vacuum polarization (LOHVP) contribution that forms part of the theoretical calculation, in the Standard Model (SM), of the anomalous magnetic moment of the muon, a_μ .

When this paper first appeared in July 2024, its motivation was to provide a result for the LOHVP that would clarify the theoretical situation at that time, ahead of the final result from the Muon $g-2$ experiment. The theoretical picture was unclear because the two different approaches to the determination of the LOHVP, the "data-driven" approach and the lattice QCD approach, gave conflicting values for part of the LOHVP. The data-driven approach, which was used to give the community preferred result for the LOHVP in the 2020 theory white paper on a_μ , had also been called into question by new experimental results for some of the data input needed for that method. The LOHVP value in the 2020 theory white paper implied there must be new physics present in the experimental muon $g-2$ determination. Using the newer experimental input led to a data-driven LOHVP with a very different conclusion, i.e. no new physics in muon $g-2$. This data-driven result also agreed with lattice QCD values for parts of the LOHVP where accurate lattice QCD results could be obtained. In July 2024 this paper provided a 0.5%-accurate value for the LOHVP which is mainly (95%) from lattice QCD, using a small contribution from the data-driven approach for the large-Euclidean-time "tail" where lattice QCD uncertainties become large. The resulting a_μ value agreed within 1sigma with the experimental value at that point and disagreed by 4sigma with the 2020 theory white paper. The impact of this in overturning the 2020 theory white paper result was clear (and indeed the paper has had significant impact).

A year later, in August 2025, the consensus in the field has changed (partly because of this paper). The 2025 theory white paper gave up the data-driven determination of the LOHVP and used a lattice QCD result with a 0.85% uncertainty obtained from averaging values from several different collaborations, including pieces from this paper. The SM a_μ from the 2025 white paper was significantly larger than that from the 2020 theory white paper and the conclusion was that "there is no tension between the SM and experiment at the current level of precision".

Viewing this paper in the light of the current situation, the key result is no longer to change the consensus on the theory LOHVP value but is now simply an improved uncertainty over the value in the 2025 theory white paper. Comparing 0.45% to 0.85% this improvement is almost by a factor of two. This does then represent a significant advance with influence on the field and I am inclined to recommend its publication (following suggestions for further improvements below).

Validity. The calculation presented here uses lattice QCD for ~95% of the LOHVP contribution and a data-driven result (i.e. using e^+e^- to hadrons data) for the remaining contribution in a region (at large Euclidean time above 2.8 fm) where lattice QCD uncertainties grow. It is the use of this technique, plus the determination of results at a finer lattice spacing than BMW used in 2020 that enables the achievement of a 0.45% uncertainty. As discussed above, the size of the uncertainty is critical to the impact of the results.

The presentation of results in the Supplementary Information has been much improved in this version of the paper, in response to comments. This includes more information about the different windows within the 0-2.8fm range used in the lattice QCD calculation with values that other lattice QCD collaborations can use for comparison in future. It also includes improved analysis of the tail contribution from Euclidean time values 2.8fm to infinity that is taken from a data-driven approach. I am happy that the points I raised in my previous report have been addressed in this version.

Originality and significance. The originality of the paper comes largely from the novel method of combining lattice QCD values from a large but finite time region with a data-driven approach that can fill in the very large time region up to infinity.

The level of accuracy achieved and how it compares to the value being used in the community is key to the significance of this paper. The paper itself is not very clear on this, in my opinion (although the title has changed to emphasise the size of the uncertainty reached). The discussion of error reductions is largely focussed on previous BMW calculations, rather than the current community consensus which is the 2025 Theory Initiative result. The paper simply says on line 212 that "Our LOHVP is in good agreement with the latest Theory Initiative combination" without making the key point that the uncertainty here is almost a factor of 2 smaller? (The reader can see it is smaller from the figure but the amount of the uncertainty reduction should be spelled out to help the reader appreciate the importance of the result).

Data and methodology. These are fine

Appropriate use of statistics and treatment of uncertainties. Fine

Conclusions. The changes made to the paper, particularly in the Supplementary Information, have improved significantly the robustness, reliability and validity of the conclusions.

References Fine

Clarity and context

I suggest below some further improvements that would help make the conclusions, along with some other points, clearer to the reader.

1) I think the first paragraph is a bit confusing for the general reader now, particularly the sentence on line 33; "Furthermore, new input data ...similar changes". The confusion arises because the reader does not know at this point why "input data" are needed for the

LOHVP, nor which LOHVP result the comment refers to. I think this needs to be changed, but I do not have a good suggestion for how to do this succinctly.

Another point of confusion in this paragraph is the reference to a 2021 lattice QCD calculation that is later, in Figures 2 and 3, tagged as BMW'20.

The motivation for the paper would be improved for the reader by making clear the uncertainty on the LOHVP in the recent update of the reference standard-model prediction (lines 34-35).

2) lines 62-62 would read more smoothly as:

... that allow QCD predictions to be made ...

3) Line 191 typo: of our -> our

4) Line 211 - from the Figure it is clear that the plotted CMD-3 data-driven result does not have "serious tensions" so this sentence needs to be more carefully written.

5) Line 212 would make a stronger ending to the paragraph if it made clear the improvement in uncertainty achieved here compared to the latest Theory Initiative combination.

6) Line 214 (. confirm or refute our results) reads a bit oddly now in 2025 that the theory consensus (including results from other lattice collaborations) and this paper agree on the picture presented. The issue now is not one of confirming or refuting but whether other lattice collaborations can achieve comparable accuracy (perhaps using the approach suggested here) and whether the uncertainty on the theory result for a_μ can become as small as the experimental uncertainty.

7) New Section 3 of the Supplementary Information: Figure 8 was rather confusing - the caption should be explicit about the regions shown as bands. From the text in section 3.1 I was expecting 3 bands but there only seemed to be 2. How the systematic uncertainty for residual excited state contamination is determined should be made clearer.

(Remarks on code availability)

Referee #2

(Remarks to the Author)

The authors are to be commended for the work they've done to strengthen the LQCD part of this work. Unfortunately, they did not respond to some big-picture comments in my first report.

There I wrote, "the lack of a result for (1.0, ∞) detracts from the value of the current version of the manuscript. I've not encountered anyone who excuses this omission..."

Also, "Everyone [I had until then encountered] wants an eventual published version of #2024-10-23178 to provide LQCD-only results for (1.0, ∞), (2.8, ∞) and the full a_μ LO-HVP. The LQCD data to obtain such results are clearly in hand.... A paper that wanted nothing but to move the science of the muon's anomalous magnetic moment forward would include this information."

And "Many of us see only a downside ... to omitting the full LQCD result."

The revision omits the requested information, and the authors' responses do not address these comments. The revision is, thus, not responsive to my report.

Let me summarize what the quoted sentences mean, in case anyone did not understand. This work is not suitable for publication without LQCD-only results for $t/m \in (1.0,\infty), (2.8,\infty)$ and the full a_μ LO-HVP.

I also wrote, "If, despite the criticism given above, the hybrid result remains the central (or, heaven forbid, only) result, change the title to "Hybrid determination of the ...". The authors did respond to the suggested change in title, saying no one would understand what "hybrid" means. This is disingenuous: every reader would be on the alert for a mixture of two (or more) methods, and the last sentence of the abstract removes the suspense. Further disingenuousness is the authors' assertion "the title does not claim that this is a lattice calculation." Calculations by researchers famous for their lattice calculations are generally assumed to be lattice calculations.

Again, in case anyone does not understand: the title of a published version should not overstate the main result.

Since the time of the first round of reports, a lot has happened. FNAL E989 has reported its final measurement, and the Muon g-2 Theory Initiative released an updated white paper. I also encountered three people who don't object to the hybrid determination and lack of LQCD-only results (i.e., the other referees).

My first report asked, however, "What happens if the next TI white paper recommends putting hybrid results aside for now?" That is essentially what has happened. The TI does not recommend a value for the full a_μ LO-HVP based on $e^+e^- \rightarrow$ hadrons or on τ -lepton decay data. This decision was taken by the experts on these approaches. The TI considered hybrid determinations as possible HVP recommendations and rejected the notion. So, although the authors provide better arguments for trusting $e^+e^- \rightarrow$ hadrons and τ decay, the community of experts doesn't fully accept them. The white paper reports problems in the understanding of the implementation of radiative corrections. Mistakes in this area could change the $e^+e^- \rightarrow$ hadrons cross section by many times the purported uncertainty. At the same time, old criticisms of the τ decay data have still not been refuted. Therefore, the authors' observation that "doubling the uncertainty" would not affect the quadrature sum, while true, misses the point.

While the authors were revising, and in anticipation of the 2025 white paper, several collaborations published (in prestigious particle physics journals) results for the long-distance window. The white paper would have been improved with LQCD-only results for this window from the work under review. The response states that the revision now provides what the TI wants. No, it does not.

Particle physicists are gradually coming to terms with the demise of one of their favourite discrepancies. The earlier Nature article played an important role in starting this reckoning. This work will not have such impact. It appears on summary plots with awkward disclaimers about its reliability. As I wrote before, the hybrid is a risk I (and many others) wouldn't take. Presenting LQCD-only results for (1.0, ∞), (2.8, ∞) and the full a_μ LO-HVP is the way to ensure a long life for the work under review.

Let me restate the main points, in case anyone does not understand. I concur with the experts in the TI who think a hybrid determination is unreliable. The unreliable results must be supplemented with LQCD-only results (surely in hand) out to infinite time extent. If the central result is a hybrid determination, the title must say so. Otherwise, I cannot recommend publication (in Nature or elsewhere).

Of course, the best paper would be LQCD-only from beginning to end.

(Remarks on code availability)

The code is so well known, I don't need to review it.

Referee #3

(Remarks to the Author)

I thank the authors for their considerable efforts to improve the manuscript by addressing the comments raised in my earlier report as well as those by the other referees. Nevertheless, there are still several significant shortcomings which prevent me from recommending the manuscript in its present form for publication.

A. Summary of key results

The main result, i.e. the leading-order hadronic vacuum polarization contribution to the muon's g-factor has changed marginally compared to the original version. In the revision the authors now also quote an estimate for the long-distance window quantity from the "hybrid" approach, i.e. from the combination of their lattice calculation with the data-driven dispersive method applied to the tail of the integrand. Furthermore, they have reverted to their pre-2020 practice of using the pion decay constant to assign a physical value to the scale-setting quantity w_0 . Thirdly, they have revisited their determination of isospin-breaking corrections to conform with the restriction of the integration interval to Euclidean times below 2.8 fm.

B. Originality and significance

Following the release of the final result from the E999 experiment at Fermilab, the publication of several new lattice results on the same quantity, and the publication of the White Paper update on the Standard Model prediction, there is continued high interest in the wider community, even though the tension between experimental observation and theoretical expectation has disappeared at the current level of precision. The revised version of the manuscript refers to some of these recent developments, yet a more thorough comparison of their estimate for the long-distance window is missing (see comment 5 in Section F of this report).

C. Data and Methodology

The main feature of the paper is the use of a "hybrid" approach that combines lattice QCD with the data-driven dispersive method. The latter is applied in the far infrared regime where the lattice calculation struggles to constrain the observable with sufficient precision. While the difficulty to map out the long-distance regime with small statistical errors is common to all lattice calculations, it could be a lot more acute for the particular discretization chosen by the authors, i.e. rooted staggered quarks, which suffers from additional discretization artefacts, commonly referred to as taste-breaking effects. In contrast to their earlier work, the authors refrain from quoting any result from their lattice calculation of the vector correlation function beyond 2.8 fm. Therefore, despite several improvements of the manuscript, I still note a lack of transparency which is particularly unfortunate given that several other groups have published new results in the run-up to the submission process.

D. Appropriate use of statistics and treatment of uncertainties

While the statistical treatment is based on standard methods, I continue to have reservations concerning the discussion of the tension with the earlier (2020) result by the authors (see my comment 4 in Section F).

E. Conclusions: robustness, validity, reliability

To judge the robustness and validity of the conclusions, additional information should be provided by the authors to enhance the transparency of the presentation of the results. This concerns, in particular, the discussion of the difference with their previous calculation, quoting a "lattice-only" estimate for the long-distance contribution and a discussion of the role of taste-breaking effects in the light-quark connected contribution. Full details are listed in Section F of this report.

Main part of the paper:

1. I still have severe reservations about the estimation of the significance of the difference between the 2020 result and the new estimate described on p9. I elaborate on this issue further below (comment 4 below).
2. The sentence "In the near future we expect that other lattice collaborations [...] confirm or refute our results" is not needed, since new results have been published and a consolidated lattice result has emerged as part of the White Paper update.

Supplementary Information:

3. Section 6, Eq. (57). I appreciate the efforts to address my comment on elaborating on the improvement of the precision relative to the 2020 publication. I am puzzled, however, why the authors only quote the uncertainties but withhold the actual values of these quantities. In order to properly assess the impact of the added ensemble, it is also necessary to judge whether the results for the 0.4-4.0 fm window are compatible within errors. This is one of several instances where the transparency of the paper should be improved.
4. The discussion of the tension with their 2020 result on p48 is essentially unchanged. No explanation is offered regarding the source of the difference, with the discussion entirely focused on assigning an error to the difference (which is, in fact, as large as the total error on the entire 2020 result). I still disagree with the way that the uncertainty on the difference has been estimated. For once, the error on the finite-volume correction should not enter this analysis at all, since the estimate for the size of the correction has remained stable and should affect both calculations in a similar fashion, regardless of its error. Secondly, I strongly disagree with the argument that the two calculations can be treated as uncorrelated, since they are still based on largely overlapping ensembles, despite different analysis procedures. By far the most likely source of the difference with the earlier result is the continuum extrapolation of the light-quark connected contribution and the role of taste-breaking effects. This is where the added ensemble at finer lattice spacing should make a real difference as it may reveal a previously unnoticed systematic effect. This scenario could be tested by computing the correlated difference between the results whose errors are listed in the first two lines of Eq. (57). I realize that those estimates arise from model averages so that correlations cannot be preserved fully, but such a procedure would lead to a much more solid estimate of the uncertainty assigned to the difference and may offer an explanation as to why the result has shifted.
5. In section 6.6 the authors provide an estimate for the long-distance window observable based on the "hybrid" method. Actually, this is not what I had requested, which was rather an estimate of the long-distance window based on lattice QCD alone. In section 10, the authors speak of a "somewhat disadvantageous uncertainty of 8%" on the 2.8-3.5 fm window relative to the data-driven approach, but this cannot serve as an excuse for not quoting an actual number for the long-distance window based on lattice data, which would make the comparison with other recent lattice calculations (and the consolidated lattice average quoted in the White Paper update) of the same quantity much more meaningful. I cannot think of a scientific argument why this information should be withheld, unless the authors find that taste-breaking effects are uncontrollably large when extending the integration from 2.8 fm to infinity. If this were indeed the case, it should be stated explicitly in the manuscript as it would point to a limitation of staggered quarks, which manifests itself as an irreducible uncertainty in the long-distance tail. Only by adding more ensembles with finer lattice spacings could the integration be pushed further into the long-distance regime. That the treatment of taste-breaking effects is a highly sensitive issue has been acknowledged by the authors when they point out that -- contrary to their earlier practice -- they restrict the correction for such effects to distances larger than 1.2 fm. To summarize this point: For the sake of transparency, the corresponding result for the long-distance window from the "lattice-only" approach must be provided. In addition, the authors should add a plot comparing their result for the long-distance window to other recent calculations (RBC:2024fic, Djukanovic:2024cmq, FermilabLatticeHQCD:2024ppc), similar to the compilation plots in Fig. 2 of the main body or Tables 6-10 in the Supplemental Information.
6. Tables 23 and 24: insert the results for the window 10-28, similar to what was done for Tables 20, 21 and 22.
7. Figure 20: 0.55 MeV  0.55 GeV in the caption.

G. References

Section 6, footnote 6: During the journal submission procedure, two more results for the long-distance window have been published (Djukanovic:2024cmq, FermilabLatticeHQCD:2024ppc) that should be included.

On p66 the authors have added references to papers on "Low-mode averaging" which should also include DeGrand:2004qw and Giusti:2004yp.

H. Clarity and content

The issue of the intrinsic precision of the lattice contribution mentioned in comment 5 above also matters because the authors have chosen to include the final precision in the revised title of the resubmission. At that point it must be stated more clearly in the abstract (i.e. the part in bold face on p2 of the main paper) that this level of precision can only be reached through the combination of both approaches: While lattice QCD is used to circumvent the problem of (presently) incompatible e^+e^- hadronic cross section data in the region of the rho-omega peak, the data-driven approach is crucial to overcome the limitations in reaching a similar level of precision in the lattice calculation of the long-distance regime. It is crucial for the reader to appreciate the complementary nature of the two methods.

(Remarks on code availability)

The issues and criticism raised in my report are not related to the code.

Referee #4

(Remarks to the Author)

The new version contains considerable improvements. In particular the new determination of the scale is both a very noticeable new work and an important clarification concerning the validity of the results. Unfortunately, the new section needs improvements. Furthermore it remains impossible to judge whether the continuum extrapolations discussed in section 6 are trustworthy. Details about these two major issues as well as a number of other comments are given in the attached pdf-file.

(Remarks on code availability)

Version 2:

Reviewer comments:

Referee #1

(Remarks to the Author)

Key results. This manuscript discusses a calculation of the leading-order hadronic vacuum polarization (LOHVP) contribution that forms part of the theoretical calculation, in the Standard Model (SM), of the anomalous magnetic moment of the muon, a_μ . The size of the LOHVP contribution been a very hot topic for several years.

This is the third version of this paper and, since it first appeared in 2024, the community consensus on the question of whether the SM a_μ agrees with experiment or not has shifted. Consequently the motivation for this paper and the message of its main result have also shifted (and this is reflected in the updated paper).

The key point now is that this paper gives a result for the LOHVP in the SM with uncertainty 0.48%, compared to the theory white paper of 2025 which has an uncertainty of 0.85% using lattice QCD calculations (which is a change from the methodology for the LOHVP applied in 2020). Since the uncertainty in the LOHVP dominates that for the a_μ determined in the SM, this improvement by almost a factor of 2 is a significant one in allowing a more stringent comparison with experiment (where the uncertainty is much smaller). This does then represent a significant advance with influence on the field and I am inclined to recommend its publication (as I was for the second version). The points I raised about the second version have all been answered. I include below a few comments on this version.

Validity This calculation is a hybrid one, using lattice QCD for ~95% of the LOHVP contribution and data-driven results (e^+e^- to hadrons and tau decay) for the remainder at large Euclidean times where lattice uncertainties grow. The size of the uncertainty, critical for the impact of the paper, comes from combination of calculations on a finer lattice (than BMW had in 2020) and the hybrid technique.

The title of the paper has been changed in this version to make clear the use of the hybrid technique (a good move, I agree), the analysis of discretisation effects has been changed in response to other referee comments (this has little effect but does provide an additional test) and the authors now include (in the Supplementary material) a value for the light-quark connected isospin symmetric "long-distance" (1.0fm to infinity) window. This is also good because it allows comparison with other lattice QCD results (from 2025) that go into 2025 theory white paper. The new BMW number has an uncertainty which is 75% of the Theory white paper average so this result can be used to improve the theory only-lattice value (since the long-distance window provides the dominant uncertainty).

Originality and significance. As discussed before, the originality comes mainly from the novel hybrid technique. The significance comes from the accuracy achieved. The paper is much clearer (for the general reader) on these points now.

Data and methodology. These are fine

Appropriate use of statistics and treatment of uncertainties. Fine

Conclusions. The changes made to the title and the additions to the Supplementary Information, have improved the robustness, reliability and validity of the conclusions.

References Fine

Clarity and context

I have a couple of comments on the current version.

1) On line 35 I suggest adding a 'now' before 'using lattice results'. I think this would make clearer the differences in the theory consensus picture from 2020 to 2025 for the general reader.

2) New Figure 14 in the Supplementary Information: It looks as if an incorrect value has been plotted here for the FNAL/HPQCD/MILC '25 point. The caption says that the values are in the WP25 scheme, but then the FNAL/HPQCD/MILC central value should be just on the edge of the purple band (as in the comparison plot in the theory white paper).

It would be useful in the caption to note that the purple band is the average from the theory white paper.

(Remarks on code availability)

Referee #2

(Remarks to the Author)

The authors are to be commended for providing the standard long-distance window in the isospin-symmetric limit. They do not, however, fold this into a lattice-only calculation of the whole HVP. They argue that the hybrid result can be judged on its own. Their stance brings me to the only review criterion about which I have any more to say, namely "Reliability".

Reliability: It is simply not known at this time whether the e^+e^- and τ -lepton are reliable for this paper's purpose. One is not worried that the quoted uncertainty is off by a factor of 2, but that the central values are wrong by a factor of 2. Allowing for a factor of 2 in the tail contributing 5% to the total would lead to an uncertainty much larger than the 0.48% claimed for HVP. So if forced to judge the uncertainty of the hybrid result, it must be judged unreliable at worst, underestimated at best, and impossible to assess in any case.

In previous reports, I proposed that the authors and journal hedge their bets with a lattice-only result. I had two reasons. First, the lattice-only result is valuable on its own, not least as a way of documenting openly how the computer time invested in the new ensemble pays off. Second, I think there is a nonzero chance that the authors' judgement of the long-distance data-driven tail will hold up, in which case there would not be any harm in publishing the hybrid result now as a possible—if not definitive—view of where HVP stands now.

Unfortunately, I cannot recommend publishing this result on its own with the 0.5% uncertainty. The risk of learning in a couple years that the uncertainty is really 1% or 2% is too large. Such an outcome would sully Nature's and the authors' reputations.

I'm not saying anything new. I tried to find a path to publication, but the authors won't take it.

(Remarks on code availability)

I didn't look at it again. My comments from previous rounds hold.

Referee #3

(Remarks to the Author)

I appreciate the effort by the authors to address the concerns raised in my previous report. Below I list a few remaining comments.

(1) I welcome the addition of subsection 6.7 which describes the determination of a "lattice only" estimate for the long-distance window observable, which is a key ingredient for testing the consistency of all available lattice calculations. Although there remains a certain scatter among different lattice results for the long-distance window, as seen from Fig. 14, the overall picture that emerges is quite consistent regarding the actual values and the quoted errors.

(2) I also appreciate the efforts to improve the discussion of the differences with the earlier results obtained by the BMW calculation, presented in subsections 6.7 and 6.9. I have one final request on this issue, which concerns the statement in the main body of the paper (lines 205ff), which reads "The difference between our current and the 2020 result, accounting for correlations among uncertainties, is $7.6(5.2) \times 10^{-10} \dots$ "

This statement is at odds with the explanation on p. 50 of the Supplemental Material where the uncertainty on the difference is quoted as 5.2 and 4.5, respectively, when considering either zero or full correlations arising from the continuum extrapolation and remaining systematics.

(3) All other requests in my previous reports have also been addressed.

With these additions and improvements, the presentation is much more transparent, allowing the reader to make detailed comparisons and assessments of the results. I am satisfied that the revised manuscript meets the requirements that merit publication, once the statement listed under item (2) has been addressed.

Hartmut Wittig

(Remarks on code availability)

Referee #4

(Remarks to the Author)

I am satisfied with the largest part of the replies and the changes made by the authors. In an attached pdf-file I discuss the part that I am not satisfied with. A revision is necessary.

(Remarks on code availability)

Changes

Following the suggestions of the referees we performed the following major changes on the paper:

- Referee 4 raised concerns about the possibility of an unknown excited-state systematic in the lattice determination of the omega baryon mass. In response to this criticism, we performed an entirely independent calculation of the scale using the leptonic pion decay rate as input in a fully blinded approach. The methodology is described in a new section, Section 3 (Scale setting with the pion decay rate). Before unblinding we decided to adopt the new scale setting for all of our lattice results. The new scale turned out to be entirely consistent with the old one, as discussed in Section 4.1 (Physical point). Since scale has direct impact on the final result, the numerical values of most of our results have changed, though well within their quoted uncertainties.
- Referee 3 raised questions about our isospin-breaking corrections. Indeed, in the previous version of the paper we took over those corrections from our older work instead of computing them in the 00 – 28 window relevant here. We remove this shortcoming by computing them directly in the 00 – 28 window. As anticipated, the new numbers are in good agreement with the ones used in the previous version of the paper. Nevertheless, we now retain the new values. Accordingly, we have revised the results in Section 6.6 (Other contributions and total).
- To answer concerns of Referee 1 and 2 we now perform a more extended analysis of the tail observable. While we keep the central value unchanged, we consider several sources of systematics and quantify them. Though the error on that observable has increased, this does not impact the final error on our determination of the HVP contribution to the muon $g - 2$ because the tail is a small contribution. Our new procedure is described in the revised Section 10 (Long-distance contributions).
- Referee 1, 2 and 3 requested that we present results for the long-distance window, which we now do in Section 6.6 (Other contributions and total).
- Referee 1, 2 and 3 requested more details about our analyses, which we satisfy by a revised presentation in Section 6 (Window observables) with five new tables (analysis data-sheets) and two new figures.

All substantial changes are highlighted in blue in the manuscript.

Answers to Referee 1

We thank the referee for their valuable comments, suggestions and criticisms. We have accepted all of them and have revised the paper accordingly. The original comments of the referee are reprinted here in blue in the order they were made. After each comment we detail the modifications which we made to our analyses and to the manuscript.

comment r1i00

The value obtained for the largest contribution to the whole calculation i.e. the light quark connected contribution is not evidenced by any plots or error budgets. They need to be included to allow the reader to judge the statements made.

We have added the missing plots and error budgets in Section 6 (Window observables).

comment r1i01

The authors split up the 0.0–2.8 fm region into four windows. The results for the first window (0.0–0.4 fm) are displayed in Table 4, having been discussed in section 5.2. The other windows are not shown. The authors stress in section 5.4 that splitting the region up into several windows is beneficial because it allows different forms of continuum extrapolation to be used in each window, reducing the overall uncertainty. It is important for the reader to see how this works in practice through plots of the results of the different windows and the fit functions used. It is also important to have error budgets for each window to see how the total uncertainty is pieced together. The lack of this information is a flaw in the manuscript that means it should not be published in its present form.

We have included the continuum limit plots and error budgets for each of these windows in Section 6.4 (Light and disconnected 00 – 28 window) and also a summary plot showing the resulting uncertainty for many possible splittings.

comment r1i02

A further issue concerns the calculation of the contribution from the region above 2.8 fm. This is done using experimental data for the cross-section to e^+e^- to hadrons. The authors note that at such large Euclidean times the contribution is dominated by the data at very low values of centre-of-mass energy. They claim that, at such low values, there is no significant disagreement in the experimental data despite the tensions seen in e^+e^- to $\pi^+\pi^-$ between CMD3 and earlier experiments. Using results from BaBar, KLOE, CMD3 and tau data they quote a result for this small but not insignificant contribution with a 1% uncertainty. The statement about the lack of tension in the experimental data seems surprising given some of the other plots shown here and also other results in the literature. For example, arXiv:2311.09523 (Figure 3) shows a 4% difference between a data-driven result using the KNT19 compilation of experimental data and that substituting CMD3 numbers for e^+e^- to $\pi^+\pi^-$ for a large-time window between 1.5 fm and 1.9 fm. Arxiv:2410.23832 Table 2 also shows a roughly 4% difference for these two cases in the 2.8fm to

infinity window. I think a more complete analysis of the experimental situation is needed here to be sure that the 1\% uncertainty on this piece is justified.

We concur with the referee, the agreement seen in our calculations of the two-pion part of this tail window is indeed surprising. We consider all four experimental data sets that fully cover the energy range that is critical for this calculation, and as the referee correctly points out, there are significant tensions between these experiments. However, these tensions are concentrated on the ρ peak, which contributes significantly less to the tail window than the full HVP contribution or the intermediate-distance window. Thus, the fact that our plots of those two quantities display disagreements between results obtained from the measurements of different experiments does not imply that those same disagreements are important to the determination of the tail.

We have updated Section 10 (Long-distance contributions) of the paper to make this suppression of the ρ peak and its effect on our results clear to the reader. We have added two additional figures and associated discussion, clearly illustrating the suppression of the region in which the experiments disagree.

However, the referee is correct that it is a difficult issue to estimate the uncertainty of the data-driven determination of the tail contribution. Since our final result for the HVP contribution to the muon $g - 2$ is largely insensitive to the uncertainty on this tail contribution, it is worth being maximally conservative when performing this estimation. In light of this, and to comprehensively address the referee's concerns, we have added four additional uncertainties to our result, to encompass all reasonable variations to the approach we take here.

- The first of these uncertainties relates to how we average the experiments. We take the full difference between the result when the two-pion spectra are integrated first, then averaged, and the result where the averaging is performed before integration.
- The second error is related to older experimental data sets. In our current approach, it really only makes sense to use datasets that fully cover the energy range that is critical for this tail calculation. In a combination of spectra approach we have added the full effect of removing these older data sets as an additional uncertainty.
- The CMD3 experiment has significantly more precise results than any of the other experiments in this low-energy region. This causes it to have a significant influence on the final result. While it would be extremely difficult to justify removing the most precise and most recent experimental result from the analysis, we conservatively quantify its effect on the final result by taking the full difference with an analysis without it as an additional uncertainty.
- Finally, to ensure there are no residual effects from the discrepancies on the ρ peak, even after the suppression by the integration kernel and quantifying the effect of CMD3, we also take the difference between analyses without KLOE or BaBar, in analyses where CMD3 is also absent, as our final additional uncertainty.

Our value with the updated uncertainty, $27.59(52)$, is 1.5 sigma away from KNT19's $26.82(23)$ that the referee is concerned about. The latter does not include the most-precise CMD3, so it is obviously lower than our value. A discussion about the relevant comparison between DHMZ19 and KNT19 has been added in the paper.

All of these systematic variations are detailed in an updated Section 10 (Long-distance contributions). The net effect is that our central value remains unchanged, but we have increased the final uncertainty on the tail from 0.9% to 1.9%. It is important to note that even if one arbitrarily doubled this new uncertainty, the change in the uncertainty on the main result of the paper would be negligible.

comment r1i03

1) The term 'Strong-isospin breaking' is used in the main text (e.g. line 85) with no description of what it is.

We removed the formulation strong-isospin breaking from the main text, so that we only talk about isospin-breaking in general. In the supplementary material, we have added a description of strong-isospin breaking where it is first mentioned.

comment r1i04

2) Line 69 - there is in fact no discussion of the top quark contribution, so perhaps drop that.

We have dropped this.

comment r1i05

3) Line 180-181 1.5 fm TO 1.9 fm ?

Thank you. This is now corrected.

comment r1i06

4) Line 189 perhaps write $6.5(5.5) \times 10^{-10}$ since the 10^{-10} was reinstated in the few lines above.

Thank you. This is now corrected.

comment r1i07

5) caption to Figure 3 - computed in present -> computed in THE present

Thank you. This is now corrected.

comment r1i08

6) caption to Table 3 horizontal -> vertical

Thank you. This is now corrected.

comment r1i09

7) some discussion of the improvement in the uncertainty on w_0 from the 2020 value would be useful. This comes largely from improvements in the systematic uncertainties. How big a role does the new fine ensemble play here? (This would be helpful in setting the bigger picture of improvements from the new finer ensemble).

We added the following discussion in Section 3 (Scale setting with the pion decay rate):

We also investigated the influence of our new fine ensemble and found that it leads to a substantial reduction in the uncertainty on w_0 . The total relative uncertainty is reduced from 0.40% to 0.23%, with the statistical error improving from 0.19% to 0.11% and the systematic one from 0.35% to 0.20%. The dominant improvement in the systematic error budget comes from better control over discretization effects, in particular the difference between Wilson and Zeuthen flows, which drops from 0.26% to 0.11%. This confirms that the finest lattice plays a central role in improving both the precision and reliability of the continuum extrapolation.

comment r1i10

8) the comment on discussion of improvement over the 2020 result also applies to the intermediate distance window.

We included the following paragraph in Section 6.3 (Windows at intermediate and long distances) to discuss the improvement due to the new ensemble:

In order to quantify the improvement on the uncertainty of $a_{\mu,04-10}^{\text{light}}$ due to the new $a = 0.048$ fm ensemble, we compare the results obtained with and without that ensemble. For consistency, the latter is computed using the distribution of w_0 obtained also without the new ensemble. We obtain the following uncertainties:

$$\begin{aligned} a_{\mu,04-10}^{\text{light}} \quad \text{without new ensemble} &\longrightarrow (0.26)(0.79)[0.84] , \\ a_{\mu,04-10}^{\text{light}} \quad \text{with new ensemble} &\longrightarrow (0.24)(0.59)[0.64] , \end{aligned}$$

implying an error reduction of $1 - 0.64/0.84 = 24\%$.

comment r1i11

9) Table 8 - adding in the result for the 'old' data-driven value for the 1.5 to 1.9 fm window from 2311.09523 to the comparison plot might be a good idea to provide the reader with the 'bigger picture' (since this was done for the intermediate distance window).

We have added the missing data-driven value to the comparison plot (now Table 10).

comment r1i12

10) section 5.3 - references for the taste-improvement procedure should be given. Referring to the 2020 paper for references is irritating for the reader.

We have added the references {Jegerlehner:2011ti}, {Sakurai:1960ju} and {Chakraborty:2016mwy} regarding the SRHO model, and the references {Lee:1999zxa}, {Sharpe:2004is}, {Aubin:2006xv} and {Aubin:2019usy} on staggered XPT, to what is now Section 6.3 (Windows at intermediate and long distance).

comment rli13

11) Beginning of section 5.4 - the time separations of the different windows look a bit arbitrary. It would be good to have a bit more discussion of how these were arrived at. A more natural break would seem to be 1.0fm since that is used for the intermediate window. Instead we have 0.6fm and 1.2fm. Is the 1.2fm break used because of statements in the section before about the time value at which taste-improvement is introduced?

We have extended the discussion on splitting the Euclidean time region in Section 6.4 (Light and disconnected 00 – 28 window) and also included a plot showing how the result and its uncertainty depend on the window splitting.

In choosing the times at which the different windows begin and end, one must make sure they that they do not become too narrow, to avoid lattice artefacts, nor too numerous, to avoid problems associated with strong correlations. We allow the fit functions to be different in each window ... We perform the multi-window analysis for different partitions of the 04 – 08 interval, varying the number of windows and the times at which they are joined (see Figure {fi:win_select}). We then monitor the continuum extrapolation errors. For this study the uncertainty coming from scale-setting is irrelevant, and therefore not included. However, we note that it still is the dominant source of systematic uncertainty in the 00 – 28 window. We observe a significant decrease in the error when using two windows instead of one, the optimal split being at 1.2 fm. Our analysis confirms the recommendation of the Theory Initiative {Aliberti:2025beg} to split the range into intermediate- and long-distance windows. Our optimal splitting time is only slightly larger than their choice of 1.0 fm. We also look at splitting the interval into three windows and find a small improvement with respect to the two window case. As a result we choose the three windows, 04 – 06, 06 – 12 and 12 – 28, for our final analysis. This allows for having simultaneously an a^2 based extrapolation in the first window, a Δ_{KS} based one in the second and an a^2 one with taste improvement in the third.

The 1.2 fm point at which we begin applying taste corrections (which is also accompanied by a 1.4 fm choice) was chosen as the time beyond which the different model-corrections describe the lattice artefacts well. This was investigated in our 2020 paper. As opposed to our 2020 work, we are not using taste corrections below this point, since some of them deteriorate the quality of the continuum extrapolations considerably. In 2020, this was not apparent, due to the lower statistics and the absence of the present finest lattice spacing.

comment rli14

12) Figure 11 needs units (fm) for t^*

We have added the missing units to the plot (now Figure 14).

comment rli15

13) In the discussion of the comparisons for the 2.8-3.5 fm window (around Figure 14) it would be useful to know how much of the 2.8 to infinity contribution this represents.

We have updated the plots to show the values of the unblinded tail results. From this, it is now straightforward to see that the 2.8-3.5 fm window represents about 65% of the one that extends to infinity.

Answers to Referee 2

We thank the referee for their valuable comments, suggestions and criticisms. We have considered all of them and have revised the paper accordingly. The original comments of the referee are reprinted here in blue in the order they were made. After each comment we detail the modifications which we made to our analyses and to the manuscript.

comment r2i00

- If, despite the criticism given above, the hybrid result remains the central (or, heaven forbid, only) result, change the title to "Hybrid determination of the \dots ".

We keep the hybrid result as the central prediction of our paper. Since the average reader will not know what “hybrid” refers to, we would like to keep the title as is. Note that the title does not claim that this is a lattice calculation but simply “a calculation.”

comment r2i01

- Correct mild errors in the Abstract:
| the "contradiction" of α_{mu} is not "recent" but has been with us for around a quarter century.
| CMD-3 did not determine $\alpha_{\text{muLO-HVP}}$ in ref. [4/61] but merely indicated how things might change substituting their data for those of earlier experiments.

We have modified the abstract according to the referee’s recommendation.

comment r2i02

- 1st paragraph main text: QFT leads to additional small corrections to α_{mu} from particles and interactions whether they are included in the Standard Model or not. That’s why the topic is so interesting!

We thank the referee and have changed the sentence in the main part of the text accordingly:

Relativistic quantum field theory introduces additional small corrections induced not only by all particles and interactions of the Standard Model (SM), but also potentially by yet undiscovered ones.

comment r2i03

- "QCD is a generalised version of QED": this phrasing appears in other papers by some of the present authors. The statement is both true and misleading, because the generalisation is a mathematical statement, while the difference in physical phenomena between QCD and QED is drastic (and why LQCD is often mandatory but standalone LQED never is). Thus, it is doubtful that the statement aids a general Nature reader.

We replaced the sentence in the main text as:

Mathematically, QCD is a generalised version of quantum electrodynamics (QED). However, QCD predicts physical phenomena that are drastically different from those described by QED.

comment r2i04

(In this vein, seven out of 14 authors of ref. [3/7] are among the 21 authors of #2024-10-23178. I think this clause is not the only (near) verbatim repeat, which some would say means the 14 new 2024 authors are plagiarising the seven departed 2020 authors. There are also many phrasings of the sort "our earlier work": is that appropriate with the overlap at hand? Although both works will often be attributed to the "BMW" collaboration, neither paper's masthead attributes the work to any collaboration at all. I don't have an opinion on this matter but raise the issue in case the Editors have thought about it and would see a need to apply policy for such circumstances.)

Both papers present work of the BMW collaboration (here also with the DMZ collaboration). As such, we do not think that referring to the earlier paper as "our work" will confuse readers.

comment r2i05

- Fig. 1 axis label: "Finite $L \ \& \ T$ " should be "Finite $L \ \& \ T$ " to conform with the notation in the text.

We have changed the label accordingly.

comment r2i06

- ad (a): "we replaced" should be phrased in the present tense, because for the reader it is happening right now in this manuscript, not some time in the past. Several other places have inconsistent usage of verb tense; please review, as I cannot track all of them.

We have corrected the inconsistent use of verb tense throughout the main paper.

comment r2i07

- Many places, including in ad (a): "data-driven" without referring to e^+e^- (or tau) will confuse readers not steeped in HVP jargon, because the LQCD analysis is also driven by (computer-generated) data.

We have included the following sentence in the main text to clarify this nomenclature:

Here and in the remainder of the paper the expression "data-driven" refers to predictions based on measurements of the hadron spectrum in e^+e^- annihilation and τ -decay experiments.

comment r2i08

- ad (b): NNLO is not defined, though spelling it out might not help many readers.

We have updated the text to spell out NNLO.

comment r2i09

- ad (c): "The first of our windows": it's RBC's window unless one is of the view that it now belongs to everybody.

We have changed the text to avoid calling it our window:

The first window corresponds to the Euclidean-time interval 0.0 to 0.4 fm ...

comment r2i10

- Paragraph starting "For the connected": "ID" is not defined. The notation for errors should be defined here (and possibly repeated in the supplement). Perhaps the first number in parentheses is the statistical uncertainty, the second number in () is the systematic uncertainty, and the number in brackets [] is the quadrature sum.

We have added a definition for ID, and included the explanation for the notation for the errors both in the main text and in the supplementary information.

comment r2i11

- Fig. 2 and surrounding text: for the plots it is necessary to have short labels, but the text and/or caption should make clear that "RBC '23" is an abbreviation for RBC-UKQCD [32/36], "Lehner '20" for Lehner and Meyer [33/40], "FHM" for Fermilab Lattice, HPQCD, and MILC, "Benton" for Benton et al., etc.

We have included the clarifications and the references in the caption.

comment r2i12

- Fig. 3: This figure is fine---don't change a thing. That said, I alert the Editors that the lower half shows what a mess the "data-driven" scenario has become, which undermines using it (as discussed above) without stronger arguments than those given in the main text or in Section 9 of the supplement.

The referee is right, the lower part shows "what a mess the data-driven scenario has become" if one wants to use it for the total $a_\mu^{\text{LO-HVP}}$. We completely agree with the referee and added the following remark to the main text:

As the figure shows, using the data-driven approach for determining the total $a_\mu^{\text{LO-HVP}}$ leads to serious tensions.

Fortunately, the data-driven method works very well for the tail. While the tensions in the total $a_\mu^{\text{LO-HVP}}$ come predominantly from the ρ -peak region, the tail observable is largely dominated by smaller center-of-mass energies where experiments agree. As a result, the same four experiments that are shown in Fig. 3 of the Main text give completely consistent results for the tail (see e.g. Fig. 10 of the Supplementary Information). This and related issues are further discussed in an expanded Section 10 (Long-distance contributions).

Also, we have updated Fig. 3 to reflect tiny changes in our result for $a_\mu^{\text{LO-HVP}}$ (prompted by suggestions from the referees) and to account for new lattice results and the Muon $g-2$ Theory Initiative's 2025 White Paper compilation.

comment r2i13

- The initials in the acknowledgments and author contributions should match the author list, so KKS and BCT for K.K. Szabo and B.C. Toth.

We have fixed the initials.

comment r2i14

- Fig. 1 (supplement): the meaning of the error bar on the physical point is not defined. The numbers in () for the beta values are inscrutable.

We have adjusted the caption to make it more clear that “the uncertainties of this point” refers to the uncertainties indicated by the error bars the referee asks about:

The horizontal and vertical axes are the squared pseudo-scalar masses, M_{ll}^2 and M_{ss}^2 , in units of the w_0 -scale, both normalized to the central value of their respective physical point... The black point denotes the (isospin-symmetric) physical point, with error bars corresponding to the uncertainties from our determination of the M_{ll} , M_{ss} and w_0 parameters in physical units.

The numbers in () listed the number of ensembles at each beta value. This was redundant with Table 1 of the Supplementary Information, so we have removed them.

comment r2i15

- FLAG is cited for mc/ms but not the underlying work, which is a violation of FLAG policy. Please cite all papers entering the average. (Some authors are FLAG members and should know better.)

We have included the citations for the papers used in constructing the FLAG average.

comment r2i16

- It is not clear whether the numbers for ms/ml are exact or are rounded from ams divided by an exact aml.

We added the following sentence to the table caption to clarify this:

The numbers are rounded to the accuracy provided by the number of displayed digits.

comment r2i17

- Bottom of p. 6: "It is usually assumed that these [hairpin] artefacts decrease\ldots" but what a reader needs to know is not (just) the usual assumption but the assumption made in the work at hand.

We modified the sentence to make our assumption clear:

Here we assume that these artefacts decrease with the same rate as $\Delta_{KS}(\xi)$; the same assumption was made in Ref. {FermilabLattice:2018zqv}.

comment r2i18

- Section 2, Omega baryon computations: | sloppiness in plural/singular: "In the continuum limit the masses of these states become degenerate." Also, there may be only one smearing parameter sigma. | more seriously, too little information on the continuum limit and strange retuning is provided. A revision should state how any readjustments of the strange quark mass propagate to uncertainty in M_Ω . In ref. [3/7], the M_Ω continuum extrapolation was slightly controversial---in response to community comments, it changed from arXiv-v1 to arXiv-v2. A revision must include a plot of the current status, not least to see how the new finer ensemble improves the extrapolation. Many people share this reservation about the current presentation. | perhaps the requested information is actually already in Table 3 of Section 3? If so, the plots in Table 3 would convey this information by presenting the extrapolation as w_Ω instead of w_0 in attometres.

We have fixed the typos.

We are not using any strange retuning. We have several ensembles with different strange masses, sometimes even at a single lattice spacing, which enable us to interpolate to the physical strange mass. We also take into account lattice artefacts in the strange mass dependence of our observables.

Regarding the improvement due to the finest ensemble we added the following into Section 3.5 (Determination of w_0 using the pion decay rate - results):

We also investigated the influence of our new fine ensemble and found that it leads to a substantial reduction in the uncertainty on w_0 . The total relative uncertainty is reduced from 0.40% to 0.23%, with the statistical error improving from 0.19% to 0.11%, and the systematic error from 0.35% to 0.20%. The dominant improvement in the systematic error budget comes from a better control of discretization effects, in particular those related to differences between Wilson and Zeuthen flows, which drop from 0.26% to 0.11%. This confirms that the finest lattice plays a central role in improving both the precision and reliability of the continuum extrapolation.

Regarding the w_0 plot we added the following sentence in Section 2.3 (Determination of w_0 using the Omega mass):

The w_0 values in the table are obtained by dividing the $w_0 M_\Omega$ product by the experimental value of the Ω baryon mass.

comment r2i19

- The beginning of Section 3 muddles a bit the distinction between the simulation meson masses and their physical values.

This section, Section 4 (Physical point and isospin decomposition), has been rewritten to include our new scale setting based on the pion decay rate and the comment of the referee was taken into account.

comment r2i20

- The definition of the lattice spacing is not made clear. If I've understood,

on a given ensemble $a=[a/w_0]GF[w_0]phys$ with $[w_0]phys$ from eq. (15). Further, I think in eq. (15) $[w_0]phys = [M_{\Omega-1}]PDG \lim_{a \rightarrow 0} ([aM_{\Omega}]cor.fit[a/w_0]GF)$ but it is never stated this way. (GF stands for gradient flow (Wilson had nothing to do with it.), and "cor.fit" for correlator fits from Section 2.)

We have clarified the definition of w_0 in Section 2.3 (Determination of w_0 using the Omega mass):

The physical value of $[w_0M_{\Omega}]_{qcd+qed}$ is obtained by substituting the physical values of the M_{ud} , M_{us} , M_{ds} and M_{Ω} in the above formula. Then dividing by the value $M_{\Omega} = 1672.45 MeV$ yields the physical value of the gradient flow scale $[w_0]_{qcd+qed}$.

Additionally, we have added the following sentence to define the lattice spacing to Section 4.1 (Physical point):

That is, on each ensemble the lattice spacing a is defined as the physical value of $[w_0]_{qcd+qed}$ from Equation (`{eq:phys3}`) divided by the value of $[w_0/a]$ measured on the given ensemble.

comment r2i21

- Caption for PDF plot in Table 3: like in similar tables, the median is presumably given by the blue vertical line. The caption states "horizontal".

We have corrected this. We have also added a paragraph to the beginning of Section 6 (Window observables) that describes all "Tables" in that section.

comment r2i22

- Captions for PDF plots in Tables 3, 4 state the 1-sigma/2-sigma band is formed at the 16th/2th and 84th/98th percentiles while the plot shows 68.4%/95.4%. It takes a while to understand how they are compatible, possibly because using % and the word "percentile" is at best redundant and probably actually incorrect. It would be better for the caption to use words closer to the content of the plot, perhaps "the 1-sigma and 2-sigma bands contain 68.3% and 95.4% of the distribution, centred at the median."

We have reformulated the text accordingly. We have added a paragraph to the beginning of Section 6 (Window observables) describing all our "Tables".

comment r2i23

- The work of ref. [30] raises the possibility that the anomalous dimension implies a power of α_s with $n < 0$ and worries that this behaviour might not be visible at current lattice spacings. The analysis should incorporate fits with $n < 0$, for example $n = -3$ as studied in some LQCD papers [37/39].

Even the authors of the cited reference consider $n = -3$ an absolutely unlikely scenario. We consider $n = 0$ and $\Delta_{KS}(a)$ dependencies, the latter of which accounts for pion-related taste-breaking effects in contributions dominated by pions, and corresponds to approximately $n = 3$ over a wide range of our lattice spacings.

comment r2i24

- Ref. [31] and follow-on works by Neil and Sitison advocate a Bayesian Akaike information criterion (BAIC) with a more severe penalty for fits that include fewer data. The submission could be improved by providing a quick reminder why, in the statistical framework adopted here, the AIC of eq. (19) (attributed to ref. [3/7]) is more appropriate than the BAIC. If the reasons are not strong, it would be good to state whether any significant change arises between the two criteria. Also, consider citing Akaike's work.

We added the following sentences to Section 5.2 (Distribution of observables) for clarification:

The AIC criterion of {Jay:2020jkz} is derived by assuming that removing a data point is equivalent to adding to the model a parameter that fits the data point exactly. Our AIC criterion is derived by directly computing the dependence of the Kullback–Leibler divergence {Kullback:1951zyt} on the number of data points entering the analysis.

We also added the references {Akaike:1974vps}, {Akaike1973} and {Akaike1978c} on Akaike's Information Criterion.

comment r2i25

- Tables 3, 4, 5, 6, 7 and 8: more information (in the text) describing the corpus of fits is necessary. How are the p values distributed? Which fits---for example which functional forms for the continuum extrapolation---are most highly weighted? When the PDF has structure, as almost all do, what features predominate fits in the various humps? (This report is certainly not the first time these questions have been raised.) How large are the dimensionful coefficients of the discretisation effects?

To many of our continuum extrapolation plots we added a color coding scheme: fits with higher/lower AIC values are now shown with a lighter/darker color. In particular, in the w_0 extrapolation we have dropped fits with very low weights, those which were missing some of the data points by far.

In the case of several histograms with apparent multi-peak structure we have added explanations in the text.

For each analysis in Sections 2, 3, and 6 we have included the percentage of fits with P -value larger than 0.1.

For all the continuum limit plots in Sections 2, 3, and 6 we have added examples of the relative contributions from each dimensionful coefficient of the discretization effects at the lattice spacings we work at.

comment r2i26

- Can the improvement in continuum extrapolations from the new $a=0.048$ fm ensemble be quantified? Was it in retrospect a good investment of computer time? Would it have been better to take some of the time and devote it to higher statistics on coarser lattices? In particular to address the t/fm in $(1.0, \text{inf})$ window in LQCD without resorting to troublesome experimental data?

We have included a paragraph to Section 6.4 (Light and disconnected 00 – 28 window) to quantify the impact of the new $a = 0.048$ fm ensemble:

Now we would like to quantify the improvement on our uncertainty resulting from including the ensemble at 0.048 fm, introduced in this work, and the additional error reduction arising from adding the tail from the data-driven approach. For this purpose we compare the uncertainties on the quantity $a_{\mu,04-40}^{\text{light}}$, obtained with and without the new lattice spacing, and the quantity $a_{\mu,04-28}^{\text{light}}$ obtained with the new lattice spacing. For consistency, the value without the new ensemble is obtained using the distribution of w_0 obtained also without the new ensemble. In order to have a fair comparison, we perform the exact same analysis in all three cases, fitting the interval without splitting into more windows. Note that since we now also include cubic fits in our analysis, this procedure is more conservative than that of our previous work {Borsanyi:2020mff}. For the uncertainties of these quantities we obtain

$$\begin{aligned} a_{\mu,04-40}^{\text{light}} \quad \text{without new ensemble} &\longrightarrow (2.10)(5.80)[6.17] , \\ a_{\mu,04-40}^{\text{light}} \quad \text{with new ensemble} &\longrightarrow (1.80)(3.46)[3.90] , \\ a_{\mu,04-28}^{\text{light}} \quad \text{with new ensemble} &\longrightarrow (0.96)(2.55)[2.72] . \end{aligned}$$

Then one can conclude that the inclusion of the new lattice spacing reduces the error by $1 - 3.90/6.17 = 37\%$, and including the tail from the data-driven approach gives a further reduction of $1 - 2.72/3.90 = 30\%$.

comment r2i27

- Table 6 does not include a line for taste-breaking effects, and the text is very terse. Given the strategy used elsewhere in this paper, shouldn't fits with and without taste-breaking corrections be included, with the AIC weighting them accordingly?

As opposed to our 2020/21 work we are not using taste corrections below 1.2 fm, since some of them deteriorate the quality of the continuum extrapolations considerably. With less statistics and without the finest lattice this was not that apparent in 2020/21.

comment r2i28

- Tables 7 and 8 refer the reader to Table 6 for the definition of symbols, etc., but Table 6 sends the reader to Table 4. An unnecessary annoyance.

We have removed these double-hop references.

comment r2i29

- The "SRHO model [used] to get the central value" of the Aubin et al. window, t/fm in (0.15,0.19). Given the general strategy of this paper, shouldn't fits with and without taste-breaking corrections be included, with the AIC weighting them accordingly?

Note, in the window fits we are flat weighting the taste corrections, NNLO vs SRHO. The AIC weighting is applied for the fit form, beta cuts, and also for the a^2 vs Δ_{KS} choice.

Long distance windows are heavily affected by taste violation. We know, that the taste splitting in the pion multiplet cannot be extrapolated to zero with a series starting with a^2 , since it falls more like a^4 . The result is that an a^2 extrapolation of the pion taste splitting undershoots zero, and consequently an unimproved a^2 extrapolation of a long distance window overshoots the continuum value. This motivates us to always use taste corrections in long distance windows. (We have maintained this choice from our previous work.)

Unimproved data could be used if, when no taste improvement is applied, we allow only fits in Δ_{KS} and not a^2 . We have made some tests on the long distance window 15 – 19 with uncorrected fits. Numbers are median(stat)(sys-w0)(sys-taste-violation)(sys-fit-form)[total]:

95.43(0.82)(0.67)(0.40)(0.63)[1.58] original
 95.71(0.95)(0.68)(1.13)(0.83)[2.33] original + unimproved
 95.42(0.82)(0.67)(0.42)(0.56)[1.54] original + unimproved with only Δ_{KS}

Not unexpectedly, adding unimproved fits increases the errors. Interestingly the central value hardly moves upwards. Still we think, that unimproved fits with a^2 are introducing an unwanted bias. Removing them from the analysis and restricting to Δ_{KS} for fits without correction, gives a result/error that is almost identical to the ones from the analysis in the manuscript.

comment r2i30

- Section 6: the notations 4hex, N3L0, N4L0 and N4L0 are not defined. The last two presumably mean the same thing. 4hex could be elegantly introduced after the words "utilized a new staggered action", I suppose. Sometimes the superscripts are I=1, I=0 instead of using the customary maths-italic I.

We defined the abbreviations and fixed the superscripts in what is now Section 7 (Finite-size effects).

comment r2i31

- A strength of this work is the estimate of finite-size effects with dedicated simulations. Even so, it would be worth calling out in the text that NNLO for $L_{\text{ref}} \rightarrow L_{\text{big}}$ is consistent with the 4hex calculation. As it stands, the reader has to mine Tables 13 and 14 for this piece of information.

We added the following text into Section 7 (Finite-size effects):

We can also compare the the $L_{\text{ref}} \rightarrow L_{\text{big}}$ finite-size effect from the 4hex simulations with the prediction of XPT, the corresponding numbers can be found in Table {ta:fv_xpt}, both for NLO and NNLO XPT. We find a good agreement between the values of the 4hex simulations and those of NNLO XPT.

comment r2i32

- Section 7 is written for deeply knowledgeable experts. It is very terse. Perhaps that is acceptable, but the papers that developed low-mode averaging should be cited.

We added the references {Neff:2001zr} and {Li:2010pw} about LMA in what is now Section 9 (Verification of isospin-breaking contributions).

comment r2i33

- Section 8 is an alternative estimate of finite-volume (aka finite-size) effects. It would be better to integrate the discussion with Section 6---at least don't have an intervening section. A table summarising the $L_{\text{ref}} \rightarrow \text{inf}$ corrections for NNLO, 4hex+NNLO, data-driven, and final (i.e., cyan band of fig. 12) would be helpful: information now in Section 6 could perhaps be added to Table 17. By the way, Table 17 is not explained in the text, although it is used in Table 11.

We have moved the two finite volume sections next to each other.

We have prepared a new summary table of finite volume effects to compare NNLO, 4hex+NNLO and data-driven approaches, in Section 8 (Data-driven determination of finite-volume corrections) and described it in the text as follows:

In Table .. we compare finite-volume effects as computed in the data-driven approach in this section to those computed in NNLO XPT and in the 4hex simulations, as described in the previous section. We find a good agreement between the various approaches. In our final results the bulk of the finite-size correction is taken from the 4hex simulations, and for a residual finite-size effect we use NNLO XPT.

The reference to Table 17 is meant as a reference in another paper.

comment r2i34

- In Section 8, it is hard to follow the connection between the words and the width of the cyan bands in fig. 12. Perhaps paragraph explaining Table 17 and, hence, the cyan bands was dropped in editing.

We have changed the bands on Figures 11 and 12 in Section 8 (Data-driven determination of the finite-volume corrections) and now we explain them both in the captions and in the text.

comment r2i35

- Using e+e- data for finite-volume corrections is subject to the same scepticism as for t/fm in (2.8,inf). As in Section 9, not all available experiments are used.

We added the following clarification to the beginning of Section 8 (Data-driven determination of finite-volume corrections):

Using a data-driven approach, in this section we provide an alternative evaluation of the finite-volume corrections to the various isovector window observables considered in this paper. This serves as a cross-check for the corresponding lattice calculation based on the 4hex ensembles, and despite being an interesting result on its own, it is not used in our final evaluation. We generically denote these finite-volume corrections as $\Delta a_{\mu}^{I=1}(L_{\text{ref}} \rightarrow \infty)$.

Also we added a paragraph on the impact of tensions in the input data and why we are not considering all experiments:

With this procedure, before including the various other systematic uncertainties described above, the finite-volume correction determined from CMD-3 data is not compatible with the one coming

from fits to BaBar and KLOE data. This is because the phase shift, the input of this calculation, is obtained by global fits that extend up to ~ 1 GeV, thus including the region dominated by the ρ peak, where discrepancies between experiments are observed. As we will see later, this effect is way smaller than the other systematics of the calculation. The same applies for including experiments, other than the above three: they would not significantly impact the final result and error for the finite-volume effect.

comment r2i36

- Both bibliographies need some hand-editing w.r.t. \bibtex code from \textsc{inSPIRE}. For example, more recent papers' authors' names include diacritics (e.g., Cè) while from older papers they are left out (e.g., Luscher instead of Lüscher). Some math/symbol material is spelled out in ugly notation (compare $\pi+\pi-(\gamma)$ in refs. [59, 60]).

We thank the referee for pointing this out. We have hand-edited the bibliography and unified the usage of diacritics.

comment r2i37

As discussed above, Section 9 is key to a decision whether to publish #2024-10-23178 or not. The "data-driven" estimate of t/fm in (2.8,inf) reduces the uncertainty in $a_{\mu}^{\text{LO-HVP}}$ hybrid determination compared with (my guess of the not provided) LQCD-only calculation.

The referee is absolutely right, the data driven determination of the tail reduces the uncertainty on the total $a_{\mu}^{\text{LO-HVP}}$. This is the reason that we have used it. In that way we can provide the community with the most precise result for $a_{\mu}^{\text{LO-HVP}}$ to date.

The text actually summarises well why one would not trust the $e+e-$ (or tau) data. The three-bullet justification has the following weaknesses:

- four experiments agree: agreement is no guarantee of reliability;

Of course, this general comment is logically true and we added this sentence to the text in Section 10 (Long-distance contributions). This is nevertheless a good indication of reliability in the experimental world, especially when employing very different complementary approaches ($e+e-$ scan, ISR high energy, ISR low energy, tau decays). This is further supported by the fact that including all available experiments would change our result by a fraction of our uncertainty (see also the last bullet point).

- compatibility with LQCD: compatibility for the full tail window is not shown---fig. 14 shows only t/fm in (2.8,3.5); fig. 14 only validates this window to the 8% level;

The referee is absolutely right, it is a validation to the 8% level. We added this sentence to the text in Section 10 (Long-distance contributions) and emphasized that this somewhat disadvantageous 8% uncertainty is the reason why we have used the data-driven approach to compute the tail.

- impact on total uncertainty is negligible: which means it disappears in a quadrature sum; yet if the uncertainty is merely at the 10% level (based on compatibility with LQCD), then that is not true; moreover, although it is

stated "all" e^+e^- experiments agree, only three are actually included. I estimate that including them would reduce the final result by 1 or 2 units (10-10), which is a bit smaller than the final uncertainty estimate, but not at all small compared with the uncertainty on $a_{\mu,28-\text{inf}}$ in eq. (50) of the supplement (27.59±0.26).

It is indeed important to study the effect of including all available measurements of the $e^+e^- \rightarrow \pi^+\pi^-$. We have now done so in Section 10 (Long-distance observables) and observe a shift of 0.16×10^{-10} , which is far less than the estimate of the referee. We added this shift to our error budget.

comment r2i38

The reasons given for choosing 2.8 fm versus 2.6 fm or 3.5 fm seem arbitrary.

We have not performed a systematic optimization of the tail's starting time. We simply require that the known experimental tensions on the ρ peak have a tiny impact on our final result for $a_{\mu}^{\text{LO-HVP}}$. A $28 - \infty$ tail satisfies that criterion.

comment r2i39

Finally, it is hard to understand the upside from taking this risk. Attached is a comparison of the 2020 result [3/7] with the present claim (inner/outer error bars are reported statistical and total uncertainties). The red (top) point is my estimate if, as inferred from fig. 14, the uncertainty on $a_{\mu,28-\text{inf}}$ is 10% of the value given in eq. (50). It is still an advance over 2020.

In summary, the case for publishing is the reduction in uncertainty shown in the figure. The case against is the risk entailed in trusting the treatment of radiative corrections of e^+e^- data and discarding the criticisms of the tau method that led to its deprecation. The risk could be mitigated by providing LQCD only results for (1.0,inf), (2.8,inf) and the full $a_{\mu}^{\text{LO-HVP}}$. Of course, if my guess of 10% uncertainty on (2.8,inf) is optimistic, then the LQCD-only result is not interesting, and it seems the large computing resource invested in the $a=0.048$ fm ensemble could have been deployed more wisely.

This paper is a hybrid computation. We have now provided all windows, as requested by the Theory Initiative, within the hybrid approach.

Gaining full control over the light, disconnected, isospin breaking, and finite volume contributions to the tail $28 - \infty$ using only lattice QCD is a significant challenge, and will probably take several years of community effort to achieve. At this point in time, it is reasonable to trust independent datasets which agree very well in the low-mass region relevant for the tail, despite possible issues concerning radiative corrections in the e^+e^- that could explain, at least partially, tensions observed in the higher-mass region around the ρ peak. As explained in Section 10 (Long-distance contributions), those tensions have a small effect on the tail that is covered by the uncertainties estimated in that section which, in any case, have negligible impact on our final result for $a_{\mu}^{\text{LO-HVP}}$.

Regarding the τ , we carried out an analysis with the τ data-set and another one without it. The result we choose to keep includes the τ but we add to it the full difference that its central value has with respect to that of the result which omits it. As a result, the value of the former is 27.59(26), to be compared with the 27.68(19) of the latter. In our conservative approach, the result we retain has, in fact, a larger uncertainty than the e^+e^- only average.

Answers to Referee 3

We thank the referee for their valuable comments, suggestions and criticisms. We have accepted all of them and have revised the paper accordingly. The original comments of the referee are reprinted here, in the order that they were made, with blue. After each comment we detail the modifications which we made to our analyses and to the manuscript.

comment r3i00

While the results are of high quality and also of high relevance for the search for new physics, I cannot help but noticing a certain lack of transparency regarding the presentation. This concerns the material per se, as well as the ability to perform comparisons with results obtained by other collaborations. Given the high interest in the muon anomaly in the entire particle physics community, I think this is a shortcoming that should be rectified in a revised version of the manuscript. I elaborate on this in section F of this report.

We have included the missing continuum limit plots and error budgets into Section 6 (Window observables). Additionally, we have computed the total value for the long distance window $10 - \infty$.

comment r3i01

To judge the robustness and validity of the conclusions, additional information should be provided by the authors. This concerns, in particular, the addition of scaling plots for the isovector contribution in the range from 1.0-2.8 fm, an extended discussion of the difference with their previous calculation and additional comments on the validity of taking over the calculation of isospin breaking effects.

We have included the continuum limit plots and the error budgets for the light contribution of the $10 - 28$ window into Section 6.3 (Windows at intermediate and long distances). Additionally, we have computed the all flavor result for the $10 - \infty$ long distance window.

We replaced the previously used isospin breaking contributions with new values, computed in the required Euclidean window $00 - 28$, as explained now in Section 6.6 (Other contributions and total).

comment r3i02

1. The authors state on p6 that the added finer lattice spacing reduces the uncertainty in the light connected contribution by 37%. Does this reduction refer to the entire integration range or to the one-sided window from 0-2.8 fm? If it is the latter, this statement cannot be verified by the reader since there is no result for the one-sided window in the earlier publication.

We have included a paragraph to Section 6.4 (Light and disconnected $00 - 28$ window) to clarify the impact of the new $a = 0.048$ fm ensemble:

Now we would like to quantify the improvement on our uncertainty resulting from including the ensemble at 0.048 fm, introduced in this work, and the additional error reduction arising from adding the tail from the data-driven approach. For this purpose we compare the uncertainties on the quantity $a_{\mu,04-40}^{\text{light}}$, obtained with and without the new lattice spacing, and the quantity $a_{\mu,04-28}^{\text{light}}$ obtained with the new lattice spacing. For consistency, the value without the new ensemble is obtained using the distribution of w_0 obtained also without the new ensemble. In order to have a fair comparison, we perform the exact same analysis in all three cases, fitting the interval without splitting into more windows. Note that since we now also include cubic fits in our analysis, this procedure is more conservative than that of our previous work {Borsanyi:2020mff}. For the uncertainties of these quantities we obtain

$$\begin{aligned} a_{\mu,04-40}^{\text{light}} \quad \text{without new ensemble} &\longrightarrow (2.10)(5.80)[6.17] , \\ a_{\mu,04-40}^{\text{light}} \quad \text{with new ensemble} &\longrightarrow (1.80)(3.46)[3.90] , \\ a_{\mu,04-28}^{\text{light}} \quad \text{with new ensemble} &\longrightarrow (0.96)(2.55)[2.72] . \end{aligned}$$

Then one can conclude that the inclusion of the new lattice spacing reduces the error by $1 - 3.90/6.17 = 37\%$, and including the tail from the data-driven approach gives a further reduction of $1 - 2.72/3.90 = 30\%$.

comment r3i03

2. In the penultimate paragraph on p8 the authors state that their new result is larger by 6.5(5.5) units of 10-10. In order to improve transparency, this should be supplemented by a discussion which part(s) in their updated analysis is/are responsible for the upward shift (there is a similar request relating to p29, see comment 11 below).

See issue r3i12.

comment r3i04

Supplementary Material:

3. The calculation of the Omega mass described in section 2 is "based on the GEVP". The procedure that is employed by the authors is known as the "generalized pencil-of-function" (GPoF), and I suggest that, for clarity, the authors change their statement that their procedure is "related to the GEVP" and mention the GPoF as well.

We added a reference to GPoF, as requested.

comment r3i05

4. When arguing for the stability of their results for the Omega baryon mass in Fig. 6, the authors provide no actual values. I suggest that the authors show the value of the ensemble average by adding a separate scale on the right-hand abscissa.

We updated the Figure (now Figure 7) to show the value of the ensemble average in GeV.

comment r3i06

5. Top figure of Table 3: why do the data points appear slightly shifted along the y-axis between the plots of the left- and right-hand sides?

This is a consequence of shifting the plotted points to the physical quark masses. Since our points scatter around the physical quark masses, in order to compare the data to the fit curves, they need to be evaluated at the same values of the masses in order to be compared. To resolve this, we use one of the fits from the plot to shift the values to the physical point. Because the plots have different fits, the quark-mass dependence in the fits is slightly different and the points are shifted by slightly different amounts.

We added an explanation of this to the start of Section 6 (Window observables), where the plots are generally described:

For readability, the points are projected to the physical quark masses using an appropriate fit from the plot. This can result in small differences between the apparent values in the left and right panel.

comment r3i07

6. In section 4.1. the authors describe their motivation for the fit ansatz of the continuum extrapolation. To this end, they use either polynomials in a^2 or in Δ_{KS} but no combination of the two. They should elaborate on why they chose to avoid an ansatz mixing both terms.

In order to quantify the presence of mixed fits in the multi-window analysis of the 04 – 28 window, we added fits like $a^2 + \Delta_{KS}$, $a^2 + \Delta_{KS} + \Delta_{KS}^2$. We exclude mixed fits like $a^2 + a^4 + \Delta_{KS}$, since a^4 is very much like Δ_{KS} on our lattices, leading to unstable fits. We obtain the following results, numbers are median(stat)(sys-w0)(sys-taste-violation)(sys-fit-form)[total]:

$$\begin{aligned} 572.11(0.96)(2.25)(0.50)(0.87)[2.76] & \quad 3\text{-win} \\ 572.12(1.08)(2.24)(0.57)(0.86)[2.84] & \quad 3\text{-win} + \text{mixed fits} \end{aligned}$$

Note, these numbers still use the omega based scale setting. We see, that the mixed fits do not change the central value at all and increase the total error by only 3%. This increase should be well within the uncertainty of our error estimate (aka the error-on-error).

In the new w_0 analysis, which uses the decay rate as input, the mixed fits are automatically included and weighted with the AIC together with fits with no mixed terms.

comment r3i08

7. In footnote 4 of the manuscript, the authors appeal to a new paper by RBC/UKQCD (which contains an estimate for the so-called long-distance window from 1.0 fm to infinity) as evidence supporting a larger value for the HVP contribution compared to their 2020/21 publication. It would be much more convincing if the authors included their own estimate for the long-distance window, to allow for a like-by-like comparison between their calculation and those of other groups. This would not only improve the transparency of the presentation but also give the authors the chance to provide more details as to which part of their calculation leads to the larger estimate (see comment 2 above).

We now give the value of the all-flavor long distance window $10 - \infty$. This was obtained using our lattice results for the $10 - 28$ region and using the data-driven approach for the rest $28 - \infty$.

comment r3i09

8. The updated estimate for the disconnected window observable presented in Table 7 differs strongly from the 2020/21 result, yet this is not discussed in the manuscript. At the same time, the two peaks in the PDF are clearly inconsistent. Judging from the two scaling plots at the top, the two-peak structure arises from the two alternative fit forms, using either polynomials in a^2 or in Δ_{KS} . This may be an example where the restriction to polynomials in either variable but not allowing for mixed terms is too restrictive. Eventually the authors leave it to the AIC to select the most likely value, but this procedure may be biased if the input is too selective. I also wonder whether the addition of a finer lattice spacing is responsible for the smaller estimate for the disconnected contribution relative to the 2020/21 publication, at least when only polynomials in a^2 are considered. I recommend that the authors comment on this issue and expand the discussion.

We added the following clarifications into the text in Section 6.3 (Windows at intermediate and long distances):

Table {ta.res.di_04_10} shows the analysis details. The two peak structure in the probability distribution arises from the systematic variation between a^2 and Δ_{KS} type fits. We also provide a comparison to the results of ... The difference compared to the value of our 2020 work comes from the addition of a new lattice spacing and that now we also include quadratic fits in the analysis - previously we only used linear continuum extrapolations.

comment r3i10

9. The authors present the isoscalar contribution to the 04-28 window observable in Table 9. They should also include a similar table for the much more relevant isovector contribution. In fact, I am puzzled why the manuscript does not contain this information, which is by far the most crucial part of the lattice calculation. By including the corresponding scaling plots, PDF distribution and table, the authors will have the chance to address the lack of transparency which I have alluded to earlier. It would also be instructive to show individual scaling plots for the three windows 04-06, 06-12 and 12-28. I understand that a simultaneous fit is performed to all three, with individually chosen fit functions. Nevertheless, one can easily reconstruct and plot the fit curves for the individual windows. This would allow the reader to assess the influence of taste violations for the three "sub-windows" and how these are treated in different kinematical regimes.

We have included the continuum limit plots and the error budgets for the three separate windows (along with the distribution and error budget for the combination) in Section 6.4 (Light and disconnected $00 - 28$ window).

comment r3i11

10. Compared to the 2020/21 publication, the errors quoted for the strange and charm contributions listed in Table 10 are larger by a factor 2 and 4, respectively. The authors should provide an explanation what led to the larger uncertainties.

Our previous results were obtained in 2020 for the strange and even earlier, in 2017, for the charm. Now we include in the continuum extrapolation logarithmic lattice artefacts, whose rôle was only explained after our 2020 work was published. Also now we include quadratic extrapolations, previously it was only linear (and without logarithms). We consider the new analysis superior to our previous work and the error estimates are more robust now.

comment r3i12

11. The discussion of the overall difference with the 2020/21 result on p29 is only focused on the estimation of the uncertainty attributed to the shift, but makes no attempt to trace the origin of the difference itself. I am also not convinced that one may treat the errors of the continuum extrapolation as uncorrelated - even if the observable is divided into several sub-windows - since the vast majority of gauge ensembles enters both the 2020/21 calculation and the 2024 update. At any rate, the difference with the earlier result is most likely due to the long-distance window, since the intermediate-distance window is consistent with the 2020/21 calculation, and a shift of about 6 units in the short-distance window would represent a 10% effect in that quantity, which is an unlikely scenario. This makes it even more important that the long-distance window is included and appropriately discussed in the manuscript.

We consider the continuum extrapolations uncorrelated, since

1. we use more sophisticated fit functions - different functions in different intervals - than in 2021, where we used the same fit function in all the interval
2. in 2021 we performed the scale setting and the window continuum extrapolation in one go - now it is split into two steps. The two continuum extrapolations differ by the lattice artefacts of the $w_0 M_\Omega$ product, which is a highly non-trivial extrapolation. (On top of that, the current version uses f_π scale setting instead of M_Ω .)
3. our observable is different, we removed the long-distance part of the correlator that is strongly dominated by taste violation.

As suggested by the referee, we included the long-distance window into Section 6.3 (Windows at intermediate and long distances) with details on the analysis, additional plots and error budget.

comment r3i13

12. As outlined in section 7, isospin-breaking contributions are taken over from the 2020/21 publication, except for the re-tuning of time cuts and fixing the value of the one-photon-reducible contribution. Given that the time cuts for the light-quark sea-QED contributions are shifted to larger values (2.2 and 2.7 fm, respectively, which corresponds to a factor of two), I find it very surprising that the statistical errors change only marginally. Furthermore, given that a hybrid approach is used, I would

expect that the evaluation of isospin-breaking corrections should be restricted to the 0-2.8 fm window, to avoid double counting in the long-distance tail. If time cuts as large as 2.7 fm are employed, I wonder whether this has been implemented correctly. The authors should elaborate on the estimation of the uncertainties of the various isospin-breaking contributions and add a discussion on the evaluation of isospin-breaking corrections in the 0-2.8 fm window.

In order to get rid of the double counting, we replaced the previously used isospin breaking contributions with new values, fitted in the Euclidean window $00 - 28$, this is explained in Section 6.6 (Other contributions and total) in the following paragraph:

In our 2020 work we computed isospin-breaking contributions for the total a_μ . In a previous version of this paper we took over the isospin-breaking contributions from our 2020 work as they are, instead of computing them in the $00 - 28$ window. We now remove this shortcoming. For this purpose we repeat the very same analysis as in our 2020 work for isospin breaking in the $a_{\mu,00-28}$ window, in the cases of the light, disconnected and strange flavors. These are called “Type-II” fits and are detailed in Section 24 of the 2020 paper {Borsanyi:2020mff}. The results are reported in Table {ta:amu}. It turns out, that the numbers obtained in 2020 for the total range and the numbers obtained now for the $00 - 28$ window are in good agreement with each other. However, we consider the errors of the $00 - 28$ window quantities more reliable, since the long distance part can be very challenging to estimate.

comment r3i14

13. In the right-hand panel of Fig. 11, I presume that the discrepancies between the evaluations of finite-size effects based on the CMD-3, BaBar and KLOE data sets reflects the well-known tensions among the evaluation of the two-pion contribution via the dispersive method. If the authors agree, they could insert a comment to this effect.

Yes, we agree and added the following comment on this effect into Section 8 (Data-driven determination of finite-volume corrections) at the end of the paragraph describing the uncertainties propagated from the fits to the experiments:

With this procedure, before including the various other systematic uncertainties described above, the finite-volume correction determined from CMD-3 data is not compatible with the one coming from fits to BaBar and KLOE data. This is because the phase shift (the input to this calculation) is obtained by global fits that extend up to ~ 1 GeV, thus including the region dominated by the ρ peak, where discrepancies between experiments are observed. As we will see later, this effect is much smaller than the other systematics of the calculation. The same applies for including experiments, other than the above three: they would not significantly impact the final result and error for the finite-volume effect.

comment r3i15

14. Regarding Figs. 13 and 14, it would be much more transparent if the absolute values were plotted rather than just the difference from the average (which can be shown separately as a band in the plots).

We have updated the plots to show the absolute values for the unblinded tail results.

comment r3i16

The discussion of the tension between data-driven and lattice estimates for the intermediate window observable on p39 lists only one lattice calculation (Ref. [39]) when there are many more such results with equal or even better precision. This should be corrected.

We corrected it by adding those results into Section 10 (Long-distance window).

Answers to Referee 4

We thank the referee for their valuable comments, suggestions and criticisms. We have accepted all of them and have revised the paper accordingly. The original comments of the referee are reprinted here, in the order that they were made, with blue. After each comment we detail the modifications which we made to our analyses and to the manuscript.

comment r4i00

References are correct concerning the latest particular publications on the various analysis methods. But there is not much effort to also cite the foundational works. In the main article this is probably due to restrictions imposed by Nature, but in the Supplementary Information it shows a somewhat insufficient consideration of the foundations / historical development. This deficit is linked in several places to a style of just stating what is done without explaining why. The latter question is then left to the reader to know or figure out. Explicit examples are given below.

We included references in several cases, as discussed below.

comment r4i01

1a) The abstract labels the reduction of uncertainty as "a factor of almost 2", while the main text makes this precise as a factor of 1.6. (This factor 1.6 has a significant uncertainty itself.)

As noted under issue r4i06, we have changed "almost 2" to 1.6.

comment r4i02

1b) The results are not given in a transparent way. The main part just lists results and numbers, which is a valid choice. However, one then expects that the Supplementary Information can be used, e.g. to understand how the uncertainties, in particular (c) and (d) of Figure 1, come about and how well one knows them. This is not really the case.

We improved the transparency by adding many new plots, tables and corresponding discussions into Section 6 (Window observables).

comment r4i03

1c) Very much related to 1b) there is a lack of information which means that the results, except for the final one, will be very hard to be checked and used by others working in the field. In my opinion, this is not acceptable given how relevant checks are and even more how relevant it is to reduce uncertainties, using also results from other computations. Several such alternative computations exist and science should profit from their combination. One also has to note that the computations were very expensive, which means it is entirely unrealistic that they will be repeated from the beginning. There are various intermediate results which are not provided in the needed form. Also the gauge field configurations which they are based on are not publicly

available. Most importantly the analysis of the numerical results is not explained and justified sufficiently. These facts mean that in the present form the publication does not really qualify as reproducible and accessible.

We have already given many intermediate results in the previous version, with the updated one we are giving even more. There are about 30 different results given in the paper, that can be crosschecked by other lattice groups. We perform comparisons for many of these, where there are already available results in the literature.

Regarding configurations and codes we added the following paragraphs into the main:

Data availability. *The datasets for the figures and tables are available from the corresponding author on request. Upon request from the PI the configurations can be given out for specific projects.*

Code availability. *A CPU-code for configuration production and measurements can be obtained from the corresponding author upon request. The Wilson flow evolution code, which was used to determine w_0 , can be downloaded from <https://arxiv.org/abs/1203.4469>.*

comment r4i04

1d) A somewhat more technical but essential issue is how the continuum limit is taken and how its uncertainty is estimated. The lack of provided information makes it difficult to judge it completely, but the largest two lattice spacings seem outside of the region which is useful for an extrapolation. It looks like the expansion in terms of the lattice spacing has broken down for those lattice spacings. Removing them might not change final results much but simply needs to be done in order to be more confident. The continuum limit of the short distance window is hard to believe the way it is taken / checked.

We have investigated the removal of these coarsest lattice spacings and found they had a negligible effect on the final result, see issue r4i24 for details. Regarding the short distance window continuum limit see issue r4i26 for details.

comment r4i05

1e) A further technical issue is the determination of the scale. The estimate of the systematic uncertainty is insufficient.

We thank the referee for pointing out an important deficiency in the scale determination using the Omega mass. Upon this we decided to change the scale setting procedure to use the pion decay rate as input. The new procedure is described in Section 3 (Scale setting using the pion decay rate). We also added the following paragraph to Section 6.1 (Blinding):

After unblinding, and during the referral process, concerns were raised by one of our referees about the possibility of an unknown excited-state systematic in the lattice determination of the Ω mass, as described in Section {se:si_omega}. Based on this criticism, the value of w_0 scale was instead calculated in an alternate blinded analysis based upon the experimental muonic pion decay rate, see Section {se:si_fpi}. The value of w_0 from this new determination was not known at the time of making this decision, but the value from the old Ω analysis was.

comment r4i06

"factor of nearly 2" needs to be changed to "factor of 1.6" (or whatever emerges).

We changed it.

comment r4i07

Figure 1 and its description: It is unclear how much of the statistical error (a) is contained again in (b-e) or whether (b-e) contain independent statistical uncertainties. E.g. (b) might contain a large statistical error. This is important since the statistical errors are much more firm than the systematic ones. A remark needs to be added.

We added to the caption of Figure 1 in the main text:

Note, that the statistical error (a) refers to that of the isospin-symmetric contribution in finite volume. The finite-size (b) and isospin-breaking (e) errors also contain a statistical component.

comment r4i08

l 129: "... light connected contribution is decreased by the new ensembles by 37% ..." The statement is not supported by the Supplementary Information. The decrease could also be due to the change in analysis (separate analysis in different windows).

We have included a paragraph to Section 6.4 (Light and disconnected 00–28 window) to clarify the impact of the new $a = 0.048$ fm ensemble:

Now we would like to quantify the improvement on our uncertainty resulting from including the ensemble at 0.048 fm, introduced in this work, and the additional error reduction arising from adding the tail from the data-driven approach. For this purpose we compare the uncertainties on the quantity $a_{\mu,04-40}^{\text{light}}$, obtained with and without the new lattice spacing, and the quantity $a_{\mu,04-28}^{\text{light}}$ obtained with the new lattice spacing. For consistency, the value without the new ensemble is obtained using the distribution of w_0 obtained also without the new ensemble. In order to have a fair comparison, we perform the exact same analysis in all three cases, fitting the interval without splitting into more windows. Note that since we now also include cubic fits in our analysis, this procedure is more conservative than that of our previous work {Borsanyi:2020mff}. For the uncertainties of these quantities we obtain

$$\begin{aligned} a_{\mu,04-40}^{\text{light}} \quad \text{without new ensemble} &\longrightarrow (2.10)(5.80)[6.17] , \\ a_{\mu,04-40}^{\text{light}} \quad \text{with new ensemble} &\longrightarrow (1.80)(3.46)[3.90] , \\ a_{\mu,04-28}^{\text{light}} \quad \text{with new ensemble} &\longrightarrow (0.96)(2.55)[2.72] . \end{aligned}$$

Then one can conclude that the inclusion of the new lattice spacing reduces the error by $1 - 3.90/6.17 = 37\%$, and including the tail from the data-driven approach gives a further reduction of $1 - 2.72/3.90 = 30\%$.

comment r4i09

l 150: "expected to be similarly small" is a speculation. Furthermore it seems to me the authors want to say "difference with schemes used" and not "difference with results obtained".

According to the request we changed the sentence as:

In {RBC:2023pvn} it is shown that the difference in the value of $a_{\mu,04-10}^{LO-HVP,light}$ obtained in the RBC-UKQCD scheme and in our scheme is approximately $0.10(24) \times 10^{-10}$, smaller than even our present uncertainties. The differences with other schemes used by the other collaborations are probably on the same level. However we emphasise that this scheme dependence in no way affects our final result for a_{μ}^{LO-HVP} , nor for the full value of $a_{\mu,04-10}^{LO-HVP}$ that includes all flavour, isospin-breaking contributions. Both are unambiguous physical quantities.

comment r4i10

l 160: "quantitatively exact" and the entire bullet point. I do not understand well what the authors want to say here. What is scheme dependent and what is not? This needs to be explained better.

We reformulated several sentences to make it clear, how the data-driven light connected contribution was obtained:

On the other hand, our new result for $a_{\mu,04-10}^{LO-HVP,light}$ differs from the data-driven one presented in {Borsanyi:2020mff} by 4.3σ . This number was obtained by using the total result $a_{\mu,04-10}^{LO-HVP}$ from the data-driven approach and subtracting all but the light-connected contributions measured in our 2020 lattice simulations.

comment r4i11

Figure 1 shows that only half of the beta-values allow for an interpolation to the physical point. The extreme case is the largest lattice spacing with a single point only, which is rather far away. I see no evidence that the global fits -- in particular those that dominate the result -- solve this issue. Based on the figure I conclude that $\beta=3.700$ and maybe also $\beta=3.7753$ need to be removed from the analysis.

See issue r4i24.

comment r4i12

p5, 1st paragraph: The smallest lattice spacing is rather small and topology change is expected to be suppressed. While the shown autocorrelation function looks good, it will improve the paper and support its conclusions to also list the topological susceptibility at the smallest lattice spacings and compare it to expectation. Of course, if that is published elsewhere, one can just point to the literature.

We added the following text to Section 1.1 (Action and esembles):

Topological properties of QCD were investigated with the 4stout action in Ref. {Borsanyi:2016ksw}, where the topological susceptibility χ was computed for a wide range of lattice spacings. The

continuum extrapolation of χ is notoriously difficult, because of the absence of exact zero modes of the staggered Dirac operator at finite lattice spacing. The behaviour towards the continuum can be much improved by rescaling χ with the square of the ratio of the Goldstone and taste singlet pion masses. On our finest lattice we find $\chi = 0.0358(29) \text{ fm}^4$ for the unimproved and $\chi = 0.0299(24) \text{ fm}^4$ for the improved susceptibility. These numbers nicely fit on the continuum extrapolation curves presented in Ref. {Borsanyi:2016ksw}. In that work it was found, that the continuum extrapolated value agrees well with the prediction of chiral perturbation theory.

comment r4i13

Figure 3: the figure compares data to a^4 while the text discusses $a^2 \alpha_s^3$. The latter should be plotted in the figure.

In Figure 3 in Section 1.2 (Taste violation) beside a^4 we now also plot $\alpha_s^n a^2$ with $n = 3$ and $n = 5$ and modified the describing sentences as:

We observe a decrease with approximately the fourth power of the lattice spacing towards our finer lattices. This is much faster than the $\alpha_s a^2$ expected from naive scaling, where α_s is the strong coupling constant at the lattice cutoff scale. The falloff is actually consistent with an $\alpha_s^n a^2$ type behavior with $n = 3$ on the coarser and with $n = 5$ towards the finer lattices.

comment r4i14

p6, last paragraph: referring to [14] does not help, since there is no justification given in [14].

We modified the sentence in Section 1.2 (Taste violation) to make our assumption clear:

Here we assume that these artefacts decrease with the same rate as $\Delta_{KS}(\xi)$; the same assumption was made in Ref. {FermilabLattice:2018zqv}.

comment r4i15

Section 2: There is a small amount of references. Various techniques go back to the literature and this should be shown. An example is that eight sources (bottom of p 7) was proposed and tested already in Phys.Lett.B 162 (1985) 160.

We added reference {Billoire:1985yn} as requested by the referee.

We added {Blossier:2009kd} at the generalized eigenvalue problem.

We added the references {Neff:2001zr} and {Li:2010pw} about the low mode averaging technique.

We added references {Jegerlehner:2011ti}, {Sakurai:1960ju} and {Chakraborty:2016mwy} regarding the SRHO model, and the references {Lee:1999zxa}, {Sharpe:2004is}, {Aubin:2006xv} and {Aubin:2019usy} on staggered XPT.

We added the references {Akaike:1974vps}, {Akaike1973} and {Akaike1978c} on Akaike's Information Criterion.

comment r4i16

The discussion on excited states on p8 is too naive. Why are the states resonances? The relation of finite volume states to resonances is complicated. Why do the authors think that the states are not close to 2-particle, 3-particle states? What is the (free) low-lying multi-particle spectrum in the present setting? Can one expect that the GEVP-type analysis determines such states? Such a discussion will impact the question whether the analysis which was carried out is sufficient, in particular to estimate its uncertainty.

Indeed, as the referee pointed out, we make no estimate of the uncertainty in our omega mass determination arising from contamination by multi-hadron excited states. We therefore changed the primary scale setting procedure to use the pion decay rate as input, see Section 3 (Scale setting using the pion decay rate). We also added remarks warning of the potential effect of multi-hadron excited states in Section 2 (Scale setting using the omega baryon mass):

The determination of the ground state baryon mass, as detailed in this Section, has a caveat. It does not account for the presence of non-resonant, scattering states in the excited state spectrum, like (Ξ, K) , (Ω, π, π) and other multi-hadron combinations. Though our single-hadron operators are expected to couple weakly to these states, we cannot guarantee that the scattering state contamination remains within our quoted precision. A similar problem occurs in the case of nucleon propagators, where the (Δ, π) states distort the ground state determination. This problem has been investigated in chiral perturbation theory {Bar:2015zwa, Tiburzi:2015tta} and also on the lattice {Hackl:2024whw}. If we compare them, it seems that chiral perturbation theory overestimates the contamination in nucleon propagators.

comment r4i17

I disagree completely with the present way of estimating the error due to the fit-range. Changing t_{\min} by 0.1 fm does not change the contribution of excited states much.

Note, we abandoned the omega baryon from our scale setting procedure in favor of the pion decay rate. Nevertheless, we investigated excited state contamination by performing multi-state fits on our omega correlators and added the following text to Section 2.2 (Omega propagator fits):

To investigate the excited state contamination of the ground state masses, we perform correlated fits to the ground state propagator with four exponentials instead of one. We introduce 100 MeV wide priors on the excited states, for the central values we use masses of Ω resonances taken from the Particle Data Book. This choice of priors assumes that the dominant contaminations come from states near the resonance energies. This is motivated by the local nature of our Omega operators, but ignores possible contributions from the non-resonant scattering states discussed at the end of this section. Results of these fits are shown in the effective mass plot of Figure ..., in case of our finest two ensembles. The ground state mass is in good agreement between the single-state and four-state fits.

comment r4i18

Therefore such a change underestimates the uncertainty. More on the speculative side, in fig. 4 there seems to be a bit of a step-like structure below and above 0.8 fm. It looks as if below 0.8 fm the GEVP captures the 2nd

excited state (or some state around there) and above it loses this information. One worries whether this means that the data points for the ground state are better between 0.5fm and 0.8fm than above. Of course this is against standard expectation. To be able to judge whether the step-structure is just a statistical fluctuation, one would like to see more (of the small a) ensemble results. In the present form, the systematic error due to excited states is not seriously accounted for. The 0.15% shown in figure 1 could be quite off.

We have updated the effective mass plot, Figure 5, in Section 2.2 (Omega propagator fits), adding results from the other ensemble at the finest lattice spacing. On this ensemble, we see no evidence for a step-like structure. We also show the result of multi-state fits onto these correlators.

comment r4i19

Section 3: The last sentence before 3.1 triggers the question whether just the number from the publication is taken or whether the correlation with the new analysis is taken into account. This needs to be stated and possibly justified.

We added the following sentence to Section 4.1 (Physical point):

The procedure is described in detail in Section {se:si_analysis}. This approach respects the statistical correlations, while most of the systematic ingredients are treated as uncorrelated. We justify this choice by noting that our basis observables have very different lattice artefacts to the ones with which we combining them.

We also added the following text to Section 6.6 (Other contributions and total):

We are not correlating the isospin-breaking, the strange, and charm computations with each other or with the light plus disconnected part. We justify this choice by noting that these observables describe very different physics, each with its own kind of lattice artefact and in case of the isospin breaking sometimes they are measured on different sets of ensembles than the isospin symmetric ones.

comment r4i20

Figure 7 and its discussion: Ref. [24] does an electro-quenched computation. Should one mention that or compare to the electro-quenched result in the BMW scheme? In [25] I could not find a definition of the scheme at all. The scheme of [25] therefore needs to be defined here, such that the reader can judge what is shown. The last sentence of section 3 is incorrect. The Kaon decomposition can be correct at the level of 0.1MeV-0.2MeV in the splittings as seen here and still the scale can be different by e.g. 1% and then a_μ by 2%. The true check for scheme-sensitivity is the one given in the main text.

In Section 4.3 (Kaon mass decomposition in different schemes) we commented on electro quenching as:

Figure ... shows a comparison with the same decomposition in the GRS scheme. The values are taken from Ref. {Giusti:2017dmp}. Note, that they are computed in the electroquenched approximation, whereas our results also include sea-valence electromagnetic effects. Sea-sea

electromagnetic effects are absent in mass isospin splittings. The neglected sea-valence effects in Ref. {Giusti:2017dmp} should be small, due to $SU(3)$ flavor suppression.

Reference [25] provides the decomposition of the kaons only, so it is not a full scheme definition. We clarified this by not referring to [25] as scheme any more, and we use their kaon decomposition for comparison only.

Also we removed the last sentence in this Section.

comment r4i21

Section 4 and continuum extrapolations in other sections: 1) What is Δm in Table 3?

It means attometers, we clarified it in the text.

comment r4i22

2) The global fit procedure and therefore the continuum extrapolations are rather intransparent. A central deficit is that it is not possible to find out what kind of fits dominate the analysis and where systematic errors come from. There is no information on the AIC weights in the form of some table or graphical representation. This needs to be changed. A first step is to drop fits with very low weights from the figures. For example, Table 3, top figures, contain a number of fits which miss the data points by far. These are plotted nevertheless and make it impossible to see where the green region below 172 in the lower figure as well as the long tail to higher values in the lower figure come from.

We updated many of our continuum extrapolation plots by adding a color coding scheme: fits with higher/lower AIC values are now shown with a lighter/darker color. In particular, in the w_0 extrapolation, in Section 2.3 (Determination of w_0 using the Omega mass) we have dropped fits with very low weights, those which were missing some of the data points by far. We have also added an explanation of the source of the long tail in the w_0 distribution to the figure caption.

comment r4i23

One also has to note that the PDF discussion sounds like objective statistics, but it depends in an arbitrary way on which fit functions have been chosen. E.g. there is an underlying assumption that the true model is contained in the set of fit functions. This is definitely not the case here. The true function describing cutoff effects is much more complicated with many powers of a and α .

We agree with the referee that the true function is complicated with many powers of a and α , but such functions are not practical in case of noisy data points and limited number of lattice spacings, since they tend to overfit the data. We therefore work with low-order polynomials, most of which produce reasonable fit qualities, e.g. in case of all the relevant single-window fits for a_μ at least one third of our fits have fit quality over 0.1. For more clarity, we have included for each analysis in Sections 2, 3, and 6 the percentage of fits with P -value larger than 0.1.

comment r4i24

A few more specific remarks follow. 3) The data points in Table 3 provide very strong evidence that the largest lattice spacing is not described by a few powers any more. The a -expansion has broken down. It could also be that the simple linear $B(a^2)$ and $C(a^2)$ are entirely insufficient to describe the deviations from the physical point at larger a . Either way, the largest one or two lattice spacings need to be dropped completely and then one would like to see what is the systematic error when more lattice spacings are dropped.

In the multi-window analysis for the light connected 04 – 28 window, we dropped the coarsest two lattice spacings, as suggested by the referee. Numbers are median(stat)(sys-w0)(sys-taste-violation)(sys-fit-form)[total]:

$$\begin{aligned} 572.11(0.96)(2.25)(0.50)(0.87)[2.76] & \quad 3\text{-win} \\ 572.29(1.01)(2.25)(0.65)(0.81)[2.83] & \quad 3\text{-win} + \text{dropcoarse} \end{aligned}$$

Note, these numbers still use the omega based scale setting. We see, that these fits do not change the central value at all and increase the total error by only 3%. This increase should be well within the uncertainty of our error estimate (aka the error-on-error).

The new w_0 analysis based on the pion decay rate does not use the coarsest lattice any more.

comment r4i25a

4) I see no justification for a fit-function with just $A'(\Delta_{KS})$ for the discretisation errors. Everywhere w_0 is involved through the lattice spacing. This is a pure gauge and gradient flow quantity, where taste splittings are very subdominant. The pure gauge and gradient flow terms in the Symanzik expansion will dominate. Of course a fit-function with two or three powers of α may be reasonable. Again dropping larger lattice spacings helps to be less sensitive to the detailed form of the extrapolation.

The new analysis for w_0 uses the pion decay rate as input, which is known to be affected by taste violation errors, hence we prefer to keep also the pure Δ_{KS} fits and let the AIC decide, whether it is a reliable description of the data or not. We use fits both with a^2 and Δ_{KS} terms, also ones where the two types of terms are mixed. In that analysis we already ignored our coarsest lattice spacing, as the referee requested.

comment r4i25b

It is also not satisfactory that there is no discussion about what is known about the discretisation errors, in particular of gradient flow quantities and how this leads to the fit-functions used. In my opinion any discussion will lead to the conclusion that the fit-function is essentially unknown but (as stated earlier) it is extremely hard to imagine that the discretisation errors at small lattice spacings are dominated by taste breaking effects.

If the fit qualities are good, we have no strong reason to exclude any of these fits. Also we tried to use an observable independent fit procedure to an extent as much as possible, and rely on the AIC to decide on the best fit.

comment r4i26

5) Table 4: the scale of the figure makes it impossible to judge what happens at small a , in particular at the level of the cited total error. In addition, for the small distance window the situation is special beyond this simple point. As noted in the text the quantity is not on-shell. Therefore there is no theory for the discretisation errors and statements such as "logarithmically enhanced" may be true at tree level but are not backed by anything beyond that. The functional forms (24) are ad-hoc. Without a demonstration that the short distance window (or if needed even shorter distance one) is compatible with a continuum perturbative estimate, the result (25) cannot be trusted. The authors have a statement that they know the sign of the log-enhanced term at tree level. They should show how they arrive at this and give the numbers. The question also arises whether it is seen in the numerical data.

We have included a comparison between lattice results and perturbation theory in Section 6.2 (Short distance window). We find good agreement below 0.3fm.

comment r4i27

6) Table 6: the "Continuum parameter" seems too small, judged from the figure.
7) Table 7: same as Table 6.

For computing the error corresponding to systematic variations we use the variance approach of {Jay:2020jzk}. That is, the systematic error corresponding to the variation between a^2 and Δ_{KS} is obtained as the variance of the two-point distribution, with Dirac-deltas located at the two means obtained with a^2 and Δ_{KS} fits.

In the case of $a_{\mu,04-10}^{\text{light}}$, the relative weights of the a^2 and Δ_{KS} are 0.45 and 0.55, respectively, while the difference between their means is 0.8 in units of 10^{-10} . The standard deviation of this two-point distribution yields the value 0.40 of Table 8. Note, that the distribution corresponding to the a^2 fits (the peak on the right) is somewhat wider in the left side, therefore, its mean is located around 0.3 units lower than the maximum of the corresponding peak.

In the case of $a_{\mu,04-10}^{\text{disc}}$, the relative weights of the a^2 and Δ_{KS} are 0.95 and 0.05, respectively, and the difference between their means is 0.16 in units of 10^{-10} , which yields the standard deviation of 0.035 given in Table 9.

comment r4i28

8) I have expressed my strong reservation concerning the probability interpretation of the plotted PDF. Still, as they appear in the paper, they show long tails and double-peak structures. The reader would then like to know where these structures come from and what this means for the systematic uncertainties.

In case of several histograms with apparent multi-peak structure we have added explanations in the text or in captions.

comment r4i29

Additional suggestions concerning 1c): A way to make the results usable and

testable for other groups is to publish the full covariance matrix of all results. Alternatively it may be convenient to publish the jackknife bins. Of course "publish" means to provide machine readable files. By "all results" I mean the results for the physical point and all different contributions to a_μ , i.e. different windows, light, strange ... I further consider it necessary to provide these numbers before extrapolation to the physical point and continuum limit as well. This level of details is standard in the field and thus a minimum for Nature publications.

We currently follow the approach of most other groups, like RBC-UKQCD, FNAL-HPQCD-MILC, ETMC, who provide many intermediate numbers without giving access to "all results". In our current work we give about 30 different intermediate results, that can be used for comparisons with other groups. The Theory Initiative requires only to produce the all-flavor result for three windows, the $00 - 04$, $04 - 10$ and $10 - \infty$, which we have now provided in the paper.

comment r4i30

It is not possible to check what is used since "Qlattice" mentioned in suppl. sect. 1.1 is unpublished. This is a major part of the analysis software.

We added a link to the Qlattice software.

Changes

Following the suggestions of the reviewers, we performed the following major changes in our analysis:

- Reviewer 4 raised concerns about our continuum-extrapolation procedure. In response to this criticism, we have modified our procedure by removing the $\Delta_{KS}(a)$ variable from the fits entirely. Instead, we have introduced an anomalous dimension to the leading a^2 term, ie. $a^2\alpha_s(a)^\gamma$, and varied the value of γ in a well-motivated range. Before unblinding, we decided to adopt this new procedure for all of our lattice results - except for the M_Ω -based determination of w_0 , which does not enter the final result and which we left as is. The new results turn out to be entirely consistent with our previous ones. Since the continuum extrapolation impacts all observables, the numerical values of most of our results have changed, though well within their quoted uncertainties.
- Reviewers 2 and 3 asked for a determination of the long-distance window based on lattice QCD alone. We have performed this calculation and present the result in Section 6.7. The value came out consistent with the White Paper average of this quantity.
- Reviewer 4 asked for details about our continuum extrapolation, beyond what is plotted in the figures. We satisfied this request by making the data points and extrapolation curves in each continuum limit plot available in computer-readable form in CERN's data repository [doi:10.5281/zenodo.17880027](https://doi.org/10.5281/zenodo.17880027). The DOI will be available upon publication, currently the files can be accessed from here.

All substantial changes with respect to our second submission are highlighted in blue in the manuscript.

Answers to Referee 1

We thank the referee for their valuable comments, suggestions, and criticisms. We have accepted all of them and have revised the paper accordingly. The comments of the referee are reprinted here in blue in the order they were made. After each comment, we detail the modifications that we made to our analyses and to the manuscript.

comment r1j00

The discussion of error reductions is largely focussed on previous BMW calculations, rather than the current community consensus which is the 2025 Theory Initiative result. The paper simply says on line 212 that "Our LOHVP is in good agreement with the latest Theory Initiative combination" without making the key point that the uncertainty here is almost a factor of 2 smaller?

5) Line 212 would make a stronger ending to the paragraph if it made clear the improvement in uncertainty achieved here compared to the latest Theory Initiative combination.

We added a sentence ending the paragraph as:

... and our uncertainty is a factor of 1.8 smaller.

comment r1j01

1) I think the first paragraph is a bit confusing for the general reader now, particularly the sentence on line 33; "Furthermore, new input data ...similar changes". The confusion arises because the reader does not know at this point why "input data" are needed for the LOHVP, nor which LOHVP result the comment refers to. I think this needs to be changed, but I do not have a good suggestion for how to do this succinctly.

Another point of confusion in this paragraph is the reference to a 2021 lattice QCD calculation that is later, in Figures 2 and 3, tagged as BMW'20.

The motivation for the paper would be improved for the reader by making clear the uncertainty on the LOHVP in the recent update of the reference standard-model prediction (lines 34-35).

We changed the sentence on line 33 to read "Furthermore, related experimental results indicated similar changes".

We changed the 2021 to 2020 in the abstract. (We consistently refer to our previous work as 'our work from 2020' in the paper.)

comment r1j02

2) lines 62-62 would read more smoothly as:
... that allow QCD predictions to be made ...

We incorporated this change.

comment r1j03

3) Line 191 typo: of our -> our

We corrected it.

comment r1j04

4) Line 211 - from the Figure it is clear that the plotted CMD-3 data-driven result does not have "serious tensions" so this sentence needs to be more carefully written.

We have rewritten the sentence as:

As the figure shows, some of the data-driven results are in serious tensions both with our and the lattice only estimates.

comment r1j05

6) Line 214 (... confirm or refute our results) reads a bit oddly now in 2025 that the theory consensus (including results from other lattice collaborations) and this paper agree on the picture presented. The issue now is not one of confirming or refuting but whether other lattice collaborations can achieve comparable accuracy (perhaps using the approach suggested here) and whether the uncertainty on the theory result for a_μ can become as small as the experimental uncertainty.

We removed the sentence with the "confirm/refute".

comment r1j06

7) New Section 3 of the Supplementary Information: Figure 8 was rather confusing - the caption should be explicit about the regions shown as bands. From the text in section 3.1 I was expecting 3 bands but there only seemed to be 2. How the systematic uncertainty for residual excited state contamination is determined should be made clearer.

We have updated the caption to say: *whose x -extents correspond to the two plateau regions (1.9 fm–2.9 fm for the first, and 2.1 fm–3.1 fm for the second) and y -extents, to the uncertainties on the associated fit results.*

We also updated the text to be clearer that there are two plateau regions considered for the pion.

Answers to Referee 2

We thank the referee for their comments, suggestions, and criticisms. We have considered all of them and have revised the paper accordingly. The original comments of the referee are reprinted here in blue in the order they were made. After each comment, we detail the modifications that we made to our analyses and to the manuscript.

comment r2j00

This work is not suitable for publication without LQCD-only results for t/fm in $(1.0, \text{inf})$, $(2.8, \text{inf})$ and the full amuLO-HVP.

We disagree that computing the long-distance window is relevant for judging our current calculation - we have provided several other windows for comparisons. However, we do agree that it would be a service to the community and we are happy to oblige. Therefore, we computed the light-connected $10 - \infty$ window using lattice QCD alone, the results are presented in Section 6.7 (Long-distance window $10 - \infty$).

Computing the other long-distance contributions (eg. isospin-breaking, finite-size) in lattice QCD, with precision going beyond our 2020 work, is a significant challenge, and will probably take several years of community effort to achieve.

comment r2j01

the title of a published version should not overstate the main result.

We updated the title to include “Hybrid”. It is a factual description of our work.

comment r2j02

How large are the dimensionful coefficients of the discretisation effects?

For each of the continuum extrapolations appearing in the plots we have provided the contributions from the A_2 , A_4 , A_6 and A_l terms at a representative lattice spacing of 0.1 fm. These can be found in the files uploaded to [doi:10.5281/zenodo.17880027](https://doi.org/10.5281/zenodo.17880027).

Answers to Referee 3

We thank the referee for their valuable comments, suggestions, and criticisms. We have accepted all of them and have revised the paper accordingly. The original comments of the referee are reprinted here, in the order that they were made, with blue. After each comment, we detail the modifications that we made to our analyses and to the manuscript.

comment r3j00

1. I still have severe reservations about the estimation of the significance of the difference between the 2020 result and the new estimate described on p9. I elaborate on this issue further below (comment 4 below).

4. The discussion of the tension with their 2020 result on p48 is essentially unchanged. No explanation is offered regarding the source of the difference, with the discussion entirely focused on assigning an error to the difference (which is, in fact, as large as the total error on the entire 2020 result). I still disagree with the way that the uncertainty on the difference has been estimated. For once, the error on the finite-volume correction should not enter this analysis at all, since the estimate for the size of the correction has remained stable and should affect both calculations in a similar fashion, regardless of its error. Secondly, I strongly disagree with the argument that the two calculations can be treated as uncorrelated, since they are still based on largely overlapping ensembles, despite different analysis procedures. By far the most likely source of the difference with the earlier result is the continuum extrapolation of the light-quark connected contribution and the role of taste-breaking effects. This is where the added ensemble at finer lattice spacing should make a real difference as it may reveal a previously unnoticed systematic effect. This scenario could be tested by computing the correlated difference between the results whose errors are listed in the first two lines of Eq. (57). I realize that those estimates arise from model averages so that correlations cannot be preserved fully, but such a procedure would lead to a much more solid estimate of the uncertainty assigned to the difference and may offer an explanation as to why the result has shifted.

The referee is right; the source of the difference with our previous work is the light-connected contribution. We make this point more explicit by comparing our old and new results, this time for the light-connected contribution only, at the end of Section 6.7 (Long-distance window $10 - \infty$). To be more transparent, we performed the comparison in our reference volume. As explained there, the difference comes partly from the new lattice spacing. The rest of the difference arises from change in analysis procedure. In addition, we compute the correlated difference, as requested by the referee, also considering the extreme possibility that the systematics are 100% correlated.

We keep our original computation of the correlated difference of our total value in Section 6.9 (Other contributions and total), which assumed zero correlation in the analysis procedures. In addition, we compute the difference assuming full correlation between the remaining systematics, as requested by the referee.

Regarding the referee's comment on the finite-size correction, we note that the 1.9 was obtained by computing the appropriate correlated difference between a_μ and $a_{\mu,00-28}$ in our `4hex` simulations. In any case, we also provide the correlated difference without finite-size corrections, in the reference volume for the light-connected contribution.

Following the referee's suggestion, we now give the results (and not only the errors) for the comparison with and without our finest lattice spacing, at the end of Section 6.7 (Long-distance window $10 - \infty$).

comment r3j01

2. The sentence "In the near future we expect that other lattice collaborations [...] confirm or refute our results" is not needed, since new results have been published and a consolidated lattice result has emerged as part of the White Paper update.

We removed that sentence.

comment r3j02

3. Section 6, Eq. (57). I appreciate the efforts to address my comment on elaborating on the improvement of the precision relative to the 2020 publication. I am puzzled, however, why the authors only quote the uncertainties but withhold the actual values of these quantities. In order to properly assess the impact of the added ensemble, it is also necessary to judge whether the results for the 0.4-4.0 fm window are compatible within errors. This is one of several instants where the transparency of the paper should be improved.

To illustrate the reduction in uncertainties, we now use the light-connected long-distance window in Section 6.7 (Long-distance window $10 - \infty$). As requested by the referee, we not only give the uncertainties but also the actual values. From this one can also assess the impact of the new ensemble. The results with and without the new ensemble are compatible within errors.

comment r3j03

5. In section 6.6 the authors provide an estimate for the long-distance window observable based on the "hybrid" method. Actually, this is not what I had requested, which was rather an estimate of the long-distance window based on lattice QCD alone. In section 10, the authors speak of a "somewhat disadvantageous uncertainty of 8%" on the 2.8-3.5 fm window relative to the data-driven approach, but this cannot serve as an excuse for not quoting an actual number for the long-distance window based on lattice data, which would make the comparison with other recent lattice calculations (and the consolidated lattice average quoted in the White Paper update) of the same quantity much more meaningful. I cannot think of a scientific argument why this information should be withheld, unless the authors find that taste-breaking effects are uncontrollably large when extending the integration from 2.8 fm to infinity. If this were indeed the case, it should be stated explicitly in the manuscript as it would point to a limitation of

staggered quarks, which manifests itself as an irreducible uncertainty in the long-distance tail. Only by adding more ensembles with finer lattice spacings could the integration be pushed further into the long-distance regime. That the treatment of taste-breaking effects is a highly sensitive issue has been acknowledged by the authors when they point out that -- contrary to their earlier practice -- they restrict the correction for such effects to distances larger than 1.2 fm. To summarize this point: For the sake of transparency, the corresponding result for the long-distance window from the "lattice-only" approach must be provided. In addition, the authors should add a plot comparing their result for the long-distance window to other recent calculations (RBC:2024fic, Djukanovic:2024cmq, FermilabLatticeHPQCD:2024ppc), similar to the compilation plots in Fig. 2 of the main body or Tables 6-10 in the Supplemental Information.

In Section 6.7 (Long-distance window $10 - \infty$), we provide our estimate of the light-connected long-distance window, based on lattice QCD alone, using the bounding method for the tail of the correlator. We compare it with the recent calculations in Figure 14 as the referee requests.

comment r3j04

6. Tables 23 and 24: insert the results for the window 10-28, similar to what was done for Tables 20, 21 and 22.

We added these missing finite-size-correction numbers for the 10 – 28 window to Tables 23 and 24, and also added the numbers for the 10 – ∞ window to all tables.

comment r3j05

7. Figure 20: 0.55 MeV  0.55 GeV in the caption.

We corrected this.

comment r3j06

Section 6, footnote 6: During the journal submission procedure, two more results for the long-distance window have been published (Djukanovic:2024cmq, FermilabLatticeHPQCD:2024ppc) that should be included.

On p56 the authors have added references to papers on "Low-mode averaging" which should also include DeGrand:2004qw and Giusti:2004yp.

We removed the footnote, since we now have an extended discussion of the light-connected long-distance window; see r3j03.

We added the two citations regarding LMA.

comment r3j07

The issue of the intrinsic precision of the lattice contribution mentioned in comment 5 above also matters because the authors have chosen to include the final precision in the revised title of the resubmission. At that point it must be stated more clearly in the abstract (i.e the part in bold face on

p2 of the main paper) that this level of precision can only be reached through the combination of both approaches: While lattice QCD is used to circumvent the problem of (presently) incompatible e^+e^- hadronic cross section data in the region of the rho-omega peak, the data-driven approach is crucial to overcome the limitations in reaching a similar level of precision in the lattice calculation of the long-distance regime. It is crucial for the reader to appreciate the complementary nature of the two methods.

We have updated the abstract to make the importance of the hybrid approach clearer:

To reach this unprecedented level of precision, we adopt a hybrid approach that includes a small, long-distance contribution obtained using input from experiments in a low-energy regime where they all agree. This approach exploits the complementary strengths of the experimental and lattice data in different energy ranges to achieve better precision than would be possible with either alone. We also improve on many aspects of the lattice calculation. In particular, we perform large-scale lattice QCD simulations on finer lattices than in Ref. {Borsanyi:2020mff}, allowing for an even more accurate continuum extrapolation.

We have also included “hybrid” in the title of the paper.

Answers to Referee 4

We thank the referee for their valuable comments, suggestions, and criticisms. We have accepted all of them and have revised the paper accordingly. The original comments of the referee are reprinted here, in the order that they were made, with blue. After each comment, we detail the modifications that we made to our analyses and to the manuscript.

comment r4j00

Comment on reply to r4i02

> We improved the transparency by adding many new plots, tables and corresponding discussions into Section 6 (Window observables).

Yes, more information has been added, in particular in section 6 (I always refer to the new sect./fig./tab. numbers.). Unfortunately this was not done everywhere and in particular not where it matters most for the final result. For example a big part of the result is the 12-28 window, Table 14, and its uncertainty dominates the total uncertainty. In this case the only information concerning the weight of fits is which fit has the highest weight. This tells very little. The color code introduced in other figures is not given here. Thus this piece and not only it remains rather intransparent.

The missing AIC color coding has been added to all tables in which it was missing. (We keep the uncorrected curves colored grey so they aren't confused with the included curves.)

comment r4j01

Comment on reply to r4i03

> Data availability.

> The datasets for the figures and tables are available from the corresponding author on request.

1. Why does one introduce the necessity to write to the author? What if he is not available for a longer period? The numbers just need to be listed.
2. In several cases, the plotted numbers are not sufficient. Examples are Tab 5: numbers for w_0/a , $fud a$; A list of the fits with their weights down to a reasonable cut (on the weights).

> Upon request from the PI the configurations can be given out for specific projects.

This is an empty statement as there is no commitment to anything. I request to remove it.

> Code availability.

> A CPU-code for configuration production and measurements can be obtained from the corresponding author upon request. The Wilson flow evolution code,

> which was used to determine w_0 , can be downloaded from <https://arxiv.org/abs/1203.4469>

Do the authors mean that the code which was used in this project for the configuration production and measurements can be obtained? Then write it.

Otherwise remove the statement since several production codes of other collaborations are publicly available.

As requested by the referee, we now provide for each continuum limit plot a list of all the data points and extrapolation curves together with corresponding AIC weights. These datafiles are uploaded to CERN's data repository doi:10.5281/zenodo.17880027.

We removed the statement regarding the configurations and changed the one about code availability to:

A CPU-code, which was used for configuration production and measurements, can be obtained from the corresponding author upon request.

comment r4j02

Comment on reply to r4i07

> Note, that the statistical error (a) refers to that of the isospin-symmetric contribution in finite volume. The finite-size (b) and isospin-breaking (e) errors also contain a statistical component.

It is easy to give the complete information: "... The finite-size (b) and isospin-breaking (e) errors also contain statistical components of xxx % and yyy % respectively. I request this change.

We have added the extra information to the manuscript as requested:

... The finite-size (b) and isospin-breaking (e) errors also contain statistical components of 0.08% and 0.16%, respectively.

comment r4j03

Comment on reply to r4i11

Referring to r4i24 does not help. My concern was also due to $\beta = 3.7753$. No evidence is shown that the extrapolation to the physical point works in that case and one wonders why ΔK_S is not relevant there. Actually the $\beta = 3.7753$ data look odd in the larger distance windows in section 6. How much influence this has, e.g., in the continuum extrapolation in Table 14 is impossible to see since the plotted data is not available, the scale of the figure is large compared to the precision and there is no information which fits dominate.

We dropped completely the coarsest lattice spacing $\beta = 3.7000$ from all the analyses. We also investigated the impact of the $\beta = 3.7753$ ensemble on a larger distance window, as requested by the referee. In case of the 10 – 28 window we get

$$\begin{array}{ll} 366.38(1.76)(2.23)[2.84] & \text{with } \beta = 3.7753 \\ 366.60(1.70)(2.22)[2.80] & \text{without } \beta = 3.7753 \end{array}$$

The impact of these ensembles is negligible on the result.

comment r4j04

Comment on reply to r4i12

> On our finest lattice we find $\chi = 0.0358(29)$ for the unimproved and $\chi = 0.0299(24)$ for the improved susceptibility. These numbers nicely fit on the continuum extrapolation curves presented in Ref. Borsanyi:2016ksw. ... The added information is very useful and reassuring. ... I think the authors

mean the curve for $T = 150$ MeV. Please add that information or the figure number.

The curve is shown in Figure S1 of {Borsanyi:2016ksw}. We added this information.

comment r4j05

Comment on reply to r4i13

> In Figure 3 in Section 1.2 (Taste violation) beside a4 we now also plot $\alpha_n a^2$
> with $n = 3$ and $n = 5$...

2The Fig. could easily be improved by plotting the curves close to the data.

We have moved the curves closer, and changed their plotted colours to keep them distinguishable from one another.

comment r4j06

Comment on reply to r4i16

> ... and also on the lattice Hackl:2024whw. If we compare them, it seems that
> chiral perturbation theory overestimates the contamination in nucleon propa-
> gators.

The reference Hackl:2024whw seems somewhat arbitrary as there has been a long debate on the question. I would avoid going into this discussion or cite a review. The final sentence is a speculation ("it seems") and it is unclear what is compared. Again, I advise to drop this part of the discussion.

We dropped this part of the discussion.

comment r4j07

Comment on reply to r4i17

The caption of the new Fig. 5 is missing the second beta value or should not contain the one given.

The figure contains two different ensembles with the same beta value and with slightly different quark masses. We fixed the caption accordingly.

comment r4j08

Comment on reply to r4i22

> We updated many of our continuum extrapolation plots by adding a color coding
> scheme: fits with higher/lower AIC values are now shown with a lighter/darker
> color.

The presentation of the results is somewhat improved, but it is still almost impossible to tell, e.g. which fits make up 70% of the weights or which five fits are the most important ... It is easy to improve it further by explicitly listing the fits with their weights down to a certain cut of a minimal weight. It may also be possible to use a better color coding ranging from red to blue or so and to adjust the scale of the figures such that lines are better resolved.

Furthermore, as remarked above essential continuum extrapolation plots are not improved at all.

We improved the presentation as requested by the referee by uploading data files corresponding to the plots to doi:10.5281/zenodo.17880027, see r4j01. Additionally, we updated the colour coding used for the AIC weights closer to the referee’s request while still preserving some readability in greyscale.

comment r4j09

Comment on reply to r4i23

> We agree with the referee that the true function is complicated with many
 > powers of a and α , but such functions are not practical in case of noisy data
 > points and limited number of lattice spacings, since they tend to overfit the
 > data. We therefore work with low-order polynomials, most of which produce
 > reasonable fit qualities, e.g. in case of all the relevant single-window fits for a
 > at least one third of our fits have fit quality over 0.1.

The authors seem to miss a basic point: A wrong function may fit the data well (have a good P-value) but yield the wrong extrapolation. The P-value only tests the function where one has data. Therefore the P-value has to be good, but does not guarantee the extrapolation is (about) correct. The latter has to be made plausible by theoretical considerations on the fit-function. One reason to show what dominates the results is to allow the reader to judge whether reasonable fit-functions are used there. In particular a polynomial of DeltaKS without an a^2 term is very hard to believe.

If fits with polynomials of DeltaKS do dominate the 12-28 window (I can’t know because the information is not given), then the result is rather doubtful. According to the table, w_0 plays an important role. The $O(a^2)$ terms in w_0 are significant as we see from the difference between different flows in Table 5. Therefore fits without a^2 terms make little sense.

To address the concerns of the referee we updated our continuum extrapolation procedure by removing fits that were polynomials of Δ_{KS} . At the same time we included an $\alpha_s(a)^\gamma$ factor in the a^2 -term and performed fits with different values of the anomalous dimension γ . The procedure is described in Section 5.1 (Fit functions). We performed the new analysis for all our observables in a blinded manner and decided to take the numbers obtained with the new procedure as our final result. After unblinding, the new results turned out consistent with the old ones.

To investigate the impact of pure a^2 effects coming from the flow, we performed our analysis for a long-distance window 10 – 28 window, in two different ways, one with the value of w_0 obtained from the Wilson-flow, and the other from the Zeuthen flow. We obtained

366.38(1.76)(2.23)[2.84]	Zeuthen-flow
366.74(1.75)(2.26)[2.85]	Wilson-flow

The difference between the two, a purely classical a^2 effect, is much smaller than other discretization errors. In our window results, we now use the Zeuthen-flow based w_0 .

comment r4j10

Comment on reply to r4i24

In the reply an example of dropping lattice spacings is given for one specific case. The tables show different numbers of lattice spacings for different observables,

ranging from 5 to 7. The reader would like to see data points for all lattice spacings, even if some are not used in the analysis.

In these plots, we use the global fits to the data to move the plotted points to the physical quark masses. Sometimes there are no good fits that include the coarsest lattice spacings, so we are unable to shift these points reliably, and hence we do not have a good way to plot them. Note, we now provide the data points and fit functions as external data files under doi:10.5281/zenodo.17880027.

comment r4j11

Comment on reply to r4i25a

> prefer to keep also the pure DeltaKS fits and let the AIC decide, whether it is a
> reliable description of the data or not.

The AIC can only decide (if at all) whether the fit is a good description of the data in the range where there are data. The extrapolated value depends on the fit-functions inserted into the AIC. Therefore the above statement does not help at all.

This fundamental problem in the continuum extrapolations remains

As requested by the referee we dropped pure Δ_{KS} fits from the analysis, see r4j09.

comment r4j12

Comment on reply to r4i26

I appreciate the comparison to perturbation theory. The results for the short distance window are now much more plausible. However, the functional forms in eq. (49) are ad hoc. Why is there no a^3 term? 1 Why is there $a^2 \log(a/w_0)$ and not $a^2 \log(a/w_0)^k$ with some other power k ? The minimum change required is to write clearly around (49) that the true (asymptotic) form is unknown since there is no theory analogous to Symanzik's theory for discretisation errors and that consequently the choices of fit functions are purely phenomenological.

We have added the following sentence here:

Note that the true asymptotic form of the discretization errors is unknown, since there is no theory analogous to Symanzik's for the discretization errors of this short-distance quantity. Therefore, the choices given in {eq:a2log} are phenomenological.

comment r4j13

The "This work" point in the comparison figure 1 and in Table 8 contains the finite size correction, while the Median in Table 8 does not. Please add a comment in the caption and use a different symbol for the uncorrected quantity.

As described in all Table captions, the plot conventions are given in the first part of Section 6 (clickable). There, the finite-size corrections are also explained. However, to make the distinction between finite and infinite volume clearer, we added Lref to the figure labels, where appropriate.

comment r4j14

> structure dependent contributions by far negligible [42,43]

This should be replaced by a bound or number that the authors assume (number and error) and also where in the papers this bound comes from. From the papers it seems to me that a sea-quark effect is neglected (sea-quark emitting a photon).

Structure-dependent amplitudes describing the emission of real photons from hadronic states are parameterized, within the Standard Model, in terms of vector and axial-vector form factors. The authors of {Desiderio:2020oej} have provided non-perturbative determinations of these form factors also for the case of the pion decay of interest to us, and in {Frezzotti:2020bfa} they have used them to explicitly quantify the impact of structure-dependent contributions with respect to the point-like term for a series of processes. For the muonic pion decay rate relevant here, Table III of {Frezzotti:2020bfa} shows that the structure-dependent and interference (between the point-like and the structure-dependent amplitudes) corrections are about **eight and six orders of magnitude smaller** than the value $\delta R_\pi = 0.0153(19)$ for real-photon-emission contributions in the point-like meson approximation, respectively.

Although the form factors of {Desiderio:2020oej} were calculated in the electro-quenched approximation, the contributions in which real photons are emitted by sea quarks vanish in the $SU(3)$ -flavor-symmetric limit and are therefore subleading compared to the valence effects included. A recent lattice calculation for kaon radiative leptonic decays {DiPalma:2025iud} confirms that these quark-disconnected contributions have a negligible impact, within the current uncertainties of the quark-connected contributions, on the vector and axial-vector form factors. Even very conservative estimates of the electro-unquenched effects on the real-photon-emission contributions would have no impact on the value of Equation (19) adopted.

We added a footnote in Section 3.3 (Electromagnetic effects) reflecting this discussion.

comment r4j15

> the first full determination

It seems incomplete to me due to the previous point. Note that I still think the approximate determination is very good progress. But full is too much.

We removed the “first full determination”.

comment r4j16

Where has the dominance of sea-sea contributions over sea-valence been "confirmed in several observables". Please give references and numbers.

We added more details to the sentence in Section 3.3 (Electromagnetic effects):

This flavour suppression of the sea-valence contributions compared to the sea-sea ones is seen in several observables in our 2020 work {Borsanyi:2020mff}. There, we call these contributions F and G , and the smallness of F compared to G is apparent in the case of hadron masses in Figures 20 and 21 and, in the case of a_μ , in Figures 25 and 26.

comment r4j17

In (17) and below,

$\Gamma(\pi \rightarrow \mu \nu)$ needs to include photons: $\Gamma(\pi \rightarrow \mu \nu \gamma)$ or so.

We fixed this.

comment r4j18

Where does the scale come from in (20,21)? It seems to come from F_{ud} which is confusing since it is determined here, and various different F_{ud} (with different isospin corrections) are present.

The numbers in Equations (20, 21) are from References {Giusti:2017dmp,DiCarlo:2019thl}. To compute those, the authors determined the valence-valence EM contribution of those masses from the ratios $[M_{ud}]_{\text{qed,vv}}/[F_{ud}]_{\text{qed}}$, ... just like in Equation (19) for the pion decay rate. From these ratios, and the physical qcd+qed values of the masses and the decay rate, one gets the numbers given in Equation (20, 21). Note that in the denominator, one could replace $[F_{ud}]_{\text{qed}}$ by any other variant of F_{ud} , since it changes the ratio by $O(\alpha^2)$ effects, that are neglected here. Also note that if only valence qed is turned on, as was done in References {Giusti:2017dmp,DiCarlo:2019thl}, no ultraviolet divergence is introduced in the scale and one can use the same lattice spacing with qed as without.

comment r4j19

The text after (21)

It would help to write the text after (21) in terms of a simple formula. In fact it would help to start the section by a split of F_{ud} as in (23) and then give and explain formulae and numbers for each piece.

We added a simple formula to clarify the text after Equation (21).

comment r4j20

(23)

F_{ud} does not seem to be the same on the two sides of the equation.

They denote the same quantity on both sides. Note, F without any index is just a fit coefficient related to the sea-valence effects, and it is not the pion decay rate F_{ud} .

comment r4j21

(25)

Do the symbols such as F_{ud} have the same meaning as in (23)? The definition of [...]''02 is also not given by (16), which is for a path integral expectation value $\langle 0 \rangle$. Which quantities in the equation have divergences?

Yes, F_{ud} in (26) has the same meaning as in (24). Equation (26) can be obtained from Equation (24) by applying differentiation with respect to the sea electromagnetic charge.

The referee is right about the derivative; our notation is just too terse. After Equation (16), we added an equation that defines these sorts of derivations, which we refer to after Equation (24). We hope this clarifies the presentation.

EM derivatives with fixed bare quark masses have divergences due to divergent EM contributions to the bare quark masses. In Equation (26) the B and C terms remove these divergences from $[w_0 F_{ud}]''_{02}$.

comment r4j22

r4n3: sect. 5.1 after (43)

Fits with just A' do not at all have the correct form for small a (the leading term at small a is $\sim a^2$). They can't be trusted at all. This holds in particular for quantities sensitive to the scale (larger distance windows) since also numerically significant a^2 terms are seen in w_0 (note the difference between "Wilson flow" and "Zeuthen flow").

We updated our fit procedure by removing fits with Δ_{KS} terms, as discussed in r4j09.

We thank the referees for their valuable comments and suggestions. We have accepted all of them and have revised the paper accordingly. The comments are reprinted here in blue in the order they were made. After each comment, we detail the resulting modifications made to the manuscript.

Answers to Referee 1

comment r1k00

1) On line 35 I suggest adding a 'now' before 'using lattice results' . I think this would make clearer the differences in the theory consensus picture from 2020 to 2025 for the general reader.

We followed the editor's suggestion to shorten the abstract so that the 2020 theory consensus no longer appears.

comment r1k01

2) New Figure 14 in the Supplementary Information: It looks as if an incorrect value has been plotted here for the FNAL/HPQCD/MILC '25 point. The caption says that the values are in the WP25 scheme, but then the FNAL/HPQCD/MILC central value should be just on the edge of the purple band (as in the comparison plot in the theory white paper).

It would be useful in the caption to note that the purple band is the average from the theory white paper.

We thank the referee for pointing out this mistake: we had accidentally plotted a value slightly different from the correct FNAL/HPQCD/MILC result. We have fixed this and also added a note in the caption, as requested.

Answers to Referee 3

comment r3k00

(2) I also appreciate the efforts to improve the discussion of the differences with the earlier results obtained by the BMW calculation, presented in subsections 6.7 and 6.9. I have one final request on this issue, which concerns the statement in the main body of the paper (lines 205ff), which reads "The difference between our current and the 2020 result, accounting for correlations among uncertainties, is $7.6(5.2)\times 10^{-10}$"

This statement is at odds with the explanation on p. 50 of the Supplemental Material where the uncertainty on the difference is quoted as 5.2 and 4.5, respectively, when considering either zero or full correlations arising from the continuum extrapolation and remaining systematics.

We revised the statement in the main text to be consistent with the Supplementary Information:

is 7.6×10^{-10} , with an uncertainty of 5.2×10^{-10} , indicating that the new result is 1.5σ higher. To obtain that result, we assume zero correlation among some of the systematics. When assuming full correlation, the uncertainty becomes 4.5×10^{-10} , and in this case, the new result is 1.7σ higher.

Answers to Referee 4

comment r4k00

My concerns are about section 3.4. Before turning to it, in eq. (17), the definition of $[AB]''$ is confusing. It invites the reader to insert, e.g., $[w_0 Fud]''$ and neither w_0 nor Fud are path integral expectation values. They are functions of many path integral expectation values.

We added an extra sentence and a formula after Equation (S17) explaining the case of functions of path integral expectation values.

comment r4k01

In order to save writing, I simplify and consider $\mu = m_d = m_s$ and all quarks have the same charge. I also do not distinguish valence and sea contributions. Also $F = Fud$.

(24) then simplifies

$$w_0 F = A + B \frac{M^2}{F^2} + E e^2 (*)$$

If F is the same on the two sides of the equation, it necessarily has all orders in e^2 , while the equation is supposed to be an expansion in e^2 . The added sentence at the end of the paragraph makes no sense to me. My remark r4j21 remains. If I ignore the author's answer and this sentence in the paper and assume that the M^2/F^2 term is to be expanded in e^2 , I get an additional term, $[M^2/F^2]'' e^2$ which does not seem to be accounted for in the text and analysis.

We added a footnote explaining why no term of the form $e^2[M^2/F^2]''$ is missing from our analysis.

comment r4k02

(25) simplifies

$$w_0 f = A + B \frac{m^2}{f^2} (**)$$

where f, m are F, M but in pure QCD. The text says that A, B are the same in these equations (*) and (**). After (26) it is said that B in (26) is taken from (25), i.e. (**). It is then finite. But right afterwards it is said that B absorbs a divergence. This is contradictory.

We added a footnote explaining that the B coefficient itself is finite, and that the divergence appears in the factor multiplying B .

comment r4k03

This is confusing. I did not find a gluonic scale in [44]. I only found f_{π} . Also in (24,25,26,28) the meson masses are made dimensionless by decay constants. Does this fit together?

We have removed the confusing statement. Instead, we added a new paragraph, explaining the scale setting of Reference {DiCarlo:2019thl}:

We also note that, in Ref. {DiCarlo:2019thl}, the authors used the QCD pion decay constant from Equation (S23) to set the scale for their calculations of the QCD++ components of the masses in Equations (S21) and (S22). An alternative scale choice—differing from the QCD pion decay constant by isospin corrections—would only impact the results in Equations (S21) and (S22) at the level of neglected $O(e^4)$ terms. This is because the measurement specifically targeted the valence QED component of these masses, which is an $O(e^2)$ quantity.

comment r4k04

In summary, I need to ask the authors to rewrite the section such that one can understand what they have done, such that there are no ambiguities, and of course explain why it is correct. The present text does not.

We have substantially rewritten the section, especially between Equations (S24) and (S26). The new version includes much more detail than the previous one.

1 Summary

I acknowledge that the authors have both made a serious effort to take some comments of the first reviews into account and have all-in-all also succeeded in doing this. However, a few problems remain and in particular it remains impossible to judge whether the uncertainties of the various continuum limits are adequately accounted for. I list those replies and changes where I suggest or require further improvements.

2 Comments on various replies

Comment on reply to r4i01

Some references have been added.

Comment on reply to r4i02

We improved the transparency by adding many new plots, tables and corresponding discussions into Section 6 (Window observables).

Yes, more information has been added, in particular in section 6 (I always refer to the new sect./fig./tab. numbers.). Unfortunately this was not done everywhere and in particular not where it matters most for the final result.

For example a big part of the result is **the 12-28 window**, Table 14, and its uncertainty dominates the total uncertainty. In this case the only information concerning the weight of fits is which fit has the highest weight. This tells very little. The color code introduced in other figures is not given here. Thus this piece and not only it remains rather intransparent.

If fits with polynomials of Δ_{KS} do dominate the 12-28 window (I can't know because the information is not given), then the result is rather doubtful. According to the table, w_0 plays an important role. The $O(a^2)$ terms in w_0 are significant as we see from the difference between different flows in Table 5. Therefore fits without a^2 terms make little sense.

Comment on reply to r4i03

I comment on the changes added to main.

I see no improvement.

Data availability.

The datasets for the figures and tables are available from the corresponding author on request.

1. Why does one introduce the necessity to write to the author? What if he is not available for a longer period? The numbers just need to be listed.
2. In several cases, the plotted numbers are not sufficient. Examples are Tab 5: numbers for w_0/a , $f_{ud}a$; A list of the fits with their weights down to a reasonable cut (on the weights).

Upon request from the PI the configurations can be given out for specific projects.

This is an empty statement as there is no commitment to anything. I request to remove it.

Code availability.

A CPU-code for configuration production and measurements can be obtained from the corresponding author upon request. The Wilson flow evolution code, which was used to determine w_0 , can be downloaded from <https://arxiv.org/abs/1203.4469>.

Do the authors mean that **the code which was used in this project** for the configuration production and measurements can be obtained? Then write it. Otherwise remove the statement since several production codes of other collaborations are publicly available.

Comment on reply to r4i07

Note, that the statistical error (a) refers to that of the isospin-symmetric contribution in finite volume. The finite-size (b) and isospin-breaking (e) errors also contain a statistical component.

It is easy to give the complete information: "... The finite-size (b) and isospin-breaking (e) errors also contain statistical components of xxx % and yyy % respectively.

I request this change.

Comment on reply to r4i11

Referring to r4i24 does not help. My concern was also due to $\beta = 3.7753$. No evidence is shown that the extrapolation to the physical point works in that case and one wonders why Δ_{KS} is not relevant there. Actually the $\beta = 3.7753$ data look odd in the larger distance windows in section 6. How much influence this has, e.g., in the continuum extrapolation in Table 14 is impossible to see since the plotted data is not available, the scale of the figure is large compared to the precision and there is no information which fits dominate.

Comment on reply to r4i12

The added information is very useful and reassuring. ... On our finest lattice we find $\chi = 0.0358(29)fm^4$ for the unimproved and $\chi = 0.0299(24)fm^4$ for the improved susceptibility. These numbers nicely fit on the continuum extrapolation curves presented in Ref. Borsanyi:2016ksw. ...

I think the authors mean the curve for $T = 150$ MeV. Please add that information or the figure number.

Comment on reply to r4i13

In Figure 3 in Section 1.2 (Taste violation) beside a^4 we now also plot $\alpha^n a^2$ with $n = 3$ and $n = 5$...

The Fig. could easily be improved by plotting the curves close to the data.

Comment on reply to r4i15

We added ...

This has been improved a bit.

Comment on reply to r4i16

... and also on the lattice Hackl:2024whw. If we compare them, it seems that chiral perturbation theory overestimates the contamination in nucleon propagators.

The reference Hackl:2024whw seems somewhat arbitrary as there has been a long debate on the question. I would avoid going into this discussion or cite a review. The final sentence is a speculation ("it seems") and it is unclear what is compared. Again, I advise to drop this part of the discussion.

Comment on reply to r4i17

The caption of the new Fig. 15 is missing the second β value or should not contain the one given.

Comment on reply to r4i22

We updated many of our continuum extrapolation plots by adding a color coding scheme: fits with higher/lower AIC values are now shown with a lighter/darker color.

The presentation of the results is somewhat improved, but it is still almost impossible to tell, e.g. which fits make up 70% of the weights or which five fits are the most important ... It is easy to improve it further by explicitly listing the fits with their weights down to a certain cut of a minimal weight. It may also be possible to use a better color coding ranging from red to blue or so and to adjust the scale of the figures such that lines are better resolved.

Furthermore, as remarked above essential continuum extrapolation plots are not improved at all.

Comment on reply to r4i23

We agree with the referee that the true function is complicated with many powers of a and α , but such functions are not practical in case of noisy data points and limited number of lattice spacings, since they tend to overfit the data. We therefore work with low-order polynomials, most of which produce reasonable fit qualities, e.g. in case of all the relevant single-window fits for a at least one third of our fits have fit quality over 0.1.

The authors seem to miss a basic point: A wrong function may fit the data well (have a good P-value) but yield the wrong extrapolation. The P-value only tests the function where one has data. Therefore the P-value has to be good, but does not guarantee the extrapolation is (about) correct. The latter has to be made

plausible by theoretical considerations on the fit-function. One reason to show what dominates the results is to allow the reader to judge whether reasonable fit-functions are used there. In particular a polynomial of Δ_{KS} without an a^2 term is very hard to believe.

Comment on reply to r4i24

In the reply an example of dropping lattice spacings is given for one specific case. The tables show different numbers of lattice spacings for different observables, ranging from 5 to 7. The reader would like to see data points for all lattice spacings, even if some are not used in the analysis.

Comment on reply to r4i25a

prefer to keep also the pure \$\Delta_{KS}\$ fits and let the AIC decide, whether it is a reliable description of the data or not.

The AIC can only decide (if at all) whether the fit is a good description of the data in the range where there are data. The extrapolated value depends on the fit-functions inserted into the AIC. Therefore the above statement does not help at all.

This fundamental problem in the continuum extrapolations remains

Comment on reply to r4i24b

Same issue as r4i24a.

Comment on reply to r4i26

I appreciate the comparison to perturbation theory. The results for the short distance window are now much more plausible. However, the functional forms in eq. (49) are ad hoc. Why is there no a^3 term? ¹ Why is there $a^2 \log(a/w_0)$ and not $a^2 \log(a/w_0)^k$ with some other power k ? The minimum change required is to write clearly around (49) that the true (asymptotic) form is unknown since there is no theory analogous to Symanzik's theory for discretisation errors and that consequently the choices of fit functions are purely phenomenological.

Comment on reply to r4i29

We currently follow the approach of most other groups, like RBC-UKQCD, FNAL-HPQCD- MILC, ETMC, ... The Theory Initiative requires only to produce the all-flavory result for three windows,...

There are other groups listing more results. More importantly it is disappointing to see that for the important $g_\mu - 2$ the amount of information published will only allow for some very specific crosschecks. It is up to the editors of Nature to decide what is required for this journal.

¹Note that an a^3 term would result from the mathematically wrong discussion cited by the authors.

3 Other comments

r4n1

The "This work" point in the comparison figure 1 and in Table 8 contains the finite size correction, while the Median in Table 8 does not. Please add a comment in the caption and use a different symbol for the uncorrected quantity.

r4n2

The new section 3

It is very good to see this new determination. Unfortunately I am not really able to follow the theoretical discussion without reading exactly what was done in [42-44] and working it out myself. Therefore I can't fully judge the result. I do not have a real doubt about the result but at least the text needs to be improved such that one can follow. I list some details (I can't refer to line numbers but go through in order and the blue pieces should help to identify the text).

structure dependent contributions by far negligible [42,43]

This should be replaced by a bound or number that the authors assume (number and error) and also where in the papers this bound comes from. From the papers it seems to me that a sea-quark effect is neglected (sea-quark emitting a photon).

the first full determination

It seems incomplete to me due to the previous point. Note that I still think the approximate determination is very good progress. But full is too much.

Where has the dominance of sea-sea contributions over sea-valence been "confirmed in several observables". Please give references and numbers.

In (17) and below,

$\Gamma(\pi \rightarrow \mu\nu_\mu)$ needs to include photons: $\Gamma(\pi \rightarrow \mu\nu_\mu\gamma)$ or so.

Where does the scale come from in (20,21)? It seems to come from F_{ud} which is confusing since it is determined here, and various different F_{ud} (with different isospin corrections) are present.

The text after (21)

It would help to write the text after (21) in terms of a simple formula. In fact it would help to start the section by a split of F_{ud} as in (23) and then give and explain formulae and numbers for each piece.

(23)

F_{ud} does not seem to be the same on the two sides of the equation.

(25)

Do the symbols such as F_{ud} have the same meaning as in (23)? The definition of $[\dots]''_{02}$ is also not given by (16), which is for a path integral expectation value $\langle O \rangle$. Which quantities in the equation have divergences?

r4n3: sect. 5.1 after (43)

Fits with just A' do not at all have the correct form for small a (the leading term at small a is $\sim a^2$). They can't be trusted at all. This holds in particular for quantities sensitive to the scale (larger distance windows) since also numerically significant a^2 terms are seen in w_0 (note the difference between "Wilson flow" and "Zeuthen flow").

I am satisfied with the largest part of the replies and the changes made by the authors. In the following I just discuss the part that I am not satisfied with.

My concerns are about section 3.4. Before turning to it, in eq. (17), the definition of $[AB]''$ is confusing. It invites the reader to insert, e.g., $[w_0 F_{ud}]''$ and neither w_0 nor F_{ud} are path integral expectation values. They are functions of many path integral expectation values.

Section 3.4

I)

In order to save writing, I simplify and consider $m_u = m_d = m_s$ and all quarks have the same charge. I also do not distinguish valence and sea contributions. Also $F = F_{ud}$.

(24) then simplifies

$$w_0 F = A + \bar{B} F^{-2} M^2 + \bar{E} e^2 \quad (*)$$

If F is the same on the two sides of the equation, it necessarily has all orders in e^2 , while the equation is supposed to be an expansion in e^2 . The added sentence at the end of the paragraph makes no sense to me. My remark **r4j21** remains. If I ignore the author's answer and this sentence in the paper and assume that the $F^{-2} M^2$ term is to be expanded in e^2 , I get an additional term, $[F^{-2} M^2]'' e^2$ which does not seem to be accounted for in the text and analysis.

II)

(25) simplifies

$$w_0 f = A + \bar{B} f^{-2} m^2 \quad (**)$$

where f, m are F, M but in pure QCD. The text says that A, \bar{B} are the same in these equations (*) and (**). After (26) it is said that \bar{B} in (26) is taken from (25), i.e. (**). It is then finite. But right afterwards it is said that \bar{B} absorbs a divergence. This is contradictory.

III)

After (28) there is the text

For the valence-valence QED part, we note that the ratio in Equation (20) does not depend on the lattice scale, assuming that scale is free of valence-valence QED contributions. This is true for w_0 , since it is defined in terms of purely gluonic quantities. (Concretely, the authors of [44] assume that the scale does not change when the valence quark charge is turned on.)

This is confusing. I did not find a gluonic scale in [44]. I only found f_π . Also in (24,25,26,28) the meson masses are made dimensionless by decay

constants. Does this fit together?

With these contradictions in the text I am getting worried that it is not just the text that is inconsistent or at least unclear but rather some term might be missing in the computation.

In summary, I need to ask the authors to rewrite the section such that one can understand what they have done, such that there are no ambiguities, and of course explain why it is correct. The present text does not.